# Hydrogen peroxide is required for light-induced stomatal opening across different plant species

Wen Shi[1], Yue Liu [1,4], Na Zhao[1], Lianmei Yao[1], Jinge Li[2], Min Fan [1], Bojian Zhong [3], Ming-Yi Bai [1] & Chao Han [1]✉

Stomatal movement is vital for plants to exchange gases and adaption to terrestrial habitats, which is regulated by environmental and phytohormonal signals. Here, we demonstrate that hydrogen peroxide ($H_2O_2$) is required for light-induced stomatal opening. $H_2O_2$ accumulates specifically in guard cells even when plants are under unstressed conditions. Reducing $H_2O_2$ content through chemical treatments or genetic manipulations results in impaired stomatal opening in response to light. This phenomenon is observed across different plant species, including lycopodium, fern, and monocotyledonous wheat. Additionally, we show that $H_2O_2$ induces the nuclear localization of KIN10 protein, the catalytic subunit of plant energy sensor SnRK1. The nuclear-localized KIN10 interacts with and phosphorylates the bZIP transcription factor bZIP30, leading to the formation of a heterodimer between bZIP30 and BRASSINAZOLE-RESISTANT1 (BZR1), the master regulator of brassinosteroid signaling. This heterodimer complex activates the expression of *amylase*, which enables guard cell starch degradation and promotes stomatal opening. Overall, these findings suggest that $H_2O_2$ plays a critical role in light-induced stomatal opening across different plant species.

The acquisition of stomata is the key innovation that benefits land plants spreading to different terrestrial environments. Stomata, which are surrounded by a pair of guard cells, are present in almost all types of vascular plants and bryophytes, including mosses and hornworts[1,2]. It has been well established that stomata in vascular plants open to acquire carbon dioxide in response to light and close to prevent water loss in response to high concentration of $CO_2$ and the phytohormone abscisic acid (ABA), optimizing carbon gain per unit water loss and allowing plants to colonize drier regions of the earth[1,3]. The stomata are opened when the volumes of guard cells are changed by exposure to light. One factor contributing to this phenomenon is the hyperpolarization of the plasma membrane driven by H⁺-ATPase (AHA) and blue light signaling[4,5]. The activation of AHA promotes acidification and potassium ions uptake

in guard cells, causing water to flow into the guard cell and providing turgor pressure to induce stomatal opening[6,7]. In addition, red light increases photosynthesis and causes a decrease of intercellular $CO_2$ concentration in leaves, which promotes stomatal opening through two Raf-like protein kinases Convergence of Blue light and $CO_2$ 1/2 (CBC1/2) and High leaf Temperature 1 (HT1)[8,9]. The rapid degradation of starch in guard cells upon light exposure is also necessary for stomatal opening[10,11]. Starch degradation in guard cells depends on β-AMYLASE1 (BAM1) and α-AMYLASE3 (AMY3), glucan hydrolases expressed specifically in guard cells, thereby directly producing glucose[11,12]. The *bam1amy3* mutant does not exhibit altered blue light-dependent ion transport, but delays the fast stomatal opening kinetics compared with the wild-type plants, which are achieved by the failure to replenish the

[1]The Key Laboratory of Plant Development and Environmental Adaptation Biology, Ministry of Education, School of Life Sciences, Shandong University, Qingdao 266237, China. [2]Shandong Provincial Key Laboratory of Plant Stress, College of Life Sciences, Shandong Normal University, Jinan 250358 Shandong, China. [3]College of Life Sciences, Nanjing Normal University, Nanjing 210023, China. [4]Present address: School of Agriculture and Biology, Shanghai Jiao Tong University, Shanghai 200240, China. ✉e-mail: hanchao@sdu.edu.cn

carbohydrate pool from starch degradation during stomatal opening[12]. Recent studies have shown that a sugar supply from mesophyll cells and an ATP supply from the cytosol are required for red light-induced starch biosynthesis in guard cell chloroplasts, which are also important for light-induced stomatal opening[13,14].

BAM1 has been confirmed as a regulatory target that modulates starch degradation in guard cells. Brassinosteroid (BR) signaling plays an important role in regulating starch degradation in guard cells through the core transcription factor BRASSINAZOLE-RESISTANT1 (BZR1), which can bind directly to *BAM1* promoter and increase *BAM1* expression[15]. Sugar supplementation and the Target of Rapamycin (TOR) kinase promote *BAM1* expression by stabilizing BZR1[16,17]. Light exposure at dawn can induce BAM1 protein accumulation in guard cells, depending on TOR kinase activity. TOR kinase promotes guard cell starch degradation and light-induced stomatal opening by phosphorylating BAM1 protein at Serine 31 in vivo, conferring stability to BAM1 protein[17].

Sucrose Non-Fermenting 1 related protein kinase 1 (SnRK1) heterotrimeric complex comprises a catalytic subunit (KIN10 or KIN11) and two regulatory subunits, namely KINβ with three isoforms (KINβ1, KINβ2, and KINβ3) and an atypical KINβγ[18]. In plant cells, SnRK1 complex interacts with TOR complex and plays an antagonistic role in TOR kinase[19,20]. Insufficient energy supply, induced by long periods of darkness or hypoxia, prompts the translocation of KIN10 subunit into nucleus to phosphorylate C-class bZIP transcription factors and regulate a series of gene expression involved in energy-consuming catabolic processes[21,22]. KINβ1 and KINβ2 possess membrane-associated myristoylation modification, which can restrict the nuclear localization of catalytic KIN10[23]. In Arabidopsis stomatal lineage cells, KIN10 localizes within the nucleus, interacting with and phosphorylating SPEECHLESS (SPCH) transcription factor to promote stomatal development[24]. Hydrogen peroxide ($H_2O_2$) accumulates remarkably in meristemoids and induces the nuclear localization of KIN10 by interfering with the interaction between KIN10 and KINβ2[25].

The earliest plants colonized land approximately 450 million years ago and encountered high concentrations of molecular oxygen[1,26]. The increase in molecular oxygen forced ancient plants to adapt to the greater abundance of reactive oxygen species (ROS) molecules[26,27]. $H_2O_2$ is a stable type of ROS in eukaryotic cells, which is controlled by a specific metabolic system and is utilized as a signaling molecule among different organs in modern plants[28]. $H_2O_2$ exhibits the spatial distribution at tissue and cellular levels to modulate plant growth and development. In the plant apical meristem, $H_2O_2$ is mainly enriched in differentiated tissues such as the root elongation zone or the shoot peripheral zone to promote cell differentiation[29–31]. In the Arabidopsis epidermis, $H_2O_2$ is highly abundant in meristemoids after asymmetric division, which is essential for the differentiation of meristemoids into stomata[25]. These specific distributing patterns of $H_2O_2$ depend on the regulating of $H_2O_2$ scavenging gene expression by specific transcriptional factors. The basic helix-loop-helix (bHLH) transcription factor UPBEAT1 (UPB1) determines the $H_2O_2$ distribution pattern in the roots by controlling specific peroxidase genes (*PRX39*, *PRX40* and *PRX57*)[29]. SPCH is responsible for establishing the $H_2O_2$ spatial pattern in the leaf epidermis during stomatal development by regulating the expression of *Ascorbate Peroxidase 1* and *Catalase 2* (*CAT2*)[25]. However, the mechanism by which $H_2O_2$ regulates the function of differentiated cells remains poorly understood.

In this study, we have demonstrated that $H_2O_2$ plays a crucial role in light-induced stomatal opening. Specifically, $H_2O_2$ accumulates within guard cells even under normal growth conditions. Removal of $H_2O_2$ by chemical treatments or genetic manipulations leads to the impaired light-induced stomatal opening. Within guard cells, $H_2O_2$ triggers the localization of KIN10 in nucleus, where KIN10 interacts with and phosphorylates bZIP30 to promote the formation of a heterodimer between bZIP30 and BZR1, subsequently inducing the expression of *BAM1* and *AMY3*. Furthermore, we showed that the requirement for $H_2O_2$ in light-induced stomatal opening is widespread across different plant species. These findings provide valuable insights into the molecular mechanisms through which $H_2O_2$ promotes stomatal opening in response to light in different plant species.

## Results
### Specific accumulation of $H_2O_2$ in guard cells under normal growth conditions is required for light-induced stomatal opening

Our previous study has shown that $H_2O_2$ accumulates specifically in meristemoids and guard cells, whereas it is less abundant in pavement cells of intact leaves in plants grown under normal growth conditions. The accumulation of $H_2O_2$ in meristemoids has been shown to promote stomatal development[25]. However, the precise role of $H_2O_2$ accumulation in guard cells on stomatal movement during normal growth conditions has yet been fully understood. To investigate the regulatory function of $H_2O_2$ in guard cell activity, we initially analyzed the $H_2O_2$ content in guard cells of the intact leaves using various techniques, including fluorescent dye 2′,7′- dichlorodihydrofluorescein diacetate ($H_2DCFDA$) and BES-$H_2O_2$-Ac and genetically encoded probe Hyper. The results confirmed a significant accumulation of $H_2O_2$ in guard cells (Fig. 1a–f). To further understand the function of enriched $H_2O_2$ in guard cell movement, potassium iodide (KI), as a scavenger of reactive oxygen species was used to remove accumulated $H_2O_2$ in guard cells. The results revealed that light promotes stomatal opening, but such effects were remarked reduced in the presence of KI (Supplementary Fig. 1a–c). Additionally, the application of diethyldithiocarbamic acid (DDC), an inhibitor of $H_2O_2$ production, also diminished the light-induced stomatal opening (Supplementary Fig. 1e–g). To strengthen the pharmacological findings, we further examined the impact of $H_2O_2$ on light-induced stomatal opening using mutants of *RBOHD and RBOHF*, which encode respiratory burst NADPH oxidase homologs (RBOHs) and are responsible for producing apoplastic $H_2O_2$ during plant defense response[32], as well as *CAT2* overexpressing plants (*p35S::CAT2-myc*) with deficiencies in $H_2O_2$ production, and the *cat2* mutant known for its higher $H_2O_2$ accumulation (Fig. 1g, h, Supplementary Fig. 2a, b). The results showed that *cat2* mutant displayed larger stomatal apertures under both the end-of-night and 1 hour light exposure conditions. Therefore, *cat2* mutant exhibit a weaker response to light exposure compared to wild-type plants. Both *p35S::CAT2-myc* and *rbohD rbohF* mutant exhibited impaired stomatal opening upon light exposure (Fig. 1i, j, Supplementary Fig. 2c). These results suggested that accumulation of $H_2O_2$ in guard cells under unstressed conditions is required for light-induced stomatal opening.

Given the importance of guard cell starch breakdown for light-induced stomatal opening, we speculated that $H_2O_2$ enrichment in guard cells might be involved in light-induced starch degradation in guard cells. To test this hypothesis, we analyzed the guard cell starch metabolism in various materials with different levels of $H_2O_2$ in guard cells. The results showed that the reducing $H_2O_2$ through overexpression of *CAT2*, mutations of *RBOHD* and *RBOHF* or treatment with KI or DDC significantly hindered guard cell starch degradation upon exposure to light (Fig. 1k, l, Supplementary Fig. 1d, h, Supplementary Fig. 2d). Conversely, the *cat2* mutant showed the decreased starch content in guard cells under both the end of night and 1 hour light exposure conditions (Fig. 1k, i, Supplementary Fig. 2d). *BAM1* is responsible for guard cell starch degradation and expressed in guard cells specifically. Quantitative RT-PCR analysis using guard cell enriched samples showed that the expression levels of *BAM1* in guard cells significantly increased in the *cat2* mutant, but decreased in *p35S::CAT2-myc* and *rbohD rbohF* mutants (Fig. 1m). These findings based on starch metabolism phenotype and qRT-PCR analysis indicated that $H_2O_2$ promotes the light-induced starch degradation in guard cells through inducing the expression of *BAM1*.

To investigate the precise role of $H_2O_2$ in stomatal movement, we conducted an analysis of stomatal apertures under varying concentrations of $H_2O_2$ in wild-type plants. We found that as the concentration of exogenous $H_2O_2$ increased, there was a gradual increase in the fluorescent intensity of H2DCFDA staining (Supplementary Fig. 3a, b). Specifically, treatment with concentrations of

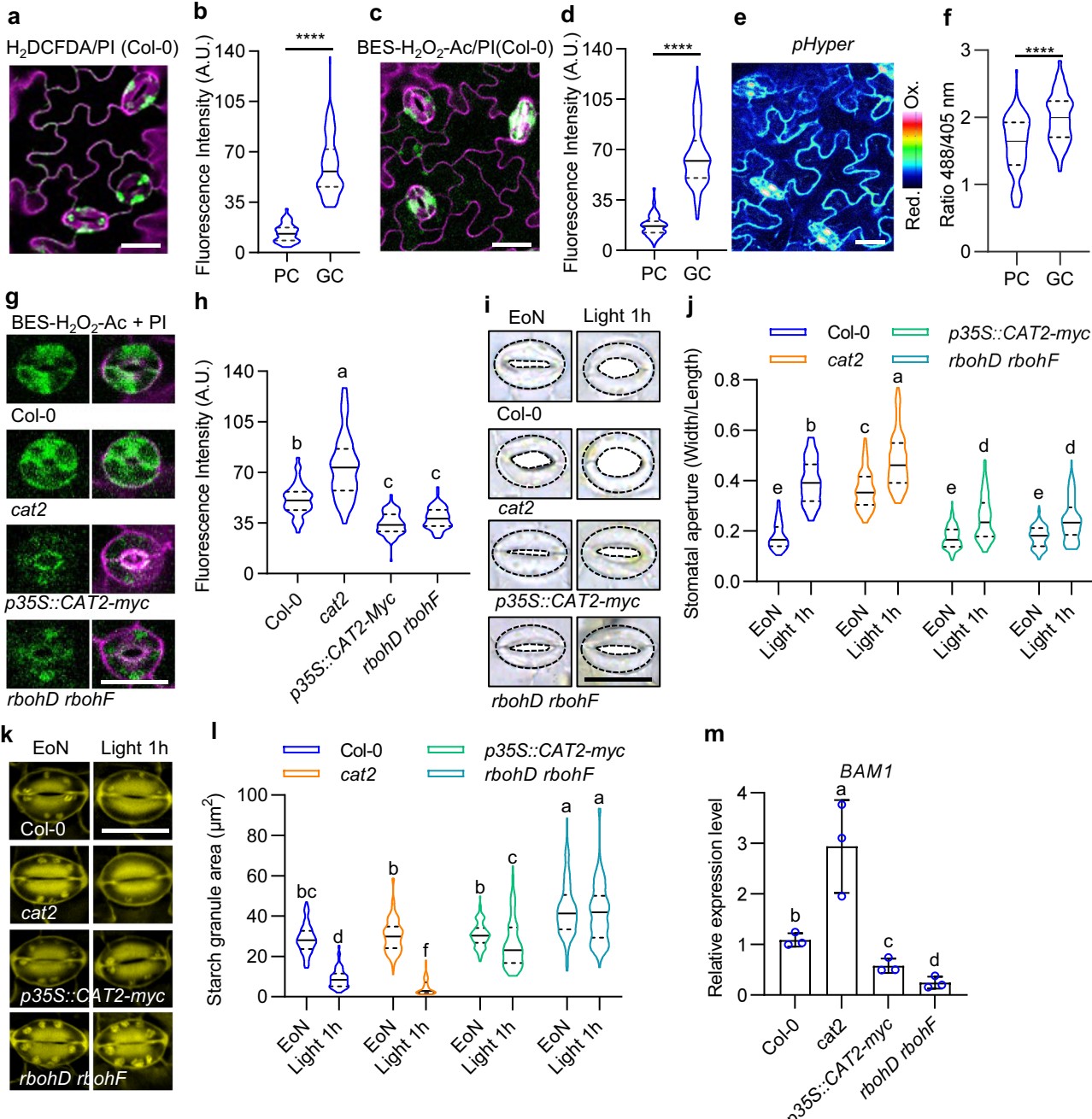

**Fig. 1 | Specific accumulation of $H_2O_2$ in guard cells under normal conditions is required for light-induced stomatal opening in Arabidopsis. a–f** Measurement of $H_2O_2$ in the epidermal cells on rosette leaves of 4-weeks-old Col-0 plants using H2DCFDA staining (**a, b**), BES-$H_2O_2$-Ac staining (**c, d**) and HyPer fluorescent (**e, f**). Col-0 or *pHyPer* transgenic line were grown on 1/2 MS medium under a 12 h light/12 h dark photoperiod with a light intensity of 100 μM m$^{-2}$ s$^{-1}$ for 28 days. The intensities of H2DCFDA, BES-$H_2O_2$-Ac, and HyPer fluorescent signals were analyzed using at least 100 guard cells from 10 different plants with ImageJ software. "PC" in confocal images represents pavement cell, "GC" represents guard cell.
**g–l** Measurement of $H_2O_2$ content using BES-$H_2O_2$-Ac staining (**g, h**) quantification of stomatal apertures (**i, j**) and starch granules (**k, l**) in guard cells, qRT-PCR analysis of the expression of *BAM1* (**m**) on rosette leaves of Col-0, *cat2, p35S::CAT2-myc, rbohD rbohF* plants. Plants were grown on 1/2 MS medium under a 12 h light/12 h dark photoperiod with the 100 μM m$^{-2}$ s$^{-1}$ light intensity for 28 days. BES-$H_2O_2$-Ac

signaling intensities, starch granules area, and the ratio of stomatal aperture width to length in at least 100 guard cells from at least 4 different plants were analyzed using ImageJ software. EoN means the end of night, and Light 1 h means the white light illumination for 1 hour after the end of night. The solid lines of violin plots represent the median; the dashed lines represent the first or third quartile. Error bars of (**m**) indicate standard deviation (S.D.). Scale bars in the pictures of stomata represent 20 μm. Asterisk and Different letters above the bars indicate statistically significant differences between samples. One-way ANOVA analysis followed by uncorrected Fisher's LSD multiple comparisons test ($p < 0.05$) were applied for $H_2O_2$ measurement of different plants; Two-way ANOVA analysis followed by Tukey's multiple comparisons test ($p < 0.05$) was applied for quantification of stomatal apertures and starch granules; Student's $t$ test was applied for cell type $H_2O_2$ content measurement (****$p < 0.0001$) and gene expression analysis ($p < 0.05$).

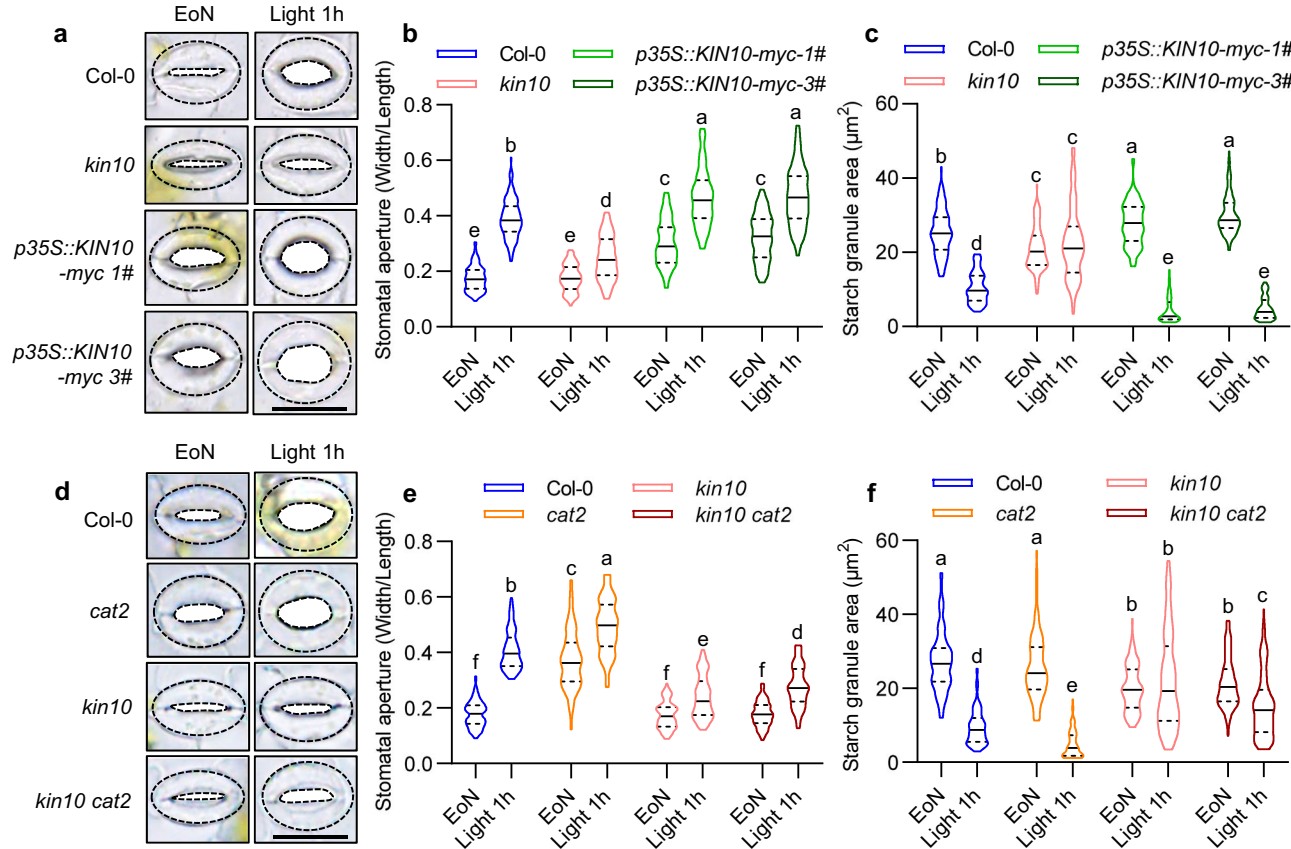

**Fig. 2 | KIN10 plays a crucial role for H$_2$O$_2$ mediated promotion of light-induced stomatal opening.** **a**–**c** Quantification of stomatal apertures (**a**, **b**) and guard cell starch granules (**c**) in the rosette leaves of Col-0, *kin10*, and *p35S::KIN10-myc* plants. **d**–**f** Quantification of stomatal apertures (**d**, **e**) and guard cell starch granules (**f**) in rosette leaves of Col-0, *kin10*, *cat2,* and *kin10 cat2* plants. All plants were grown on 1/2 MS medium under a 12 h light/12 h dark photoperiod with a 100 µM m$^{-2}$ s$^{-1}$ light intensity for 28 days. The starch granules area and the ratio of stomatal aperture width to length from more than 100 guard cells of at least 4 different plants were measured using ImageJ software. Scale bars in the pictures of stomatal images represent 20 µm. EoN means the end of night, and Light 1 h means the white light illumination for 1 hour after the end of night. The solid lines of violin plots represent median, the dashed lines represent first or third quartile. Different letters above the bars indicate statistically significant differences between samples (Two-way ANOVA analysis followed by Tukey's multiple comparisons test, $p < 0.05$).

10 µM or 30 µM of H$_2$O$_2$ resulted in larger stomatal apertures and less starch content in guard cells under both the end-of-night and 1 hour light exposure conditions. On the other hand, treatment with 100 µM or 1 mM H$_2$O$_2$ impaired stomatal opening and starch degradation in guard cells upon light exposure (Supplementary Fig. 3c, d). These results suggested that accumulation of H$_2$O$_2$ in guard cells under unstressed conditions is necessary for light-induced guard cell starch degradation and stomatal opening.

### KIN10 is required for the promoting effects of H$_2$O$_2$ on light-induced guard cell starch degradation and stomatal opening

Previous study showed that the specific accumulation of H$_2$O$_2$ in meristemoids promotes stomatal development by inducing the nuclear localization of KIN10 to phosphorylate and stabilize SPCH[25]. Therefore, we hypothesized that the specific accumulation of H$_2$O$_2$ in guard cells under normal growth conditions promotes light-induced guard cell starch degradation and stomatal opening by controlling KIN10 activity. To test this hypothesis, we first analyzed the guard cell starch content and stomatal apertures in the intact leaves of wild-type plants, *kin10* mutant, and *p35S::KIN10-myc* transgenic plants at the end of night and after exposure to light for 1 hour. The stomatal apertures in *kin10* mutant failed to open fully, whereas in the *p35S::KIN10-myc* transgenic lines, stomatal apertures were larger (Fig. 2a, b, Supplementary Fig. 4a). Furthermore, the starch levels in guard cells in *kin10* mutant remained lower than those in wild-type plants, and the starch was not degradable upon light exposure. However, in two of the

*p35S::KIN10-myc* transgenic lines, the guard cell starch exhibited higher levels when the end of the night and lower levels after light exposure (Fig. 2c, Supplementary Fig. 4b). To investigate whether KIN10 is involved in the H$_2$O$_2$-mediated guard cell starch breakdown and stomatal opening, we analyzed the guard cell starch content and stomatal apertures in the intact leaves of wild type, *kin10*, *cat2,* and *kin10 cat2* double mutants. The results showed that the loss-of-function mutation in KIN10 prevented guard cell starch degradation and stomatal opening, whereas the *cat2* mutant exhibited less starch content in guard cells and larger stomatal apertures after light exposure. The *kin10 cat2* double mutant displayed a similar phenotype to the *kin10* mutant in terms of stomatal movement and guard cell starch metabolism (Fig. 2d–f, Supplementary Fig. 4c, d). Meanwhile, the *kin10* mutant did not regulate H$_2$O$_2$ content in guard cells of either wild-type plants or *cat2* mutant (Supplementary fig. 5a, b). These results suggested that KIN10 plays the critical roles in H$_2$O$_2$-promoted guard cell starch degradation and stomatal opening.

To investigate whether H$_2$O$_2$ promotes starch degradation and stomatal opening in guard cells by inducing the nuclear localization of KIN10, we analyzed the subcellular localization of KIN10 in guard cells. The results revealed the pronounced nuclear localization signal of KIN10 in guard cells. Treatment with KI or overexpression of *CAT2*, as well as mutations in *RBOHD* and *RBOHF* to reduce H$_2$O$_2$ content in the guard cells, significantly reduced the nuclear distribution of KIN10. Conversely, in the guard cells of *cat2* mutants, KIN10 exhibited a more robust nuclear localization signal (Supplementary Fig. 6a–d).

Furthermore, treatment with exogenous $H_2O_2$ at varying concentrations resulted in an increased nuclear-to-cytoplasmic ratio of KIN10 protein. $H_2O_2$ ranging from $30\,\mu M$ $H_2O_2$ to $1\,mM$ $H_2O_2$ induced a similar nuclear-to-cytoplasmic ratio of KIN10 protein, indicating that the nuclear localization of KIN10 protein is sensitive to $H_2O_2$ (Supplementary Fig. 6e, f). These results suggested that $H_2O_2$ plays a vital role in promoting the nuclear localization of KIN10 in guard cells.

To further investigate the role of nuclear-localized KIN10 in $H_2O_2$-mediated starch degradation and stomatal opening, we created transgenic plants in the *kin10* mutant background with KIN10 fused to a nuclear localization signal sequence (NLS-KIN10) or a nuclear export signal sequence (NES-KIN10) at the N-terminus, under the control of the native *KIN10* promoter (Supplementary Fig. 7a). Gene expression analysis showed that both *pKIN10::KIN10-YFP/kin10* and *pKIN10::NLS-KIN10-YFP/kin10* successfully restored *BAM1* gene expression in *kin10* mutant. However, *pKIN10::NES-KIN10-YFP/kin10* failed to rescue the reduced *BAM1* expression in *kin10* mutant (Supplementary Fig. 7b). As expected, the *pKIN10::NLS-KIN10-YFP* plants exhibited lower guard cell starch content and increased stomatal apertures compared to wild-type plants and *pKIN10::KIN10-YFP/kin10* transgenic plants in response to light exposure. In contrast, the *pKIN10::NES-KIN10-YFP* plants showed cytosolic localization of KIN10 in guard cells, resulting in similar starch content and stomatal apertures as observed in the *kin10* mutant (Supplementary Fig. 7c, d). Previous studies have shown that KINβ2 subunit in SnRK1 complex can repress KIN10 nuclear localization[23]. The overexpression of *KINβ2* restricts the nuclear localization of KIN10 in guard cells (Supplementary Fig. 8a, b). Additionally, the *p35S::KINβ2-YFP* lines exhibit a constant level of starch in guard cells before and after 1 hour light exposure, even when treated with $30\,\mu M$ $H_2O_2$. Moreover, the stomatal apertures in *p35S::KINβ2-YFP* lines also failed to increase after light exposure (Supplementary Fig. 8c, d). These data suggested that the overexpression of *KINβ2* prevents $H_2O_2$ induced KIN10 nuclear localization, nuclear KIN10 promoted stomatal opening and guard cell starch degradation. Consistently, *kinβ2* mutant exhibits slightly more starch under the end of night, less starch after 1 hour light exposure, as well as increased stomatal apertures. (Supplementary Fig. 8e, f). These findings confirm the essential role of KIN10 nuclear localization in stomatal opening and starch metabolism.

## KIN10 interacts with and phosphorylates bZIP30 to promote guard cell starch degradation and stomatal opening

Given the kinase activity of KIN10 and its nuclear localization in the context of guard cell starch degradation and stomatal opening, it can be hypothesized that KIN10 functions by modulating the activity of a transcription factor specifically expressed in guard cells. Among the potential interacting proteins of KIN10 that have been identified by yeast two-hybrid (Y2H) screening, with KIN10 as bait[33], bZIP30 transcription factor was selected for further investigation. To verify the interaction between KIN10 and bZIP30, we performed the Y2H assay and found that KIN10 interacted with bZIP30 in yeast (Fig. 3a). The ratiometric bimolecular fluorescence complementation (rBiFC) assays showed that the strong YFP fluorescence signal was observed in both the nucleus and cytosol in tobacco leaf epidermal cell when KIN10-nYFP was co-transformed with bZIP30-cYFP but not with histone H3-cYFP (Fig. 3b). The protein-protein pull-down assays showed that glutathione S-transferase (GST) fusion protein GST-bZIP30 interacted with the maltose-binding protein (MBP) fusion protein MBP-KIN10 but not with MBP alone (Fig. 3c). Co-immunoprecipitation assays performed with *pbZIP30::bZIP30-YFP* transgenic plants showed that bZIP30-YFP interacted with endogenous KIN10 in plants (Fig. 3d). These results suggested that KIN10 interacts with bZIP30 in vivo and in vitro.

To investigate the potential role of bZIP30 in the regulation of stomatal function, we conducted several experiments. First, we cloned the genomic sequence of *bZIP30*, which includes both the encoding and promoter regions, and fused it with a YFP tag. The *pbZIP30::bZIP30-YFP* construct exhibited a universal expression pattern in all epidermal cells, but displayed a high nuclear-to-cytoplasmic ratio specifically in guard cells and stomatal lineage cells. This expression and nuclear localization pattern were similar to those observed for the KIN10 protein (Supplementary Fig. 9a–c). To further determine whether bZIP30 has a similar function to KIN10, we examined the mutant phenotype of *bzip30* in terms of guard cell starch metabolism and stomatal opening in response to light. The results showed that the mutation of bZIP30 represented similar phenotypes with *kin10* mutant and resulted in the stable guard cell starch and impaired stomatal opening upon light exposure (Fig. 3e–g, Supplementary Fig. 10a, b). In addition, *bzip30* mutant exhibited the deficiency of stomatal development and repressed the number of stomata, which is similar to *kin10* mutant (Supplementary Fig. 9d–g). Overexpression of *KIN10* promoted guard cell starch degradation and stomatal opening upon light exposure, but such promoting effects were decreased in *bzip30* mutant, suggesting that bZIP30 is required for KIN10-promoted guard cell starch degradation and stomatal opening (Fig. 3e–g, Supplementary Fig. 10a, b).

Considering the important role of KIN10 in $H_2O_2$-mediated starch degradation and stomatal opening, and bZIP30 is located downstream of KIN10 to regulate stomatal movement, we speculated that bZIP30 may also be involved in $H_2O_2$-mediated starch degradation and stomatal movement. To test this hypothesis, we analyze the guard cell starch content and stomatal apertures in the intact leaves of wild-type plant, *bzip30* and *kin10* mutant with or without exogenous $H_2O_2$ treatment. The results showed that $H_2O_2$ treatment induced larger stomatal apertures and less starch content in guard cells, but these effects were reduced in *kin10* and *bzip30* mutants (Supplementary Fig. 11a, b). Furthermore, $H_2O_2$ treatment induced the expression of *BAM1* in wild-type plants, but had very weak effects on *kin10* and *bzip30* mutants (Supplementary Fig. 11c, d). These results indicated KIN10 and bZIP30 play important roles for $H_2O_2$-promoted light-induced guard cell starch degradation and stomatal opening.

Considering a series of bZIP transcription factors as direct substrates for KIN10, it was speculated that KIN10 could phosphorylate bZIP30. An in vitro kinase assay was performed by application GST-SnAK2 to activate MBP-KIN10. The results showed that MBP-KIN10 phosphorylated MBP-bZIP30 in the presence of GST-SnAK2 (Fig. 3h). Mass spectrometry analysis revealed that KIN10 phosphorylated the Serine-18 and Threonine-22 residues of bZIP30 (Fig. 3i and Supplementary Fig. 12a). To investigate the function of KIN10-mediated phosphorylation on starch degradation in guard cells and stomatal opening, the phosphorylated residues of bZIP30 were mutated to glutamate (S18/T22D) to mimic phosphorylation, or to Alanine (S18/T22A) to prevent phosphorylation. The transformation of the *pbZIP30::bZIP30-YFP* and *pbZIP30::bZIP30^{S18/T22D}-YFP* successfully rescued the defective stomatal opening and guard cell starch degradation phenotypes of *bzip30*. However, the transformation of the *pbZIP30::bZIP30^{S18/T22A}-YFP* construct failed to rescue the stomatal function phenotype of the *bzip30* mutant (Fig. 3j, k and Supplementary Fig. 12b,c). These results proved that the KIN10-dependent phosphorylation of bZIP30 enhances its function of regulating stomatal opening and starch degradation in guard cells.

## bZIP30 directly induces the expression of *BAM1* and *AMY3*

To elucidate the underlying mechanism of the regulation of starch metabolism in guard cells by the KIN10-bZIP30 module, we investigated the expression levels of genes involved in guard cell starch degradation using quantitative RT-PCR. In both *kin10* and *bzip30* mutants, we observed a significant inhibition of *BAM1* and *AMY3* expression (Fig. 4a, b). Chromatin immunoprecipitation quantitative PCR (ChIP-qPCR) assays using *pbZIP30::bZIP30-YFP* plants showed that bZIP30 directly binds to the promoters of *BAM1* and *AMY3* (Fig. 4c,

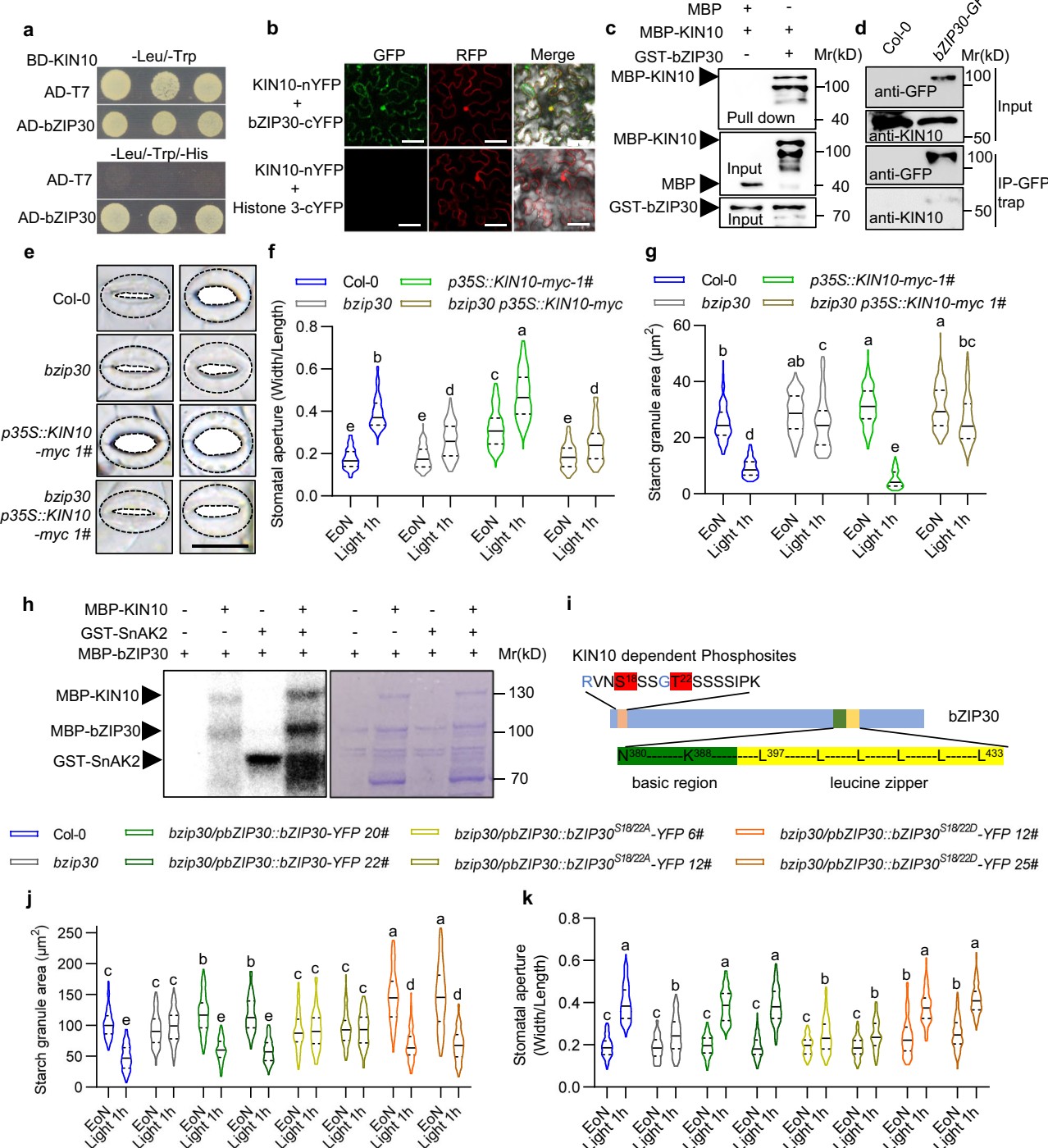

**Fig. 3 | KIN10 interacts with and phosphorylates bZIP30. a–d** Y2H (**a**) ratiometric BiFc (**b**), In vitro pull-down (**c**), and CoIP (**d**) assays show the interaction between KIN10 and bZIP30. BD-KIN10 was used as the bait for Y2H. Co-transforming BD-KIN10 with an empty-AD vector serves as the negative control. Scale bars in confocal images of ratiometric BiFc represent 20 μm. In pull-down assay, recombinant MBP and MBP-KIN10 proteins were incubated with GST-bZIP30 bound to glutathione agarose beads. Immunoblot assay was performed with anti-MBP antibody to detect the proteins bound to GST-bZIP30. Protein extract from 10-day-old seedlings of *pbZIP30::bZIP30-YFP* and Col-0 plants were used for CoIP assay. bZIP30-YFP was precipitated using GFP-Trap agarose beads. The immunoblots were probed with anti-GFP and anti-KIN10 antibodies. **e–g** Quantification of stomatal apertures (**e, f**) and guard cell starch granules (**g**) in the rosette leaves of Col-0, *bzip30*, *p35S::KIN10-myc* and *bzip30 p35S::KIN10-myc* plants. Seedlings of these plants were grown on 1/2 MS medium under a 12 h light/12 h dark photoperiod with a 100 μM m⁻² s⁻¹ light intensity for 28 days. **h** KIN10 phosphorylates bZIP30 in vitro.

Left: gel image with ATP-γ-p32-labeled proteins. Right: gel image stained with coomassie brilliant blue. **i** Schematic representation of the bZIP30 protein indicating the phosphorylation sites by KIN10. The residues marked in red color indicates the residues phosphorylated by KIN10. **j, k** Quantification of guard cell starch granules (**j**) and stomatal apertures (**k**) in the cotyledons of Col-0, *bzip30*, and different bZIP30-complemented transgenic plants. Seedlings of these plants were grown under the same conditions for 10 days. The starch granules area and the ratio of stomatal aperture width to length from more than 100 guard cells in at least 12 different plants were measured using ImageJ software. EoN means the end of night, and Light 1 h means the white light illumination for 1 hour after the end of night. The solid lines of violin plots represent the median; the dashed lines represent the first or third quartile. Different letters above the bars indicate statistically significant differences between samples (Two-way ANOVA analysis followed by Tukey's multiple comparisons test, $p < 0.05$).

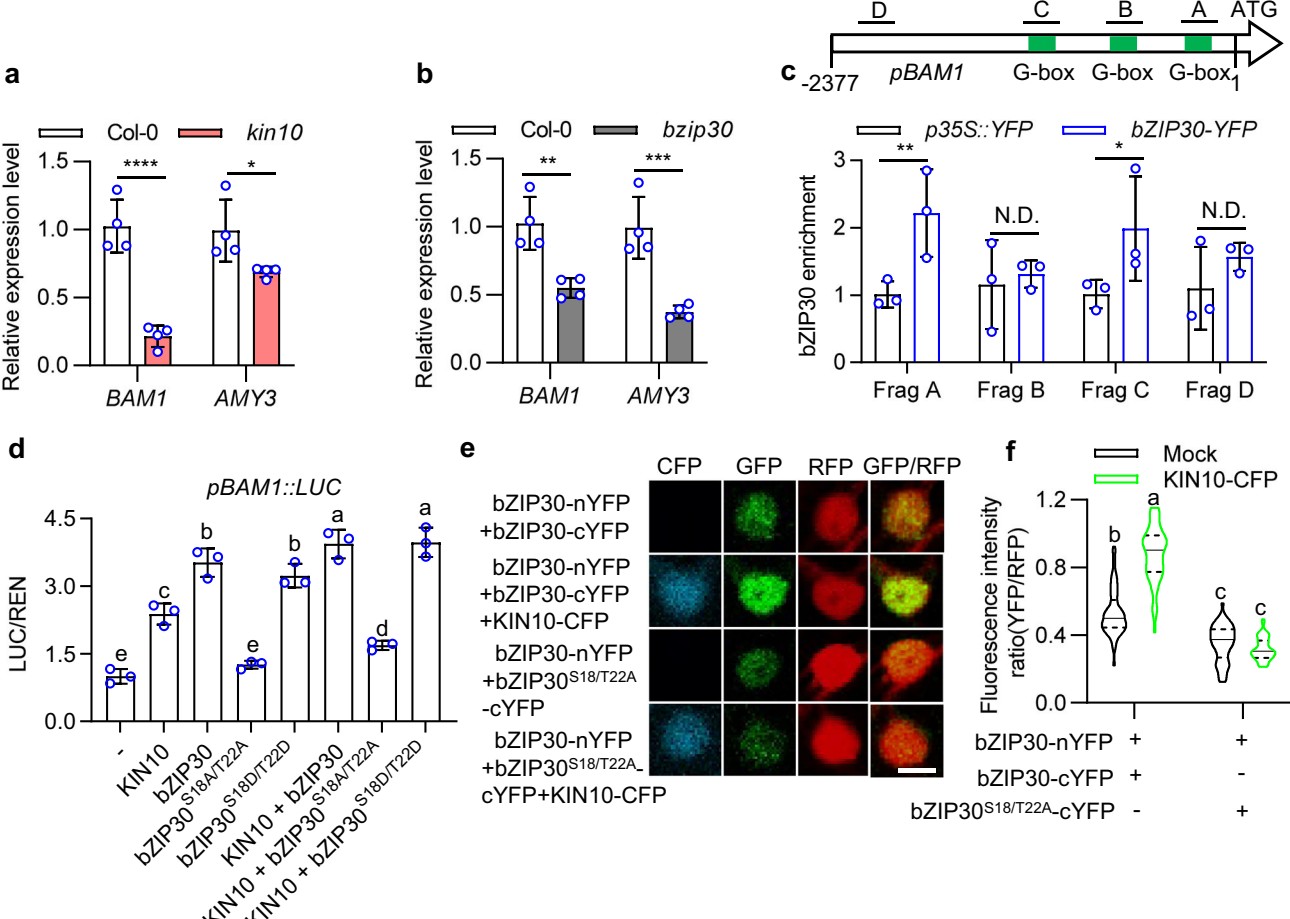

**Fig. 4 | bZIP30 induces *BAM1* and *AMY3* expression. a, b** qRT-PCR analysis of the expression of *BAM1* and *AMY3* in Col-0, *kin10* and *bzip30* plants. Plants were grown on 1/2 MS medium under a 12 h light/12 h dark photoperiod with a 100 μM m⁻² s⁻¹ light intensity for 28 days. Error bars indicate standard deviation (S.D.). Asterisks between bars indicate statistically significant difference between samples (Two-way ANOVA analysis followed by uncorrected Fisher's LSD multiple comparisons test, *$p < 0.05$, **$p < 0.01$, ***$p < 0.001$, and ****$p < 0.0001$). **c** Quantitative ChIP-PCR showed the direct binding of bZIP30 to *BAM1* promoter. Seedlings of *p35S::YFP* and *pbZIP30::bZIP30-YFP* were used to performed ChIP assays. The levels of bZIP30 binding were calculated as the ratio between *pbZIP30::bZIP30-YFP* and *p35S::YFP*, and then normalized to that of control gene *PP2A*. Error bars indicate S.D. Asterisks between bars indicate statistically significant differences between samples (Two-way ANOVA analysis followed by uncorrected Fisher's LSD multiple comparisons test, $p < 0.05$). **d** Transient assays showed that the expression of *BAM1* was induced by KIN10-YFP, bZIP30-YFP and bZIP30^S18/T22D-YFP, but not by bZIP30^S18/T22A-YFP. The promoter of *BAM1* and fused to the luciferase reporter gene were co-transfected

with KIN10-YFP, bZIP30-YFP, bZIP30^S18/T22A-YFP or bZIP30^S18/T22D-YFP into Arabidopsis mesophyll protoplasts. The luciferase activities were normalized with *Renilla* luciferase. Error bars represent S.D. Different letters above the bars indicate statistically significant differences between samples (One-way ANOVA analysis followed by uncorrected Fisher's LSD multiple comparisons test, $p < 0.05$). **e, f** rBiFC confocal images showed that KIN10-dependent phosphorylation promotes the homodimerization of bZIP30 but not the interaction of bZIP30 with phospho-mutated bZIP30 in plants. The construct of *p35S::bZIP30-nYFP-p35S::RFP-p35S::bZIP30-cYFP* or *p35S::bZIP30-nYFP-p35S::RFP-p35S:: bZIP30^S18/T22A-cYFP* is co-transferred with or without *p35S::KIN10-CFP* in tobacco leaves. The fluorescent signals of YFP (BiFC) and RFP (reference) in the nucleus of tobacco epidermal cells were determined using ImageJ software. At least 50 interaction signals in different epidermal cells were analyzed. The solid lines of violin plots represent median, the dashed lines represent first or third quartile. Different letters above the bars indicated statistically significant differences between the samples (Two-way ANOVA analysis followed by Tukey's multiple comparisons test, $p < 0.05$. Scale bar, 10 μm).

Supplementary Fig. 13a). Transient expression assays showed that the luciferase (LUC) reporter driven by the *BAM1* or *AMY3* promoter exhibited increased activity when KIN10, bZIP30 and bZIP30^S18/T22D were transfected individually, and this activity was further enhanced when KIN10 and bZIP30 or bZIP30^S18/T22D were co-expressed. However, transfection of the bZIP30^S18/T22A failed to activate the promoter activities of *BAM1* and *AMY3*. Additionally, it suppressed the inductive effect of KIN10 transfection (Fig. 4c, Supplementary Fig. 13b). These results revealed that KIN10 enhances bZIP30 activity to induce the expression of *BAM1* and *AMY3*.

To investigate the impact of KIN10-mediated phosphorylation on the activity of bZIP30, we initially analyzed the influence of KIN10 on the subcellular localization of bZIP30. When bZIP30 was transiently expressed in tobacco epidermal cells, it was observed to be localized in both the cytoplasm and nucleus. The co-expression of KIN10 did not

have any significant impact on the subcellular localization of bZIP30. Additionally, bZIP30^S18/T22A-GFP and bZIP30^S18/T22D-GFP exhibited similar subcellular localization with bZIP30-GFP in tobacco leaves (Supplementary Fig. 14a, b). Likewise, in Arabidopsis epidermis, bZIP30^S18/T22A-YFP and bZIP30^S18/T22D-YFP exhibited a comparable nuclear-to-cytoplasmic ratio to that of bZIP30-YFP in different epidermal cell of Arabidopsis leaves (Supplementary Fig. 14c, d). These results indicated that KIN10-mediated phosphorylation had no effect on the subcellular localization of bZIP30. The bZIP transcription factors generally perform dimerization to regulate downstream gene expression[34]. To determine whether KIN10 regulates the dimerization of bZIP30, we performed an rBiFC assay in tobacco leaves to analyze the interaction between bZIP30 proteins. The results showed that bZIP30 interacts with itself and forms a homodimer in the nucleus of tobacco epidermal cells. However, it was observed that the interaction between bZIP30

and non-phosphorylatable variant bZIP30[S18/T22A] was significantly weakened than the interaction of wild-type bZIP30 dimer. Notably, the co-expressing KIN10 enhanced the interaction of bZIP30 dimer, but not bZIP30/ bZIP30[S18/T22A] dimer (Fig. 4e, f). These results suggested that KIN10-mediated phosphorylation promotes bZIP30 homodimerization to increase its transcriptional activity.

## The bZIP30-BZR1 transcriptional complex integrates BR and $H_2O_2$ signals to induce guard cell starch degradation and stomatal opening

$H_2O_2$ has been reported to act interdependently with BR for inducing guard cell starch degradation and stomatal opening[15]. Our results showed that the KIN10-bZIP30 module is required for $H_2O_2$-induced guard cell starch degradation and stomatal opening. Therefore, we propose that KIN10 and bZIP30 may play a role in BR-mediated regulation of guard cell starch degradation and stomatal opening. To test this hypothesis, we analyzed the effects of brassinolide (BL), the most active BR, on the guard cell starch content and stomatal apertures in wild-type plants, *kin10* and *bzip30* mutants. The results showed that BL treatment results in less starch content in guard cells and larger stomatal apertures upon light exposure, but had weak effects in both *kin10* and *bzip30* mutants (Fig. 5a, b). Additionally, BL treatment induced the expression of *BAM1* in wild-type plants, but such promoting effects were reduced in both *kin10* and *bzip30* mutants (Supplementary Fig. 15a, b). These results suggested that KIN10-bZIP30 module is required for BR-induced guard cell starch degradation and stomatal opening.

$H_2O_2$ induces the oxidation of BZR1, which is a key transcription factor in the BR signaling pathway, leading to enhanced interaction with the bZIP transcription factor G-box binding factor 2 (GBF2) and increased expression of *BAM1*[15]. Therefore, we speculate that $H_2O_2$ might induce the interaction between BZR1 and bZIP30. The results showed that bZIP30 interacted with BZR1 in the nucleus of epidermal cells in tobacco leaves. The interaction between bZIP30 and BZR1[C63S], which attenuates the most crucial oxidation residues of BZR1[35], is repressed compared with the interaction of wild-type BZR1-bZIP30 heterodimer. $H_2O_2$ treatment enhanced the interaction between bZIP30 and BZR1, but not the interaction between bZIP30 and BZR1[C63S] (Fig. 5c, d). The recombinant protein pull-down assay revealed that GST-bZIP30 interacts with MBP-BZR1 but not MBP alone. $H_2O_2$ treatment remarkably enhances the interaction between GST-bZIP30 and MBP-BZR1, but not the interaction between GST-bZIP30 and MBP-BZR1[C63S] (Fig. 5e, f). On the other side, the interaction of BZR1 and bZIP30[S18/T22A] is repressed compared to the interaction of wild-type BZR1-bZIP30 heterodimer and did not respond to $H_2O_2$ treatment (Fig. 5g, h). In addition, co-transformation with *p35S::KIN10-CFP* construct also enhances the interaction of BZR1 and bZIP30 but not the interaction of BZR1 and bZIP30[S18/T22A] in tobacco leaves (Supplementary Fig. 16a, b). These results suggested that $H_2O_2$-induced oxidation of BZR1 and KIN10-dependent phosphorylation of bZIP30 are required for BZR1-bZIP30 heterodimer formation, which could help induce *BAM1* and *AMY3* expression and guard cell starch degradation downstream of BR and $H_2O_2$ signaling.

## $H_2O_2$ is required for stomatal opening across different plant species

The specific accumulated $H_2O_2$ in guard cells under normal growth conditions is required for light-induced stomatal opening in Arabidopsis, we wondered whether this phenomenon exists widely in other plant species. To test this hypothesis, we examined the $H_2O_2$ content and stomatal apertures in different plant species, including the lycophyte *Selaginella doederleinii*, the fern *Marsilea quadrifolia*, and the angiosperms *Triticum aestivum* and *Arabidopsis thaliana*. The results showed a significant accumulation of $H_2O_2$ in guard cells of all tested plant species under normal growth conditions (Fig. 6a, Supplementary

Fig. 17a–d). Removing $H_2O_2$ by treating the plants with KI or DDC led to impaired stomatal opening upon light exposure in *Triticum aestivum* and *Marsilea quadrifolia* plants (Fig. 6b–e). These results suggested that the accumulation of $H_2O_2$ in guard cells, even when plants grown under normal growth conditions, is necessary for promoting stomatal opening in response to light in various plant species.

To further confirm $H_2O_2$ exhibits conserved function in different plant species, we cloned the homolog genes of BZR1, bZIP30, and KIN10 from *Selaginella moellendorffii*. The rBiFC assays showed that SmbZIP30 interacts with SmBZR1 in tobacco leaves, and this interaction enhanced by $H_2O_2$ treatment (Supplementary Fig. 18a, b). $H_2O_2$ treatment induces SmKIN10 protein nuclear localization, which is consistent with KIN10 protein from Arabidopsis (Supplementary Fig. 18c–f). Protein alignment analysis revealed that KIN10-dependent phosphorylation sites of bZIP30, oxidative modification site of BZR1, and G-box binding sequences from *BAM1* promoter are widely present from bryophyte to angiosperms (Supplementary Fig. 19–21). These findings suggested guard cell specific accumulated $H_2O_2$ in different plant species seems to apply a common mechanism to promote light-induced stomatal opening.

## Discussion

Among ROS, $H_2O_2$ in particularly plays crucial roles in various plant growth and development processes, as well as in stress responses. This study further demonstrates the essential role of $H_2O_2$ in light-induced stomatal opening. Surprisingly, $H_2O_2$ accumulates within guard cells even under normal growth conditions. Removal of $H_2O_2$ by chemical treatments or genetic manipulations impaired stomatal opening in response to light. Within guard cells, $H_2O_2$ serves a dual function. First, it facilitates the translocation of KIN10 into the nucleus, leading to the phosphorylation of bZIP30. Second, $H_2O_2$ can oxidize and modify BZR1. The phosphorylated bZIP30 by KIN10 and the oxidation-modified BZR1 by $H_2O_2$ are more likely to form a heterodimer, which in turn induces the expression of *BAM1* and *AMY3*. This ultimately promotes starch degradation and stomatal opening. Furthermore, the accumulation of $H_2O_2$ in guard cells to promote stomatal opening is widespread across different species. Overall, these findings uncovered the specific role of $H_2O_2$ in promoting stomatal opening through the activation of the KIN10-bZIP30-BZR1 module (Fig. 6f).

$H_2O_2$ has been known to act as a molecular signal promoting stomatal closure[36]. When plants grow under stress conditions, $H_2O_2$ accumulates and activates S-type anion channel through $Ca^{2+}$ currents and receptor-like kinase Guard cell hydrogen peroxide-resistant 1 (GHR1), resulting in stomatal closure[37,38]. However, in this study, we showed that $H_2O_2$ also accumulated in guard cells even when plants grow under normal growth conditions, and this specific accumulation of $H_2O_2$ is essential for light-induced stomatal opening. Removal of $H_2O_2$ through KI or DDC treatment, overexpression of *CAT2*, or loss-of-function of *RBOHD* and *RBOHF* genes all result in stable guard cell starch accumulation and impaired stomatal opening when exposed to light. Exogenous $H_2O_2$ treatment showed that it could promote stomatal opening at the concentrations lower than 30 μM, but not at concentrations higher than 50 μM. Furthermore, higher concentrations of $H_2O_2$, from 50 μM to 1000 μM $H_2O_2$, inhibit the light-induced stomatal opening. Consistent with this, $H_2O_2$ activates BZR1 and KIN10 at low concentrations. In the normal growth state, the accumulation of $H_2O_2$ induces the oxidative modification of BZR1, promoting the formation of a complex between BZR1, ARF6, and PIF4, which promotes elongation growth of plants[35,39]. In stomata, $H_2O_2$-induced oxidative modification of BZR1 facilitates the formation of a complex between BZR1 and GBF2, thereby promoting stomatal opening[15]. $H_2O_2$ can induce the translocation of KIN10 from the cytoplasm to the nucleus at all concentrations. However, high levels of $H_2O_2$ have been reported to inhibit KIN10 kinase activity[40]. Taken together, these results indicate that $H_2O_2$ regulates stomatal movement in a concentration-dependent

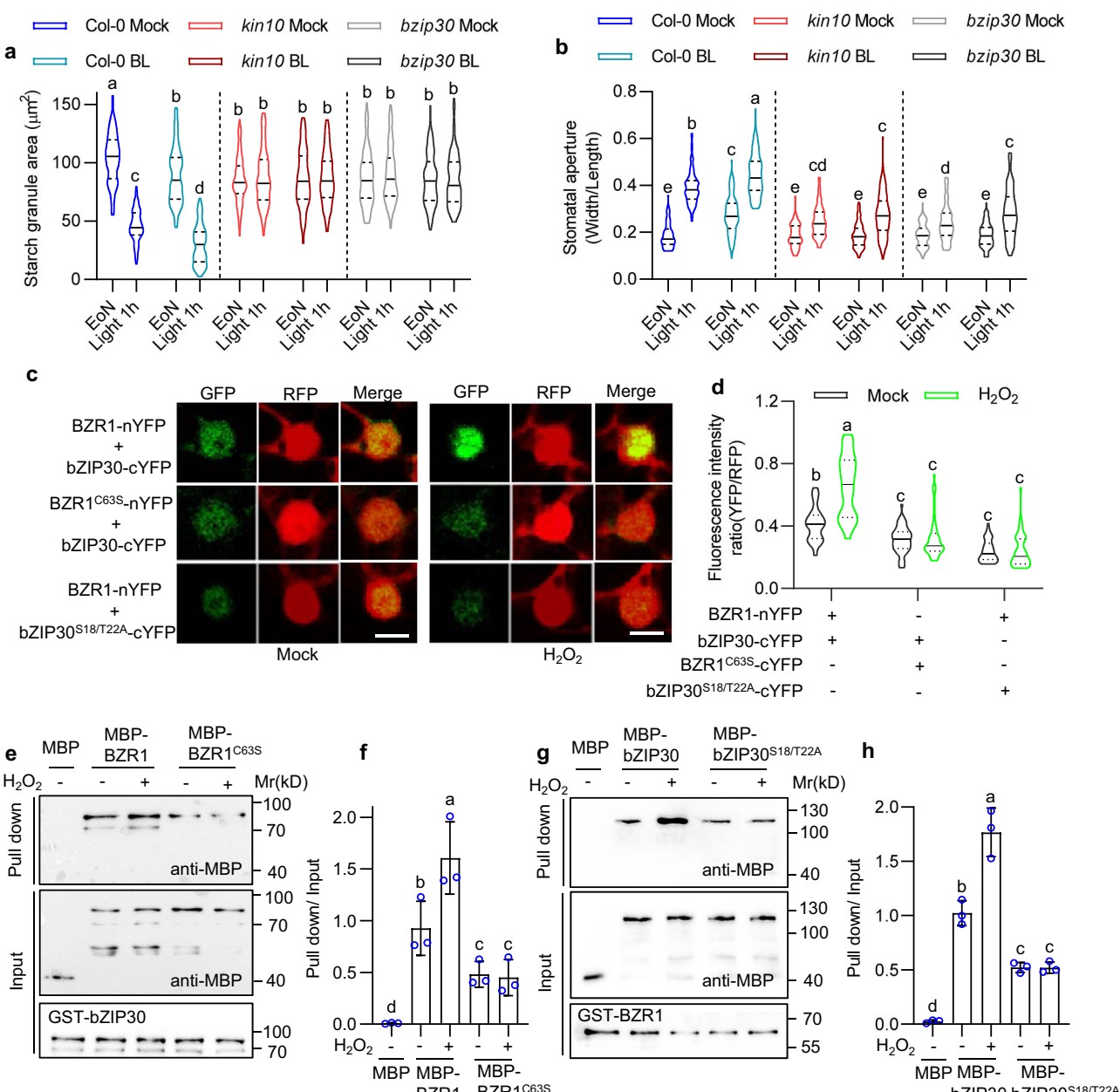

**Fig. 5 | The KIN10-bZIP30 module integrates BR and H₂O₂ signals to promote light-induced stomatal opening. a, b** Quantification of guard cell starch granules (**a**) and stomatal apertures (**b**) in Col-0, *kin10,* and *bzip30* plants with or without BL treatment. Seedlings were grown on 1/2 MS medium under a 12 h light/12 h dark photoperiod with a 100 μM m⁻² s⁻¹ light intensity for 10 days, transferred to the medium with or without 100 nM BL to grow for 2 h before EoN. The starch granules area and stomatal aperture from more than 100 guard cells in at least 12 different plants were measured. EoN means the end of night, and Light 1 h means the white light illumination for 1 hour after EoN. **c, d** rBiFC confocal images showed that H₂O₂ enhanced the interaction between BZR1 and bZIP30, which is required for both BZR1 oxidation and bZIP30 phosphorylation. The construct of *p35S::BZR1-nYFP-p35S::RFP-p35S::bZIP30-cYFP, p35S::BZR1^C63S^-nYFP-p35S::RFP-p35S::bZIP30-cYFP,* and *p35S::BZR1-nYFP-p35S::RFP-p35S::bZIP30^S18/T22A^-cYFP* is transferred in tobacco leaves, and treated with or without 1 mM H₂O₂ for 3 h before observation. The ratio of fluorescent signals of YFP (BiFC) and RFP (reference) was calculated. At least 50

interaction signals in different epidermal cells were analyzed. Scale bars represents 10 μm. **e–h** In vitro pull-down assays showed that H₂O₂ enhances the interaction between BZR1 and bZIP30. Recombinant MBP, MBP-BZR1 and MBP-BZR1^C63S^ proteins were incubated with GST-bZIP30 bound to glutathione agarose beads (**e, f**). Recombinant MBP, MBP-bZIP30, and MBP-bZIP30^S18/T22A^ proteins were incubated with GST-BZR1 bound to glutathione agarose beads (**g, h**). All these incubations were performed with or without 1 mM H₂O₂ for 3 hours. Immunoblot assay was performed with anti-MBP antibody to detect the proteins bound to GST-bZIP30 or GST-BZR1. Error bars represent the standard deviation. The solid lines of violin plots represent median, the dashed lines represent the first or third quartile. Different letters above the bars indicate statistically significant differences between samples. Two-way ANOVA analysis followed by Tukey's multiple comparisons test ($p < 0.05$) was applied for **a, b,** and **d**. One-way ANOVA analysis followed by Uncorrected Fisher's LSD multiple comparisons test ($p < 0.05$) was applied for **f** and **h**.

manner, promoting stomatal opening at low concentrations and stomatal closure at high concentrations.

H₂O₂ is distributed differently among different tissues and different cell types, which determines the specific function of tissues or

cells. Our previous study revealed that H₂O₂ is specifically concentrated in the meristemoids and guard cells of intact leaves of plants grown under normal growth conditions[25]. In the meristemoids, H₂O₂ triggers the nuclear localization of KIN10[25]. The nuclear localized

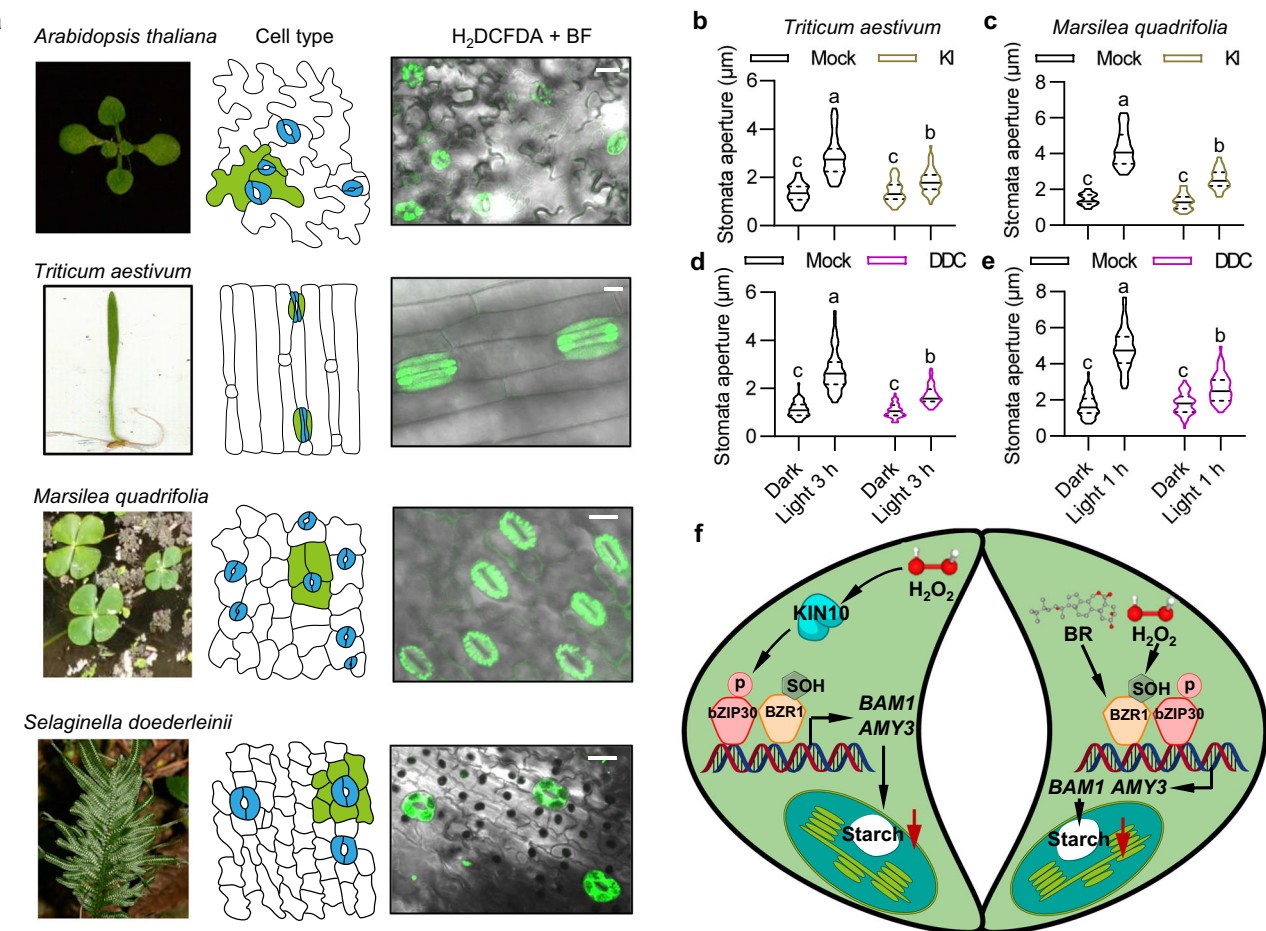

**Fig. 6 | H₂O₂ is required for light-induced stomatal opening across different plant species. a** Staining of H₂O₂ in the epidermal cells of *Arabidopsis thaliana* (dicotyledonous), *Triticum aestivum* (monocotyledonous), *Marsilea quadrifolia* (ferns) and *Selaginella doederleinii* (lycophytes) using H₂DCFDA. The stomata and surrounded subsidiary cells in abaxial epidermis of leaves in different plant species are highlighted by blue and green color, respectively. Scale bars in confocal images for H₂O₂ staining represent 20 μm. **b−e** Quantification of stomatal apertures in the leaves of *Triticum aestivum* (**b**, **d**) and *Marsilea quadrifolia* (**c**, **e**) with or without KI or DDC treatment. The leaves of *Triticum aestivum* and *Marsilea quadrifolia* were pre-incubated in the presence or absence of 1 mM KI or 1 mM DDC in the dark for 2 h before illumination with white light (100 μmol m⁻² s⁻¹) and for 3 h and 1 h, respectively. The width of stomata of at least 50 guard cells from 8 different plants were measured using ImageJ software. The solid lines of violin plots represent median, the dashed lines represent first or third quartile. Different letters above the bars indicate statistically significant differences between samples (Two-way ANOVA analysis followed by Tukey's multiple comparisons test, $p < 0.05$). **f** A proposed working model for the promoting effects of H₂O₂ for light-induced stomatal opening. H₂O₂ enrichment in guard cells of plant grown under normal growth conditions promotes the localization of KIN10 in nucleus, in which it interacts with and phosphorylates bZIP30 transcription factor. BR induces the dephosphorylation and nuclear localization of BZR1. Guard cell enriched H₂O₂ induces the oxidation of BZR1 at cysteine 63. The phosphorylated bZIP30 by KIN10 and oxidation-modified BZR1 by H₂O₂ are more likely to form a heterodimer to induce *BAM1* and *AMY3* expression, thereby promoting guard cell starch degradation and stomatal opening.

KIN10 phosphorylates and stabilizes SPCH, which promotes stomatal development[24]. SPCH, in turn, suppresses the expression of genes responsible for scavenging ROS, such as *CAT2* and *APX1*, leading to the accumulation of H₂O₂ in the meristemoids and establishing a positive feedback loop[25]. Here we showed that specific accumulated H₂O₂ in guard cell is required for stomatal opening and guard cell starch degradation. The expression patterns of *CAT2* and *APX1* in Arabidopsis epidermal cells exhibits the lower level in guard cells[25], suggesting a possible reason for the specific accumulation of H₂O₂ in guard cells. The identification of the specific transcription factors responsible for inhibiting the low expression of *CAT2* and *APX1* in guard cells remains to be explored. Another possibility for the accumulation of H₂O₂ in guard cell may be related to their specific metabolic characteristics. Guard cells require a high demand for energy in the form of ATP to facilitate the stomatal movement, which is due to the large number of ion channels and transporters involved in these processes[14,41]. ROS, including H₂O₂, are generated by the NADH dehydrogenase of the respiratory chain Complex I. Guard cells have a higher number of mitochondria and greater oxidative phosphorylation ability compared to mesophyll cells[42,43]. This may explain why H₂O₂ specifically accumulates in guard cells. However, further research is needed to understand how guard cells accumulate H₂O₂ under normal growth conditions.

Starch degradation in guard cells is required for H₂O₂ at low concentration promoted stomatal opening upon light exposure. In this study, we have showed that removal H₂O₂ content through applying DDC or KI inhibit starch degradation in guard cell. Exogenous treatment of H₂O₂ with low concentration promotes starch degradation and *BAM1* expression, which depends on the function of KIN10 and bZIP30. Our previous study also showed that the mutations of *BAM1* and *AMY3* also repressed H₂O₂ induced stomatal opening[15]. All these data suggested that guard cell starch degradation is one of the reasons for H₂O₂ promoted stomatal opening. And it is also worth to test whether H₂O₂ with low concentration regulates the activity of ion channels, including AHA or S-Type anion channels.

$H_2O_2$ promoted starch degradation in guard cells is influenced by stage of leaf development. Firstly, the starch amount in guard cells in quite different between cotyledon of seedlings and mature leaves. Starch synthesis of guard cells relies on the sugar transport from mesophyll cells[13]. For seedlings, there is only two apical meristems (shoot and root) for consuming carbon source. However, mature plants contain much more sink tissues, including much more amounts of stomata, to consuming nutrient. Therefore, a single guard cell in mature plants possess less carbon source for starch synthesis. Notably, our data has shown that plants with different genetic construct exhibit similar starch dynamic after light exposure between seedlings and mature leaves. Secondly, the starch degradation rate and starch content during EoN in the guard cells of *cat2* mutant exhibited differences between cotyledon and rosette leaves. We detected the accumulation of $H_2O_2$ in the guard cells of *cat2* mutant compared to that of wild type plants, which may weaken the ability of starch synthesis. The demand of starch synthesis in cotyledon is much higher than that of rosette leaves. Therefore, *cat2* mutant with weak ability of starch synthesis only exhibits the difference with wild type plant in cotyledon, which needs to accumulate more starch owing to more carbon source supply.

$H_2O_2$ is the important redox signaling molecule that reflects oxygen gradients and influences the development of multicellular organisms throughout evolution[26]. In this study, we showed that the requirement of $H_2O_2$ for light-induced stomatal opening has been proved to exist in angiosperms including Arabidopsis and wheat, as well as in basal vascular plants such as *Selaginella doederleinii* (lycophytes) and *Marsilea quadrifolia* (ferns). Treatment with ROS scavenger like KI or DDC could reduce $H_2O_2$ content in guard cells and inhibit light-induced stomatal opening in Arabidopsis, wheat and *Marsilea quadrifolia*. These results are consistent with previous studies that blue light can induce stomatal opening and conductance in major lineage of both eusporangiate and leptosporangiate ferns together with a lycophyte (*Selaginella kraussiana*)[44]. Blue light can also stimulate ROS production and gene expression related to starch metabolism, thereby enhancing the speed of stomatal opening in leptosporangiate fern *Nephrolepis exaltata*[44]. Over all, these data suggested that specific accumulation of $H_2O_2$ in guard cell exists among vascular plants widely and is essential for light induced stomatal opening.

Approximately 450 million years ago, the earliest land plants began to colonize terrestrial environments, which exposure them to various stress conditions, such as the atmosphere and radiation[1]. In order to survive in these new environments, plants underwent crucial adaptations, one of which was the evolution of stomata. Fossil records from Devonian period provide clear evidence of stomata in early land plant, indicating their origin in plants[45]. Both bryophytes and vascular plants, which are the two distinct lineages of living plants, use the conserved SPCH-like bHLH transcription factors to regulate stomata formation[46]. In this study, the specific accumulation of $H_2O_2$ in guard cells is observed in basal lineage of vascular plants, such as *Selaginella doederleinii*, and *Marsilea quadrifolia*. Removal of $H_2O_2$ by KI or DDC treatment inhibits the stomatal opening after light exposure in *Marsilea quadrifolia*. We also showed that $H_2O_2$ facilitates the nuclear localization of KIN10 and the formation of the transcriptional heterodimer between BZR1 and bZIP30, which induces the expression of *BAM1* and *AMY3*. Moreover, $H_2O_2$ also promotes nuclear localization of KIN10 homologs from *Selaginella moellendorffii*, and enhanced the interaction of BZR1 and bZIP30 homolog from *Selaginella moellendorffii*. The conserved function of BZR1 and KIN10 in plant development and metabolism suggest that these signaling pathway have been present since the evolution of land plants, from moss to angiosperms[47,48]. Overall, these findings indicate that land plants have developed $H_2O_2$ signaling in guard cells and the KIN10-BZR1-bZIP30 module to facilitate rapid stomata movement in response to light.

## Methods

### Plant materials and growth conditions

The Arabidopsis wild-type seedling and all transformation background used in this study were ecotype Columbia (Col-0). In this study, the mutants and transgenic plants used including *cat2*[35], *rbohD rbohF*[35], *p3SS::CAT2-myc*[25], *kin10*[24], *p3SS::KIN10-myc*[24], *pKIN10::KIN10-YFP*[24], *pBAM1::GFP*[15] and *p3SS::KINβ2-YFP*[25] have been described previously. The T-DNA insertion mutants of *bzip30* (SALK_076998), *kinβ2-1* (SALK_037416C) and *kinβ2-2* (SALK_052521) were ordered from the Arabidopsis Biological Resource Center. The transgenic plants expressing similar levels of *pKIN10::KIN10-YFP/kin10*, *pKIN10::NES-KIN10-YFP/kin10*, *pKIN10::NLS-KIN10-YFP/kin10*, *pbZIP30::bZIP30-YFP/ bzip30*, *pbZIP30::bZIP30^{S18AT22A}-YFP/bzip30* and *pbZIP30::bZIP30^{S18DT22D}-YFP/bzip30* were selected for further phenotypic analysis. Plants were grown in a greenhouse with white light at 100 µmol m$^{-2}$ s$^{-1}$ under a 12 h light/12 h dark photoperiod at 22 °C for general growth. For chemical treatment, seedlings of the indicated plants treated with or without $H_2O_2$, BL, DDC, or KI for different times.

### Plasmid constructs and transgenic plants

Full-length coding regions of bZIP30, KIN10, NLS-KIN10, NES-KIN10, KINβ2 and BZR1 without stop codon were amplified by PCR and cloned into pENTR™/SD/D-TOPO™ vectors (Thermo Fisher). And then all these entry clones recombined with destination vector Px-YFP (p35S::X-YFP), p1390-MH (p35S:X-Myc-His), pDEST15 (N-GST), pMAL2CGW (N-MBP), pGAL4ADGW (GAL4AD-X), pGAL4BDGW (GAL4BD-X). The promoter and genomic DNA of bZIP30 were amplified by PCR and cloned into pENTR™/SD/D-TOPO™ vector (Thermo Fisher), and then recombined with destination vector pEG-TW1 (Native promoter::YFP) to generate *pbZIP30::bZIP30-YFP*. The constructs of bZIP30^{S18A/T22A} and bZIP30^{S18/T22D} were performed using fast mutagenesis system kit (TransGen). Oligo primers used in this study are listed in Supplementary Data 1. All binary vector constructs were introduced into *Agrobacterium tumefaciens* (strain GV3101) and transformed into Col-0, *bzip30* or *kin10* plants by the floral dipping method.

### Fluorescence Imaging and quantification of $H_2O_2$

The H2DCFDA staining assay was performed slightly modified as described previously[25]. Briefly, seedlings were incubated in 10 µM H2DCFDA in 10 mM Tris-HCl (pH 7.2) for 10 min under dark condition. For BES-$H_2O_2$-Ac staining, seedlings were infiltrated with BES-$H_2O_2$-Ac solution (50 µM BES-$H_2O_2$-Ac in 1/2 MS liquid medium) and incubated at room temperature for 1 hour under dark condition. Excess H2DCFDA and BES-$H_2O_2$-Ac were removed by washing five times with 10 mM Tris-HCl (pH 7.2) and 1/2 MS liquid medium respectively. H2DCFDA and BES-$H_2O_2$-Ac staining images with Propidium Iodide (PI) marked cell outlines were captured on LSM880 laser scanning confocal microscope (Zeiss, Oberkochen, Germany). HyPer is a genetically encoded YFP-based $H_2O_2$ sensor that is specific sensitive to $H_2O_2$ but not other form of ROS species. We amplified the HyPer from the pHyper-cyto plasmid and prepared HyPer transgenic plant we used. Fluorescence intensity of pHyper transgenic plants was quantified after excitation at 488 nm or 405 nm and Emission was collected at 530 nm. A fluorescence ratio was calculated as 488/405 using ImageJ software. The average gray values based on fluorescent signals of H2DCFDA staining, BES-$H_2O_2$-Ac staining or pHyper from the whole stomata or the pavement cells were selected and quantified using ImageJ software.

**Guard cell starch quantification.** The quantification of starch in guard cells followed a previously described method[15,17]. Briefly, epidermal peels of cotyledon of wild-type plants grown in different conditions were harvested at the indicated time points and immediately soaked in fixation solution (50% methanol and 10% acetic acid) for at least 24 h, at 4 °C. Samples were first de-stained by soaking in 75% ethanol 25% acetic acid solution at room temperature for 2 hours. After removing

the ethanol solution, the cotyledon was washed with water. Starch granules were stained with a modified pseudo-Schiff propidium iodide (PI) staining method[49]. The stained cotyledon was fixed in Hoyers solution (30 g gum Arabic, 200 g chloral hydrate, 20 g glycerol and 50 mL water) after incubation in chloral hydrate solution (4 g chloral hydrate, 1 mL glycerol and 2 mL water). The samples were observed using an LSM700 laser scanning confocal microscope system (Zeiss, Oberkochen, Germany). The excitation wavelength was set at 555 nm and the emission was collected between 580 nm and 660 nm. The data of starch granule areas was showed in violin plot, among which the solid lines represent median, the dashed lines represent first or third quartile.

**Stomatal apertures measurement.** Stomatal apertures were measured as described in our previous study[15]. The cotyledons of both the wild-type and different mutants were collected at the specified time points and divided between two sections of transparent Scotch tape. One piece of tape was attached to the lower (abaxial) side of the leaves, while the other piece was attached to the upper (adaxial) side. The lower side, containing pavement and guard cells, was carefully separated and placed onto a glass microscope slide. Images of multiple stomata were captured using a ×20 magnification. At least 100 guard cells from more than 4 (rosette leaves) or 12 (cotyledon) different plants were measured using ImageJ software.

## Pull-down assays
GST-bZIP30, GST-BZR1, MBP-KIN10, MBP-bZIP30, MBP-bZIP30$^{S18/T22A}$, MBP-BZR1 and MBP-BZR1$^{C63S}$ were purified from *Escherichia coli* bacteria using glutathione beads (GE Healthcare) or amylose resin (NEB), respectively. Glutathione beads containing 1 μg of GST fused protein were incubated with 1 μg MBP or MBP fused protein in pull-down buffer (20 mM Tris-HCl, pH 7.5; 100 mM NaCl; 1 mM EDTA), with or without 1 mM H$_2$O$_2$ at room temperature for 1 hour and the beads were washed 4 times with wash buffer (20 mM Tris-HCl, pH 7.5; 300 mM NaCl; 1 mM EDTA; 0.5% TritonX-100). The proteins were eluted from beads by boiling in 50 μl 2 × SDS sample buffer and separated on 8% SDS-PAGE gels. Gel blots were analyzed by immunoblot analysis with an anti-MBP antibody (TransGen Biotech, Cat: HT701-01,1:5000 dilution) and anti-GST antibody (TransGen Biotech, Cat: HT601-01,1:5000 dilution).

## Co-immunoprecipitation assays
Arabidopsis seedlings expressing *pbZIP30::bZIP30-YFP* and Col-0 were grown on 1/2 MS solid medium for 10 days. Plant materials were harvested and ground in liquid nitrogen and then extracted in lysis buffer containing 20 mM HEPES-KOH, pH 7.5; 40 mM KCl; 1 mM EDTA; 0.5% Triton X-100; 1 mM PMSF and 1× protease inhibitor cocktail (Sigma Aldrich). After centrifugation at 4 °C, 12,000 × g for 10 min, the supernatant was incubated with GFP-Trap agarose beads (Chromotek) at 4 °C for 1 hour, and the beads were washed 4 times using wash buffer (20 mM HEPES-KOH, pH 7.5; 40 mM KCl; 1 mM EDTA; 300 mM NaCl; and 0.1% Triton X-100). The proteins were eluted from the beads by boiling with 2 × SDS sample buffer, analyzed by SDS-PAGE, transferred to nitrocellulose membrane and immunoblotted with anti-GFP (TransGen Biotech, Cat: N20610,1:5,000 dilution) and anti-KIN10 (Agrisera, Cat: AS10-919, 1:500 dilution) antibodies.

## rBiFC assays
Full length cDNA of BZR1, BZR1$^{C63S}$, bZIP30 or bZIP30$^{S18/T22A}$ were amplified by PCR and cloned into the pDONR221-P1P4 or pDONR221-P3P2 vector using the BP recombination reaction (Invitrogen), respectively. The 2in1 LR reaction were performed with destination vector pBiFCt-2in1-NN and different pDONR221 vectors. Agrobacterial suspensions containing *p35S::BZR1-nYFP-p35S::RFP-p35S::bZIP30-cYFP*, *p35S::BZR1$^{C63S}$-nYFP-p35S::RFP-p35S::bZIP30-cYFP*, *p35S::BZR1-nYFP-p35S::RFP-p35S::bZIP30$^{S18/T22A}$-cYFP*, *p35S::bZIP30-nYFP-p35S::RFP-*

*p35S::bZIP30-cYFP*, *p35S::bZIP30-nYFP-p35S::RFP-p35S::bZIP30$^{S18/T22A}$-cYFP* or *p35S::SmBZR1-nYFP-p35S::RFP-p35S::SmbZIP30-cYFP* constructs were injected into the epidermis of tobacco leaves. The transfected plants were kept in the greenhouse for at least 48 h at 22 °C, and then treated with mock solution or 1 mM H$_2$O$_2$ for 3 h. Fluorescent signals were visualized by using the LSM880 laser scanning confocal microscope (Zeiss) and the signal intensities of YFP and RFP were determined by ImageJ software.

## Stomatal index quantification and epidermal cell size determination
The Arabidopsis cotyledons were decolored in 75% ethanol 25% acetic acid solution. Then cotyledons were submerged into Hoyer's solution until turning transparent completely for microscope observation. The ratio of the number of guard cells and total epidermal cells were calculated as stomatal index. The pictures of cotyledon abaxial epidermis were edited in Adobe Illustrator software based on DIC picture. The size of epidermal cells was calculated by ImageJ software.

## Chromatin immunoprecipitation (ChIP) assay
Seedlings of *pbZIP30::bZIP30-YFP*, and *p35S::YFP* were grown on 1/2 MS solid medium for 12 days under a long day photoperiod, harvested and cross-linked in 1% formaldehyde for 30 min using vacuum. Immunoprecipitation was performed as previously described, using GFP-Trap agarose beads (Chromotek). ChIP products were analyzed by qPCR, and the fold enrichment was calculated as the ratio between *pbZIP30::bZIP30-YFP*, and *p35S::YFP*, and then normalized by *PP2A* (*At1g13320*) gene, which was used as an internal control. The ChIP experiments were performed with three biological replicates.

## RT-qPCR analysis
Plants of different genetic background were grown 1/2 MS medium under a 12 h light/12 h dark photoperiod with a 100 μM m$^{-2}$ s$^{-1}$ light intensity for 28 days. Stomata from rosette leaves were enriched thought Stomata Tape-Peel method[50]. The rosette leaves epidermis on tape were frozen in liquid nitrogen. The harvested epidermis were then used for total RNA extraction (TransGen Biotech). First-strand cDNAs were synthesized by employing RevertAid reverse transcriptase (Thermo Fisher Scientific) and were subsequently utilized as templates for quantitative PCR. The quantitative PCR analysis was performed in CFX connect real-time PCR detection system (Bio-Rad) using of SYBR green reagent (Roche, Basel, Switzerland), in combination with gene-specific primers (Supplementary Data 1).

## Transient gene expression assays
Protoplast isolation and PEG transformation was performed based on former protocol[51]. Plasmid DNAs were extracted using the Qiagen Plasmid Maxi Kit according to manufacturer instructions. Approximate 5 × 10$^4$ isolated mesophyll protoplasts were used for transformation with a mixture of 10 μg of DNA and incubated under dark condition overnight. Protoplasts were harvested through centrifugation and lysed in 100 μL passive lysis buffer (Promega). Firefly and Renilla (as internal standard) luciferase activities were measured using a dual-luciferase reporter kit (Promega).

## In vitro kinase assay and phosphopeptide analysis
MBP-KIN10, MBP-bZIP30 and GST-SnAK2 proteins were expressed and purified from *Escherichia coli*. MBP-bZIP30 was incubated with MBP-KIN10 and/or GST-SnAK2 in the kinase buffer (25 mM HEPES pH 7.4, 10 mM MgCl$_2$, 50 mM KCl, 1 mM DTT and 30 μM cold ATP) containing [γ$^{32}$P] ATP (10 μCi) for 1 hour. The reaction was stopped by addition of 5× loading buffer. Proteins were separated by 8% SDS-PAGE. After nonradioactive in vitro kinase assays, proteins were digested with trypsin and endoproteinase Asp-N. The digested peptide mixtures were injected into QExactive HF hybrid quadrupole-Orbitrap mass

spectrometer (ThermoFisher Scientific) for Mass spectrometer analysis. The phosphorylated residues in MBP-bZIP30 were identified by Maxquant software.

## Statistical analysis

Statistical analysis was performed by GraphPad Prism version 9.0 software. Student $t$ test was applied to calculate the P values between two samples. For comparing more than two samples, One-way ANOVA with multiple comparisons test was used. For comparing more than two samples with two different factors, two-way ANOVA with multiple comparisons test was used. The sample size in this study is mainly determined according to prior experiences, which are based on the reproducibility and statistical significance of the results during the experiments. The results of the statistical analyses are shown in the Supplementary Data 2.

## Reporting summary

Further information on research design is available in the Nature Portfolio Reporting Summary linked to this article.

## Data availability

All data generated or analyzed during this study are included in the main text and Supplementary Information. All Arabidopsis genes involved in this study can be found at TAIR (www.arabidopsis.org), with the following accession numbers: *KIN10* (AT3G01090), *bZIP30* (AT2G21230), *BAM1* (AT3G23920), *AMY3* (AT1G69830), *BZR1* (AT1G75080), Other species genes involved in this study can be found at PLAZA with following accession numbers: SMO358G0686 (*SmKIN10*) (https://bioinformatics.psb.ugent.be/plaza/versions/plaza_v5_dicots/genes/view/SMO358G0686), SMO351G0386 (*SmBZR1*) (https://bioinformatics.psb.ugent.be/plaza/versions/plaza_v5_dicots/genes/view/SMO351G0386), SMO143G0256 (*SmbZIP30*) (https://bioinformatics.psb.ugent.be/plaza/versions/plaza_v5_dicots/genes/view/SMO143G0256). The protein sequences of bZIP30 and BZR1 orthologs in represented species involved in this study can be found at PLAZA. The raw file for Mass spectrometer analysis of KIN10-dependent bZIP30 phosphosites has been deposited in the ProteomeXchange partner repository under accession number PASS05862. Extra data are available from the corresponding author upon request. Source data are provided with this paper.

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

## Acknowledgements
We thank Haiyan Yu, Yuyu Guo, and Xiaomin Zhao from the Analysis and Testing Center of SKLMT (State Key Laboratory of Microbial Technology, Shandong University) for assistance with the laser scanning confocal microscopy. We thank Jianyong Shen from Xishuangbanna Tropical Botanical Garden and Hongsheng Jiang from Wuhan Botanical Garden for providing *Selaginella doederleinii* and *Marsilea quadrifolia* plants. This work was supported by grants from the National Natural Science Foundation of China (32325006 to M.-Y.B., 32300927 to W.S., 32270351 to C.H., 32070210 to M.-Y.B., 31970306 to M.F., 32100236 to J.L.), Agricultural Variety Improvement Project of Shandong Province (2022LZGC001 to M.-Y.B.), Science and Technology Department of Shandong Province (ZR2022YQ20 to C.H., 2018ZX08005-01Bb to C.H., 2018ZX08005-01B-004 to C.H.) and China Postdoctoral Science Foundation (2023M732077 to W.S.).

## Author contributions
W.S. and C.H. together designed the experiments. W.S. and C.H. performed a statistical analysis of the content of starch in guard cells and stomatal apertures. W.S. N.Z. performed transient expression. C.H. and Y.L. performed the kinase assays and mass spectrometer analysis. W.S. performed RT-qPCR. ChIP-qPCR, western blot, protein pull-down assay, and subcellular location analysis, and all other experiments. W.S., Y.L., J.L., L.Y. and N.Z. generated *pKIN10::KIN10-YFP/kin10*, *pKIN10::NLS-KIN10-YFP/kin10*, *pKIN10::NES-KIN10-YFP/kin10*, *pKIN10::KIN10-YFP/cat2*, *pKIN10::KIN10-YFP/rbohD rbohF*, *pKIN10::KIN10-YFP/p35S::CAT2-myc*, *p35S::KIN10-myc/bzip30*, *pBAM1::GFP/bzip30*, *pBAM1::GFP/kin10*, *pbZIP30::bZIP30-YFP/bzip30*, *pbZIP30::bZIP30^{S18/T22A}-YFP/bzip30* and *pbZIP30::bZIP30^{S18/T22D}-YFP/bzip30* transgenic plants. M.F., B.Z., and M.-Y.B. provided the critical discussion. W.S. and C.H. wrote the manuscript.

## Competing interests
The authors declare no competing interests.
