## [Peer Review file · Nature Communications]

REVIEWER COMMENTS

Reviewer #1 (Remarks to the Author):

This manuscript describes the importance of H₂O₂ accumulation within guard cells for the degradation of starch and the opening of stomata in response to light. Using a molecular genetic approach and biochemical assays, the authors investigate the molecular mechanisms underlying H₂O₂ regulatory effect on stomatal function. In general, the work deals with a very important topic and in some instances provide some novel data helping to clarify the role of ROS in the regulation of stomatal function. However, I have several reservations about the choice of plant material and the interpretation of mutant phenotypes, as I discuss in detail below.

Major issues

- The use of young seedlings to study stomatal behaviour and metabolism is questionable. The authors perform most of their experiments using seedlings of 8-10 days (actually not even consistently, each experiment is done with seedlings that have been in the light for 4, 8 or 10 days, not clear why).

While there are similarities in stomatal behavior between cotyledons and mature leaves, there are also notable differences, primarily related to their roles in the plant's life cycle. Cotyledons are adapted for early seedling development and may have stomatal characteristics that are optimized for their specific functions, which can differ from those of mature leaves. The exact behavior of cotyledon stomata can be influenced by environmental conditions and the specific requirements of the plant at different growth stages. For example, cotyledons typically have a lower stomatal density compared to mature leaves. This lower density may affect the rate of gas exchange and transpiration in cotyledons. Furthermore, the responsiveness of cotyledon stomata may vary depending on the stage of development and the specific environmental conditions. Mature leaves often exhibit more fine-tuned and sophisticated stomatal responses.

Also, using cotyledons to study metabolism may not be representative of the situation in a mature leaf. Cotyledons primarily serve as seed storage organs and may not engage in as much photosynthesis as mature leaves. The stomatal behaviour in cotyledons may be more focused on water conservation and gas exchange for metabolic processes other than photosynthesis. While cotyledons can carry out some metabolic processes, their primary role is not photosynthesis. In contrast, mature leaves are specialized for photosynthesis. The types and levels of enzymes involved in carbohydrate metabolism may differ between cotyledons and mature leaves. The regulation of carbohydrate metabolism can vary between cotyledons and mature leaves. While there are shared aspects of carbohydrate metabolism in both cotyledons and mature leaves, the specific metabolic pathways, priorities, and functions differ due to their roles in different stages of the plant's life cycle. Cotyledons are adapted for early seedling growth and rely on stored reserves, while mature leaves are optimized for photosynthesis and the production of new carbohydrates.

So, I am not sure the results obtained from experiments conducted in cotyledons can be taken as representative for mature leaves.

- My second biggest reservation is about the interpretation of many mutant phenotypes, particularly, the guard cell starch phenotypes. The authors are not expert in metabolism, and sadly this appears very clear when they try to interpret the mutant phenotypes. I just give you one or two examples:

- Lines 119-120. *cat2* mutant has more open stomata but they seem to open less in response to light. Description of mutant phenotype is not correct.

- Lines 128-129. First of all, authors cannot talk about rate of starch breakdown, as they only measured one time point. Second, it's true that *cat2* has a bit less starch than WT at the EoN, but it seem to degrade similar amount as WT! So, conclusion at lines 129-130 is not matching the results. To me, because *cat2* accumulates less starch but degrades similarly to WT, it suggests that the mutation is not affecting GC starch degradation in response to light, but rather the ability to accumulate starch.

- I have reservations about the phenotype of *kinb2* and overexpressors. I explain below:

- Extended fig. 5a: *kinb2* mutant has slightly more starch at the EoN, then degrades starch similarly to WT. Again, authors cannot talk about speed of degradation. This is anyway a very mild phenotype. They do not have data about double *kinb1b2*? Maybe there is redundancy. Anyway, in the over-expressor 35S::*KINB2* where is localized *KIN10*? What happens if we treat 35S::*KINB2* with H_2O_2 ? Conclusions are vague and do not fully reflect the presented data.

- Extended fig. 4b: forced cytosolic localization of *KIN10* has impact of amount of GC starch, accumulate less, so here as well it's an impact on synthesis, not degradation.

- Lines 301-302. I have again reservation in the interpretation of the GC starch data. In Fig. 5a, Col-0 BL has less starch than Col-0 Mock, so it is difficult to conclude from these data that it degraded more. Same applies to stomatal opening. Col-0 BL has more open stomata at the EoN compared to Col-0 Mock.

- Also the phenotype of *kin10* and *bzip30*, already under control conditions they are impaired... it's not a conditional phenotype or it is not treatment dependent (H_2O_2 or BL treatment). Maybe results from a pleiotropic phenotype.

- The third major concern I have is about the conservation of H_2O_2 distribution in vascular plants. The authors have not done enough experiments to come to this conclusion. So the title is inappropriate.

- Fig. 6a. The pattern of H_2O_2 staining in *Marsilea* significantly differs from that of other species. This is not discussed.

- Lines 363-364. Based on my comment above, I do not agree that the pattern of H_2O_2 accumulation is conserved across vascular plants.

- Lines 370-372. I also have strong reservations about their definition of spatial distribution, Normally, spatial distribution refers to subcellular distribution (e.g. cytosol vs chloroplasts). I

disagree that the authors showed spatial pattern of H₂O₂. It's simply the accumulation of H₂O₂ to a certain concentration. Title does not reflect the data.

- One additional serious reservation is about the choice to study ROS by using the dye. Based on my personal experience, the H₂DCFDA is not reliable, and certainly risky to use it to quantify levels of H₂O₂. The intensity of the staining signal does not necessarily reflect the actual amount of H₂O₂. Nowadays, there are very powerful genetically encoded biosensors for ROS, which are much better suited to study changes in ROS amounts.

Additional concerns:

- Lines 36-39. In amy3bam1 mutant, rate of proton pumping is similar that of WT, so the suggestion that Glc generated by starch degradation is necessary for energizing transport across the PM is not actually supported by the available data in the literature.

- In experiments in Fig. 1, I am concerned about the use of KI. What are potential side effects of KI?

- Lines 131-134. Treatment with H₂O₂ and stomatal opening was done previously (their own work, ref 13). The claim that it's new is not correct.

- kin10 is a sugar sensors, also very important for mesophyll cells. The authors do not show whether kin10 a starch and/or sugar phenotype under their conditions.

- KIN10 has an impact on GC starch degradation but we do not know if this is via H₂O₂ levels. What are the levels of H₂O₂ in kin10? And in kin10cat2?

- All these transgenics lines created with 35S promoter. Do we expect any side effects? Starch and sugars levels in the leaves?

- Lines 230-231. Extended fig. 7a: treatment of WT GCs with H₂O₂. It's not clear that treatment enhanced GC starch degradation, as the levels of starch in WT treated with H₂O₂ are lower than Mock. It seems these results are not consistent with results presented in Extended Fig. 2D.

- Gene expression analyses were performed in entire seedlings, are not guard cell-specific. It's available an easy protocol to isolate GCs for qPCR analyses.

- In general, how do the author reconcile that fact that both AMY3 and BAM1 are redox-regulated enzyme, when they suggest that oxidation modify BZR1 to activate transcription? I cannot imagine how these two degrading enzymes can work under what the authors suggest to be oxidizing conditions.

- Discussion is repetitive and lack depth.

- Line 103. It's already known that the effect is concentration dependent.

- Line 122. It's the PROPER accumulation of H₂O₂ that is important.

- Line 128. It's not rate, they just did one time point.

- Lines 145-148. It not the spatial accumulation, it's the amount of H₂O₂.
- Analyses of GCstarch and stomatal aperture of kin10 and cat2 mutant in Fig. 2 is redundant.
- Experiments in Extended fig. 3: DAPI staining?
- Figure legend Fig. 3a, a, KIN10 interacts with bZIP30 in yeast. This is uninformative. Please write a proper figure legend.
- Material and methods are incomplete.
- English is very poor.

Reviewer #2 (Remarks to the Author):

In a previous manuscript (Shi et al 2022; Nature Comms. 13: 5040), the authors of this current manuscript established an exciting connection between H₂O₂ distribution and stomata development. This current manuscript extends their previous discovery by studying the role of H₂O₂ distribution on stomata function – particularly stomata opening in response to light. They demonstrate that endogenous H₂O₂ levels in guard cells (which they manipulate using genetics as well as pharmacological agents) affects the ability of stomata to open in response to light. They demonstrate that H₂O₂-mediated stomata regulation requires KIN10, the catalytic subunit of SnRK kinase. They present a model where H₂O₂ induces KIN10 movement into the nucleus in guard cells, where it phosphorylates a bZIP30 transcription factor. Using mutants, they convincingly demonstrate that bZIP30 is important for stomata opening. They also show that the phosphorylated bZIP30 interacts with BZR1, and they act together to activate AMY3/BAM1 expression. These two enzymes have been previously shown to activate guard cell starch degradation.

Overall, the experiments are of high quality. The manuscript is complete in that the authors have put together a full, plausible pathway – from the signalling molecule H₂O₂, all the way to the expression of starch degradation enzymes. The authors also provide convincing evidence that nuclear localisation of KIN10 in guard cells is important for guard cell regulation, and the implication of bZIP30 in stomata regulation is also novel. For these reasons, I think this is a strong manuscript, but I have a few questions/issues (mainly regarding presentation) for the authors to consider.

Major points:

1. Lines 71-79: I felt this part of the introduction could benefit from revision. I do not think it is completely necessary to mention all the effects of H₂O₂ in the development of apical meristems. This part could probably be shortened. However, the reader would benefit from a clear definition of what is meant by “spatial pattern” or “spatial accumulation” of H₂O₂ throughout the manuscript. Clearly state what spatial scale is relevant - sub-cellular, cell-type, tissue level or organ level.

2. Line 111: A word of caution regarding KI treatments: Potassium iodide influences H₂O₂ levels, but it is also a potent starch stain. It is one of the key ingredients of Lugol’s solution. It is possible that KI can block starch degradation due to it complexing with the starch itself. For this reason, I find the genetic manipulations of H₂O₂ (e.g., the analysis of *cat2* mutants) more convincing than the pharmacological treatments. I am not saying such experiments should be removed, but I think the possible effects of KI on the starch should be directly discussed in the manuscript, unless the authors can prove that KI has no effect on starch degradation *in vivo*.

3. Lines 132-147: Here, I think the authors need to better explain what is defined as a “normal” and “stress” concentration. Also, Lines 132-133 probably need a citation.

4. Lines 181-182: It is striking that the nuclear localisation of KIN10 changes in the *cat2* and *rbohD rbohF* mutants. However, this is the KIN10 localisation adopted after long-term exposure to altered H₂O₂ levels. However, the paper focuses on more dynamic changes that lead to rapid stomata opening in response to light. In this regard, I wonder whether the authors have tried simply treating the plants with different H₂O₂ concentrations, as in the experiment of Extended Data Fig 2, and assessed whether the KIN10 localisation in the nucleus is most pronounced in the 10-30 μ M range?

5. Lines 202-206: The introduction of bZIP30 is quite abrupt. The authors say they chose to focus on it because it was in the Y2H and it was related to bZIP29 (which is known to be expressed in stomata). If this is the case, I am wondering why they did not test for an interaction between KIN10 and bZIP29? I am guessing that bZIP29 was not present in the Y2H screen with KIN10, but this needs to be clearly stated.

6. Line 220: What is meant by “universal”? The images in Extended Data Fig 6a-d are too small to see the localisation clearly, and the cell area plots need to be explained in the text. What are they meant to show?

7. Lines 246-248: Here it is stated that the phosphomimic mutant only “rescues” the defective stomatal opening and guard cell starch degradation phenotypes of *bzip30*. However, why does it only “rescue”? Wouldn’t the authors expect that this phosphomimic bZIP30 is constitutively active, and should rescue beyond the WT version? Indeed, in Fig 3i and 3j, the phosphomimic lines have an

EoN stomata aperture and starch contents that are statistically different from the Col-0. This should be discussed.

8. Line 265: In this assay, BAM1 and AMY3 activation is indeed further enhanced when KIN10 and bZIP30 are co-expressed. But why is KIN10 alone able to very substantially activate BAM1 and AMY3 expression? Could it be working with an endogenous bZIP30 in the cells? If the phosphomimic form of bZIP30 is used in this assay, would it match the expression levels achieved with KIN10+bZIP30? Also, I could not find the methods related to this experiment (I am very sorry if I missed it) – in what system were these proteins transiently expressed?

Minor points:

In general, I find that this manuscript needs a thorough proofread for spelling and minor grammatic errors. I can only mention a few here:

Line 7: omit “the”

Line 11: “prefers” sounds like the protein makes an active choice to be there.

Line 23: environments

Line 23: “are surrounded”

Line 32: “the guard cell”

Line 33: “on the other side”

Line 56: “Under insufficient energy supply”

Line 73: This sentence should not start with “And”.

Line 80: “epidermis”

Line 99: “previous”

Line 284: “interacts”

Line 318: “on the other side”

Line 325: “which could help induce BAM1...”

Line 399: “degradation”

Line 425: omit “of”

Line 447: omit “does”

Line 535: “pbZIP30:bZIP30”

Reviewer #3 (Remarks to the Author):

The MS by Shi et al explores the potential distribution and role of H₂O₂ in starch degradation in guard cells and the impact on stomatal aperture. A considerable amount of work has gone into the manuscript using a range of approaches and techniques and a great deal of data are presented, however there are still some major concerns with the MS. Additionally, due to the large number of approaches and questions addressed, the MS is difficult to follow in places and lacks focus. This would reduce appeal to a broad audience expected by this journal.

The rationale for the importance of this study is not clear, the importance of the findings and conclusions are not put into context.

There is a lack of convincing evidence for many of the statements made and this is acknowledged by the authors who use phrases such as “...this might suggest” and “...this might show. More details regarding this are highlighted below.

The MS has a focus (including the title) on spatial distribution of H₂O₂ – the imaging processes reporting this are not entirely clear or convincing and also not quantified apart from a few small images.

The abstract introduces a number of genes, proteins, TF that are not explained and expects the reader to have a clear understanding of the interactions between all of these.

KI was used to reduced H₂O₂ – how can the authors be completely convinced that the amount and entry into the tissue was identical between experiments or uniform across tissues and experiments?

Line 25 – “..believed to “ – the role of stomata is well established in plant physiology in leaves and green tissue.

The authors only consider stomatal responses to blue light – however this is not the only pathway for guard cell responses to light, or starch degradation and furthermore the response to blue light is not conserved across species.

There are several paragraphs in the introduction that are focussed on a small part of the MS – e.g the section on the early land plants. This is not the focus of the paper, however it is a large focus of the MS.

Line 68 –“ forces ancient plant to adapt the more abundant...”??L

The MS needs editing for English and grammar – and there are numerous odd phrases used e.g. line 33 “one the other side” – on the other hand?; Line 55 “Under energy supplies insufficient” – unclear.

Line 82 – which ROS scavenging genes.

The images suggest that the ROS is accumulating in the guard cell chloroplasts. What are the wavelengths of detection for the H2DCFDA? The staining is not in all cells in some of the images e.g. Extended data Fig.2 – explain.?

rhobD and F need to be introduced to the reader.

It is not clear if images and apertures were taken on the same material – the material and methods suggest not. This is not appropriate as it is clear that there is significant heterogeneity within the images. Furthermore, there is limited quantification of images to directly relate activities to apertures.

It would be useful to see images of the starch granules – rather than just the quantification of these.

Line 128 – “accelerated breakdown” or less observed when measured. Was there a control at the start to ensure that breakdown was accelerated?

The authors do not provide any information about where the H₂O₂ is generated in non-stress tissues and how this differs from stress conditions when stomatal closure is induced by H₂O₂.

Line 141 – there is no figure for stress conditions?

Extended data set 2 – the spatial variation in H₂O₂ production is not convincing in these images and there is limited quantification of this.

Line 152 – “a hypothesis regarding ...” – what was the hypothesis?

Line 160+ - loss of Kin10 – prevented starch breakdown but was it because the starch was low to start with?

Can the authors confirm that the transgenic and mutants used have similar expression patterns and activities in all tissues measured? This could account for some of the spatial variation and general variation in measurements?

Although aperture was measured, for full functional assessment – physiology would be required. As there is change in both density and function – one could simply be compensating for the other.

YFP signal could be influenced by many physical factors within the leaf as well as the measurement approaches and is therefore not entirely reliable.

The later part of the MS focuses on the interactions between various proteins and TF and uses a range of approaches to assess them – this feels and reads like a different paper, and is often not linked to function.

The discussion section does not fully reflect the findings and focuses on the evolutionary aspects of the MS, which is the simplest and least explored apart from the H₂O₂ potentially been detected

and the same genes being present. The discussion should focus on the main findings and put these into context. This is another major weakness of the MS.

Line 426 – it has been clearly shown that CO₂ fixation does take place in guard cells and that all the enzymes are present. The authors need to correct this to not mislead readers.

Line 427 – although NTTT were shown to be important in this paper – application of DCMU also showed that there was a role for GC electron transport.

Reviewer #1 (Remarks to the Author):

This manuscript describes the importance of H₂O₂ accumulation within guard cells for the degradation of starch and the opening of stomata in response to light. Using a molecular genetic approach and biochemical assays, the authors investigate the molecular mechanisms underlying H₂O₂ regulatory effect on stomatal function. In general, the work deals with a very important topic and in some instances provide some novel data helping to clarify the role of ROS in the regulation of stomatal function. However, I have several reservations about the choice of plant material and the interpretation of mutant phenotypes, as I discuss in detail below.

Major issues

- The use of young seedlings to study stomatal behaviour and metabolism is questionable. The authors perform most of their experiments using seedlings of 8-10 days (actually not even consistently, each experiment is done with seedlings that have been in the light for 4, 8 or 10 days, not clear why).

While there are similarities in stomatal behavior between cotyledons and mature leaves, there are also notable differences, primarily related to their roles in the plant's life cycle. Cotyledons are adapted for early seedling development and may have stomatal characteristics that are optimized for their specific functions, which can differ from those of mature leaves. The exact behavior of cotyledon stomata can be influenced by environmental conditions and the specific requirements of the plant at different growth stages. For example, cotyledons typically have a lower stomatal density compared to mature leaves. This lower density may affect the rate of gas exchange and transpiration in cotyledons. Furthermore, the responsiveness of cotyledon stomata may vary depending on the stage of development and the specific environmental conditions. Mature leaves often exhibit more fine-tuned and sophisticated stomatal responses.

Also, using cotyledons to study metabolism may not be representative of the situation in a mature leaf. Cotyledons primarily serve as seed storage organs and may not engage in as much photosynthesis as mature leaves. The stomatal behaviour in cotyledons may

be more focused on water conservation and gas exchange for metabolic processes other than photosynthesis. While cotyledons can carry out some metabolic processes, their primary role is not photosynthesis. In contrast, mature leaves are specialized for photosynthesis. The types and levels of enzymes involved in carbohydrate metabolism may differ between cotyledons and mature leaves. The regulation of carbohydrate metabolism can vary between cotyledons and mature leaves. While there are shared aspects of carbohydrate metabolism in both cotyledons and mature leaves, the specific metabolic pathways, priorities, and functions differ due to their roles in different stages of the plant's life cycle. Cotyledons are adapted for early seedling growth and rely on stored reserves, while mature leaves are optimized for photosynthesis and the production of new carbohydrates. So, I am not sure the results obtained from experiments conducted in cotyledons can be taken as representative for mature leaves.

Response: Thank you for pointing this out. We appreciate your explanation of the difference in starch metabolism, stomatal movement, and H₂O₂ content between cotyledons and mature leaves, and we fully agree with your perspective.

To investigate whether H₂O₂ is also accumulated in guard cells of mature leaves, and plays a similar role in guard cell starch metabolism and stomatal movement in both cotyledon and mature leaves grown under normal growth conditions, we conducted an analysis of guard cell H₂O₂ content, starch metabolism and stomatal movement in cotyledons and 4-week-old rosette leaves. The results revealed a significant enrichment of H₂O₂ in guard cells on both cotyledon and rosette leaves, using various H₂O₂ detection methods. Furthermore, we observed impaired guard cell starch degradation and stomatal opening in H₂O₂-deficient plants such as *p35S::CAT2-myc* and *rbohD rbohF*, on both cotyledon and rosette leaves (Fig. 1i-l). Additionally, *kin10*, *cat2 kin10*, *bzip30*, and *bzip30 p35S::KIN10-myc* plants exhibited impaired guard cell starch degradation and stomatal opening in the rosette leaves (Fig. 2, and Fig. 3e-g). These findings indicate that H₂O₂ is enriched in the guard cells of both cotyledons and mature leaves, and it plays a crucial role in light-induced guard cell starch degradation and stomatal opening in both types of leaves.

Fig. 1 Specific accumulation of H₂O₂ in guard cells under normal conditions is required for light induced stomatal opening in Arabidopsis.

a-f Measurement of H₂O₂ in the epidermal cells on rosette leaves of four-weeks-old Col-0 plants using H₂DCFDA staining (**a, b**), BES-H₂O₂-Ac staining (**c, d**) and HyPer fluorescent (**e, f**). Col-0 or *pHyPer* transgenic line were grown on 1/2 MS medium under a 12 h light/12 h dark photoperiod with a light intensity of 100 $\mu\text{M m}^{-2} \text{s}^{-1}$ for 28 days. The intensities of H₂DCFDA, BES-H₂O₂-Ac and HyPer fluorescent signals were analyzed using at least 100 guard cells from 10 different plants with ImageJ software. “PC” in confocal images represents pavement cell, “GC” represents guard cell. “Red” represents reduced state, “Ox” represents oxidized state. Scale bars represent 20 μm . Asterisk indicated statistically significant differences between the samples (Student’s t test, **** $p < 0.0001$). **g, h** Measurement of H₂O₂ content in guard cells on rosette leaves of Col-0, *cat2*, *p35S::CAT2-myc*, *rbohD rbohF* plants using BES-H₂O₂-Ac staining. Plants were grown on 1/2 MS medium under a 12 h light/12 h dark photoperiod with the 100 $\mu\text{M m}^{-2} \text{s}^{-1}$ light intensity for 28 days. H₂DCFDA and BES-H₂O₂-Ac signaling intensities in at least 100 guard cells from 10 different plants were analyzed using ImageJ software. Different letters above the bars indicate statistically significant differences between samples (One-way ANOVA analysis followed by uncorrected Fisher’s LSD multiple comparisons test, $p < 0.05$). **i-l** Quantification of stomatal apertures (**i, j**) and

guard cell starch granules (**k, l**) in rosette leaves of Col-0, *cat2*, *p35S::CAT2-myc* and *rbohD rbohF* plants. Plants were grown on 1/2 MS medium under a 12 h light/12 h dark photoperiod with a $100 \mu\text{M m}^{-2} \text{s}^{-1}$ light intensity for 28 days. The starch granules area and the ratio of stomatal aperture width to length from more than 100 guard cells of at least 4 different plants were measured using ImageJ software. EoN means the end of light, and Light 1 h means the white light illumination for 1 hour after the end of night. Scale bars in the pictures of stomata represent $20 \mu\text{m}$. Different letters above the bars indicate statistically significant differences between samples (Two-way ANOVA analysis followed by Tukey's multiple comparisons test, $p < 0.05$). **m** qRT-PCR analysis of the expression of *BAM1* in Col-0, *cat2*, *p35S::CAT2-myc* and *rbohD rbohF* plants. Seedlings of Col-0, *cat2*, *p35S::CAT2-myc* and *rbohD rbohF* were grown 1/2 MS medium under a 12 h light/12 h dark photoperiod with a $100 \mu\text{M m}^{-2} \text{s}^{-1}$ light intensity for 28 days. Stomata from rosette leaves were enriched through Stomata Tape-Peel method. Error bars indicate standard deviation (S.D.). Different letters above the bars indicate statistically significant differences between samples (Student t' test, $p < 0.05$).

Figure 2. KIN10 is required for H₂O₂ promoted light-induced guard cell starch degradation and stomatal opening.

a-c Quantification of guard cell stomatal apertures (**a, b**) and starch granules (**c**) of rosette leaves in Col-0, *kin10*, *p35S::KIN10-myc* plants. **d-f** Quantification of guard cell stomatal apertures (**d, e**) and starch granules (**f**) of rosette leaves in Col-0, *kin10*, *cat2* and *kin10 cat2* plants. Plants were grown on 1/2 MS medium under a 12 h light/12 h photoperiod with a $100 \mu\text{M m}^{-2} \text{s}^{-1}$ light intensity for 28 days. The starch granules area and the ratio of stomatal aperture width to length from more than 100 guard cells in at least 4 different plants were measured using ImageJ software. Scale bars in the pictures of stomata represent $20 \mu\text{m}$. EoN means the end of light, and Light 1 h means the white light illumination for 1 hour after the end of night. Different letters above the bars indicate statistically

significant differences between samples (Two-way ANOVA analysis followed by Tukey's multiple comparisons test, $p < 0.05$).

Fig. 3 KIN10 interacts with and phosphorylates bZIP30. e-g, Quantification of guard cell stomatal apertures (e, f) and starch granules (g) in Col-0, *bzip30*, *p35S::KIN10-myc* and *bzip30 p35S::KIN10-myc* plants. Seedlings of Col-0, *bzip30*, *p35S::KIN10-myc*, *bzip30 p35S::KIN10-myc* plants were grown on $\frac{1}{2}$ MS medium under a 12 h light/12 h photoperiod with a $100 \mu\text{m}^2 \text{s}^{-1}$ light intensity for 28 days. The starch granules area from more than 100 guard cells in at least 12 different plants and the ratio of stomatal aperture width to length from more than 100 guard cells in at least 5 different plants were measured using ImageJ software. EoN means the end of light, and Light 1 h means the white light illumination for 1 h after the end of night. Different letters above the bars indicate statistically significant differences between samples (Two-way analysis ANOVA followed by Tukey's multiple comparisons test, $p < 0.05$).

- My second biggest reservation is about the interpretation of many mutant phenotypes, particularly, the guard cell starch phenotypes. The authors are not expert in metabolism, and sadly this appears very clear when they try to interpret the mutant phenotypes. I just give you one or two examples:

Response: Sorry for this misleading. It has been reported that guard cell starch degradation is required for stomatal opening during the transition from dark to light (Horrer et al., 2016). β -amylase (BAM1) is specifically expressed in guard cells and is responsible for guard cell starch degradation. Studies have shown that *bam1-1* mutant consistently exhibits high levels of starch in guard cells at both the end of night (EoN) and upon 1 hour light exposure (Light 1 h) (Horrer et al., 2016; Han et al., 2022). Therefore, BAM1 is required for guard cell degradation. Similarly, BR-deficient mutant *det2* and BR-insensitive mutant *bzr-h* exhibit undegradable starch content in guard cells and impair stomatal opening (Li et al., 2020; Han et al., 2022). RT-qPCR analysis

showed that *BAMI* expression is inhibited in both *det2* and *bzr-h* mutants (Han et al., 2022). These findings indicated that BR signaling and BZR1 function are required for starch degradation in guard cells and stomatal opening through regulation of *BAMI* expression. In this study, *p35S::CAT2-myc* transgenic line and *rbohD rbohF* double mutant, which contain lower level of H₂O₂ in guard cells, exhibit the stable starch content before and after 1 hour light exposure, and the impaired stomatal opening. RT-qPCR analysis showed that *BAMI* expression is also inhibited in *p35S::CAT2-myc* transgenic line and *rbohD rbohF* mutant. Therefore, accumulation of H₂O₂ in guard cells is required for light induced stomatal opening and guard cell starch degradation.

Horrer, D. et al. Blue light induces a distinct starch degradation pathway in guard cells for stomatal opening. *Curr Biol.* **26**, 362-370 (2016).

Li, J.G. et al. Brassinosteroid and hydrogen peroxide interdependently induce stomatal opening by promoting guard cell starch degradation. *Plant Cell* **32**, 984-999 (2020).

Han, C. et al. TOR promotes guard cell starch degradation by regulating the activity of β -AMYLASE1 in Arabidopsis. *Plant Cell* **34**, 1038-1053 (2022).

- Lines 119-120. *cat2* mutant has more open stomata but they seem to open less in response to light. Description of mutant phenotype is not correct.

Response: Thank you for pointing this out. We have revised the description of *cat2* mutant phenotype as follows: The results revealed that *cat2* mutant displayed larger stomatal apertures compared to wild-type plants under both dark and 1-hour light exposure conditions.

- Lines 128-129. First of all, authors cannot talk about rate of starch breakdown, as they only measured one time point. Second, it's true that *cat2* has a bit less starch than WT at the EoN, but it seems to degrade similar amount as WT! So, conclusion at lines 129-130 is not matching the results. To me, because *cat2* accumulates less starch but degrades similarly to WT, it suggests that the mutation is not affecting GC starch degradation in response to light, but rather the ability to accumulate starch.

Response: Thank you for pointing this out. To investigate the role of accumulated H₂O₂ in guard cells, we analyzed the starch content phenotype of the mutants with different

levels of H₂O₂ in guard cells. The *p35S::CAT2-myc* transgenic plants and *rbohD rbohF* double mutant, which have lower level of H₂O₂ in guard cells, exhibited the stable starch content before and after 1 hour light exposure, and the impaired stomatal opening. RT-qPCR analysis showed that *BAM1* expression was also inhibited in *p35S::CAT2-myc* and *rbohD rbohF*. Therefore, the accumulation of H₂O₂ in guard cell is required for light induced stomatal opening and guard cell starch degradation. Additionally, we observed that an increase in *BAM1* expression increased in *cat2* mutant, which led to decrease of starch content and larger stomatal apertures under both the end of night and 1 hour light exposure conditions. We have revised the description of starch degradation about *cat2* mutant (Fig. 1g-m).

Fig. 1 Specific accumulation of H₂O₂ in guard cells under normal conditions is required for light induced stomatal opening in Arabidopsis.

g, h Measurement of H₂O₂ content in guard cells on rosette leaves of Col-0, *cat2*, *p35S::CAT2-myc*, *rbohD rbohF* plants using BES-H₂O₂-Ac staining. Plants were grown on 1/2 MS medium under a 12 h light/12 h dark photoperiod with the 100 µM m⁻² s⁻¹ light intensity for 28 days. H₂DCFDA and BES-H₂O₂-Ac signaling intensities in at least 100 guard cells from 10 different plants were analyzed using ImageJ software. Different letters above the bars indicate statistically significant differences between samples (One-way ANOVA analysis followed by uncorrected Fisher's LSD multiple

comparisons test, $p < 0.05$). **i-l** Quantification of stomatal apertures (**i, j**) and guard cell starch granules (**k, l**) in rosette leaves of Col-0, *cat2*, *p35S::CAT2-myc* and *rbohD rbohF* plants. Plants were grown on 1/2 MS medium under a 12 h light/12 h dark photoperiod with a $100 \mu\text{M m}^{-2} \text{s}^{-1}$ light intensity for 28 days. The starch granules area and the ratio of stomatal aperture width to length from more than 100 guard cells of at least 4 different plants were measured using ImageJ software. EoN means the end of light, and Light 1 h means the white light illumination for 1 hour after the end of night. Scale bars in the pictures of stomata represent $20 \mu\text{m}$. Different letters above the bars indicate statistically significant differences between samples (Two-way ANOVA analysis followed by Tukey's multiple comparisons test, $p < 0.05$). **m** qRT-PCR analysis of the expression of *BAMI* in Col-0, *cat2*, *p35S::CAT2-myc* and *rbohD rbohF* plants. Seedlings of Col-0, *cat2*, *p35S::CAT2-myc* and *rbohD rbohF* were grown 1/2 MS medium under a 12 h light/12 h dark photoperiod with a $100 \mu\text{M m}^{-2} \text{s}^{-1}$ light intensity for 28 days. Stomata from rosette leaves were enriched through Stomata Tape-Peel method. Error bars indicate standard deviation (S.D.). Different letters above the bars indicate statistically significant differences between samples (Student t' test, $p < 0.05$).

- I have reservations about the phenotype of *kinb2* and overexpressors. I explain below:
- Extended fig. 5a: *kinb2* mutant has slightly more starch at the EoN, then degrades starch similarly to WT. Again, authors cannot talk about speed of degradation. This is anyway a very mild phenotype. They do not have data about double *kinb1b2*? Maybe there is redundancy. Anyway, in the over-expressor *35S::KINB2* where is localized KIN10? What happens if we treat *35S::KINB2* with H_2O_2 ? Conclusions are vague and do not fully reflect the presented data.

Response: Thank you for pointing this out. The overexpression of *KIN β 2* restricts the nuclear localization of KIN10 in guard cells (Supplementary Fig. 8a, b). Additionally, the *p35S::KIN β 2-YFP* lines exhibit a constant level of starch in guard cells before and after 1 hour light exposure, even under $30 \mu\text{M H}_2\text{O}_2$ condition. Moreover, the stomatal apertures in *p35S::KIN β 2-YFP* lines also failed to increase after light exposure (Supplementary Fig. 8c, d). These findings indicate that the nuclear localization of KIN10, which is controlled by KIN β 2, is required for guard cell starch degradation and stomatal opening. Consistently, *kin β 2* mutant exhibits slightly more starch under the end of night, less starch after 1 hour light exposure, and increased stomatal apertures. This further confirms the essential role of KIN10 nuclear localization in stomatal opening and starch metabolism (Supplementary Fig. 8e, f).

Supplementary Fig.8 KIN β 2 represses light-induced stomatal opening. **a, b** KIN β 2 restrict KIN10 protein nuclear localization in guard cell. *pKIN10::KIN10-YFP* and *pKIN10::KIN10-YFP/p35S::KIN β 2-RFP* plants were grown on 1/2 MS medium under a 12 h light/12 h dark photoperiod with the 100 $\mu\text{M m}^{-2} \text{s}^{-1}$ light intensity for 10 days. The ratio of KIN10-YFP nuclear and cytoplasmic signal intensity from at least 100 guard cells of 8 different plants were analyzed using ImageJ software. Scale bars represents 20 μm . Asterisk indicated statistically significant differences between the samples (Student's t test, **** $p < 0.0001$). **c, d** Quantification of guard cell starch granules (**c**) and stomatal apertures (**d**) in Col-0 and *p35S::KIN β 2-YFP* plants with or without H_2O_2 . Seedlings were grown on 1/2 MS medium under a 12 h light/12 h dark photoperiod with a 100 $\mu\text{M m}^{-2} \text{s}^{-1}$ light intensity for 10 days, then transferred to the medium with or without 30 μM of H_2O_2 to grow for 2 h before the end of night, and then harvested at the indicated time points. **e, f** Quantification of guard cell starch granules (**e**) and stomatal apertures (**f**) in Col-0, *kin β 2-1* and *kin β 2-2* plants. Seedlings of these plants were grown on 1/2 MS medium under a 12 h light/12 h dark photoperiod with the 100 $\mu\text{M m}^{-2} \text{s}^{-1}$ light intensity for 10 days. The starch granules area and the ratio of stomatal aperture width to length from more than 100 guard cells of at least 12 different plants were measured using ImageJ software. EoN means the end of light, and Light 1 h means the white light illumination for 1 h after the end of night. Different letters above the bars indicate statistically significant differences between samples (Two-way ANOVA analysis followed by Tukey's multiple comparisons test, $p < 0.05$).

- Extended fig. 4b: forced cytosolic localization of KIN10 has impact of amount of GC starch, accumulate less, so here as well it's an impact on synthesis, not degradation.

Response: Thank you for pointing this out. We created transgenic plants in the *kin10* mutant background with KIN10 fused to a nuclear localization signal sequence (NLS-KIN10) or a nuclear export signal sequence (NES-KIN10) at the N-terminus, under the control of the native *KIN10* promoter (Supplementary Fig. 7a). Gene expression analysis showed that both *pKIN10::KIN10-YFP/kin10* and *pKIN10::NLS-KIN10-YFP/kin10* successfully restored *BAM1* gene expression in *kin10* mutant. However, *pKIN10::NES-KIN10-YFP/kin10* failed to rescue the reduced *BAM1* expression in *kin10* mutant (Supplementary Fig. 7b). As expected, the *pKIN10::NLS-KIN10-YFP* plants exhibited lower guard cell starch content and increased stomatal apertures compared to wild-type plants and *pKIN10::KIN10-YFP/kin10* transgenic plants in response to light exposure. In contrast, the *pKIN10::NES-KIN10-YFP* plants showed cytosolic localization of KIN10 in guard cells, resulting in similar starch content and stomatal apertures as observed in the *kin10* mutant (Supplementary Fig. 7c, d).

Supplementary Fig. 7 The nuclear localized KIN10 is essential for the light-induced stomatal

opening. **a** Subcellular localization of KIN10 protein in guard cell of *pKIN10::KIN10-YFP/kin10*, *pKIN10::NES-KIN10-YFP/kin10* and *pKIN10::NLS-KIN10-YFP /kin10* transgenic plants. Scale bars represents 20 μm . **b**, qRT-PCR analysis of the expression of *BAMI* in Col-0, *kin10*, *pKIN10::KIN10-YFP/kin10*, *pKIN10::NES-KIN10-YFP/kin10* and *pKIN10::NLS-KIN10-YFP/kin10* plants. Plants of Col-0, *kin10*, *pKIN10::KIN10-YFP/kin10*, *pKIN10::NES-KIN10-YFP /kin10* and *pKIN10::NLS-KIN10-YFP /kin10* were grown 1/2 MS medium under a 12 h light/12 h dark photoperiod with a $100 \mu\text{M m}^{-2} \text{s}^{-1}$ light intensity for 28 days. Stomata from rosette leaves were enriched through Stomata Tape-Peel method. Error bars indicate standard deviation (S.D.). Different letters above the bars indicate statistically significant differences between samples (One-way ANOVA analysis followed by uncorrected Fisher's LSD multiple comparisons test, $p < 0.05$). **c, d** Quantification of guard cell starch granules (**c**) and stomatal apertures (**d**) in Col-0, *kin10*, *pKIN10::KIN10-YFP/kin10*, *pKIN10::NES-KIN10-YFP/kin10* and *pKIN10::NLS-KIN10-YFP/kin10* plants. Seedlings were grown on 1/2 MS medium under a 12 h light/12 h dark photoperiod with the $100 \mu\text{M m}^{-2} \text{s}^{-1}$ light intensity for 10 days. The starch granules area and the ratio of stomatal aperture width to length from more than 100 guard cells of at least 12 different plants were measured using ImageJ software. EoN means the end of light, and Light 1 h means the white light illumination for 1 h after the end of night. Different letters above the bars indicate statistically significant differences between samples (Two-way ANOVA analysis followed by Tukey's multiple comparisons test, $p < 0.05$).

- Lines 301-302. I have again reservation in the interpretation of the GC starch data. In Fig. 5a, Col-0 BL has less starch than Col-0 Mock, so it is difficult to conclude from these data that it degraded more. Same applies to stomatal opening. Col-0 BL has more open stomata at the EoN compared to Col-0 Mock.

Response: Thank you for pointing this out. We have revised the description of guard cell starch in the presence or absence of BL. Our finding indicated that BL treatment induces *BAMI* expression, which in turn reduces guard cell starch content and increases stomatal apertures. Our previous study has shown that BR-deficient mutant *det2* and BR-insensitive mutant *bzr-h* have high levels of undegradable starch in guard cells and impaired stomata opening (Li et al., 2020; Han et al., 2022). RT-qPCR analysis showed that *BAMI* expression is inhibited in both *det2* and *bzr-h* mutants (Han et al., 2022). These data revealed that BR and BZR1 function are required for starch degradation in guard cells and stomatal opening through regulating *BAMI* expression. On the other side, BL treatment could enhance *BAMI* promoter activity, but such promoting effects were reduced in both *kin10* and *bzip30* mutants (Supplementary Fig. 15a, b). Consequently, BL treatment leads to reduced starch content in guard cell under the end of night and

light exposure conditions.

Li, J. G., Fan, M., Hua, W., Tian, Y., Chen, L. G., Sun, Y., & Bai, M. Y. (2020). Brassinosteroid and Hydrogen Peroxide Interdependently Induce Stomatal Opening by Promoting Guard Cell Starch Degradation. *The Plant cell*, 32(4), 984–999.

Han, C., Hua, W., Li, J., Qiao, Y., Yao, L., Hao, W., Li, R., Fan, M., De Jaeger, G., Yang, W., & Bai, M. Y. (2022). TOR promotes guard cell starch degradation by regulating the activity of β -AMYLASE1 in Arabidopsis. *The Plant cell*, 34(3), 1038–1053.

Supplementary Fig. 15 The KIN10-bZIP30 module integrates BR and H₂O₂ signals to induce *BAMI* gene expression.

Mutations of KIN10 and bZIP30 inhibit BL induced *BAMI* expression. Transgenic plants *pBAM1::GFP*, *pBAM1::GFP/kin10* and *pBAM1::GFP/bzip30* were grown on 1/2 MS medium under 12 h light/12h dark photoperiod with 100 $\mu\text{M m}^{-2} \text{s}^{-1}$ for 10 days, and then treated with or without 100 nM BL for 3 hours. GFP signal from more than 100 guard cells in at least 10 different plants were analyzed by ImageJ software. Scale bars represent 20 μm . Different letters above the bars indicated statistically significant differences between the samples (Two-way analysis ANOVA followed by Tukey's multiple comparisons test, $p < 0.05$). Error bars indicate standard deviation (S.D.).

• Also the phenotype of kin10 and bzip30, already under control conditions they are impaired... it's not a conditional phenotype or it is not treatment dependent (H₂O₂ or BL treatment). Maybe results from a pleiotropic phenotype.

Response: We agree with the reviewer's assessment that *kin10* and *bzip30* mutants exhibit impaired stomatal opening and the inability to degrade guard cell starch in response to light exposure under normal growth conditions. However, our data demonstrates that treatment with H₂O₂ or BL can promote stomatal opening, but such promoting effects were significantly reduced in *kin10* and *bzip30* mutants, indicating

that both KIN10 and bZIP30 are required for H₂O₂- or BR-promoted stomatal opening.

Based on our findings, we propose that bZIP30 forms a heterodimer with BZR1 to facilitate *BAMI* expression and stomatal opening. Upon BL treatment, BZR1 undergoes dephosphorylation and translocation from the cytoplasm to the nucleus. H₂O₂ treatment not only induces the nuclear localization of KIN10, enhancing the phosphorylation of bZIP30, but also leads to the oxidative modification of BZR1. Phosphorylated bZIP30 by KIN10 and oxidation-modified BZR1 by H₂O₂ are more likely to form a heterodimer, resulting in the promoting stomatal opening, the induction of *BAMI* expression and the starch degradation.

- The third major concern I have is about the conservation of H₂O₂ distribution in vascular plants. The authors have not done enough experiments to come to this conclusion. So the title is inappropriate.

Response: We propose that H₂O₂ plays a crucial role in light-induced stomatal opening, and our study aimed to investigate the widespread nature of this mechanism across various plant species. Specifically, we examined the lycophyte *Selaginella doederleinii*, the fern *Marsilea quadrifolia*, and the angiosperms *Triticum aestivum* and *Arabidopsis thaliana*. Our findings revealed a significant accumulation of H₂O₂ in the guard cells of *Selaginella doederleinii*, *Marsilea quadrifolia*, *Triticum aestivum* and *Arabidopsis thaliana* under normal growth conditions. Removing H₂O₂ through KI or DDC treatment led to impaired stomatal opening when exposed to light.

In terms of molecular mechanisms, our work in *Arabidopsis* showed that bZIP30 and BZR1 transcriptional complex is essential for *BAMI* gene expression and stomatal opening. Therefore, we cloned the homolog genes of *BZR1*, *bZIP30* and *KIN10* from *Selaginella moellendorffii*. We found that H₂O₂ treatment facilitates the interaction between SmBZR1 and SmbZIP30 in tobacco leaves. Furthermore, H₂O₂ treatment induces SmKIN10 protein nuclear localization, which is consistent with KIN10 protein from *Arabidopsis* (Supplementary Fig. 18a-f). Protein alignment analysis revealed that KIN10-dependent phosphorylation sites of bZIP30, oxidative modification site of BZR1 and G-box binding sequences from *BAMI* promoter are widely present from

bryophyte to angiosperms. These findings suggested guard cell specific accumulated H_2O_2 in different plant species seems to apply a common mechanism to promote light-induced stomatal opening.

Although we acknowledge that the species of plants we investigated may not be representative of all vascular plants, our results indicate that H_2O_2 -mediated stomatal opening in response to light occurs at least in the studied plants. Therefore, to more accurately reflect the content of this manuscript, we have revised the title to: Hydrogen Peroxide is Required for Light-Induced Stomatal Opening across Different Plant Species

Supplementary Fig.18 H_2O_2 promotes nuclear localization of KIN10 homolog protein and the heterodimer of bZIP30 and BZR1 homolog protein from *Selaginella moellendorffii*. **a, b** H_2O_2 promotes SmbZIP30 and SmBZR1 interaction in tobacco leaves. The rBiFC construct of *p35S::SmBZR1-nYFP-p35S::RFP-p35S::SmbZIP30-cYFP* was transferred in tobacco leaves, and treated with 1 mM H_2O_2 for 3 h before observation. The fluorescent signals of YFP (BiFC) and RFP (reference) in the nucleus of tobacco epidermal cells were determined using ImageJ software. At least 50 interaction signals from more than 50 different epidermal cells were analyzed using ImageJ software. Asterisks between bars indicate statistically significant difference between samples (Student t' test, **** $p < 0.0001$). Scale bars in confocal images represent 20 μm . **c-f** H_2O_2 promotes AtKIN10 (**c, d**) and SmKIN10 (**e, f**) nuclear localization in tobacco leaves. AtKIN10-GFP and SmKIN10-GFP were transformed in tobacco leaves, then treated with or without 1 mM H_2O_2 for 3 h before observation. Nuclear and cytoplasmic fluorescent signals of AtKIN10-GFP or SmKIN10-GFP from at least 50 epidermal cells were measured using ImageJ. Scale bars in confocal images represent 20 μm .

• Fig. 6a. The pattern of H₂O₂ staining in *Marsilea* significantly differs from that of other species. This is not discussed.

Response: Thank you for pointing this out. H₂O₂ staining pattern revealed that H₂O₂ is accumulated in guard cells but not surrounded cells. However, H₂O₂ distribution in the cells depends on H₂O₂ content during laser exposure. To further confirm the H₂O₂ distribution pattern in epidermal cells of different plant species, we performed both BES-H₂O₂ and H₂DCFDA staining (Fig. 6a and Supplementary Fig. 17a-d). The results showed that H₂O₂ accumulated in guard cells but not in the cell surrounded stomata in different plant species.

Figure 6. H₂O₂ is required for light-induced stomatal opening across different plant species. a Staining of H₂O₂ in the epidermal cells of *Arabidopsis thaliana* (dicotyledonous), *Triticum aestivum* L (monocotyledonous), *Marsilea quadrifolia* (ferns) and *Selaginella doederleinii* (lycophytes) using H₂DCFDA. The stomata and surrounded subsidiary cells in abaxial epidermis of leaves in different plant species is highlighted by blue and green color respectively. Scale bars in confocal images for H₂O₂ staining represent 20 μm.

Supplementary Fig.17 Specific accumulated H₂O₂ in guard cell exists in different plant species.

BES-H₂O₂-Ac staining of H₂O₂ in the epidermal cells of *Arabidopsis thaliana* (dicotyledonous), *Triticum aestivum* (monocotyledonous), *Marsilea quadrifolia* (ferns) and *Selaginella doederleinii* (lycophytes). Scale bars in confocal images for H₂O₂ staining represent 20 μm.

- Lines 363-364. Based on my comment above, I do not agree that the pattern of H₂O₂ accumulation is conserved across vascular plants.

Response: Thank you for pointing this out. To investigate the widespread nature of H₂O₂ function in stomata, we check the H₂O₂ distribution in epidermis of different plant species. At least we observed that H₂O₂ accumulates in the guard cells of *Selaginella doederleinii*, *Marsilea quadrifolia*, *Triticum aestivum* and *Arabidopsis thaliana* under normal growth conditions. Removing H₂O₂ by KI or DDC treatment resulted in the impaired stomatal opening upon light exposure (Fig. 6). Moreover, our research showed that H₂O₂ can stimulate the translocation of the homologous protein of KIN10 from *Selaginella moellendorffii* to the nucleus. We also found that H₂O₂ can promote bZIP30 and BZR1 homologs from *Selaginella moellendorffii* to form a complex (Supplementary Fig. 18a-f). Protein alignment analysis revealed that KIN10 dependent phosphorylation sites of bZIP30, oxidation modification site of BZR1 and G-box binding sequence from *BAMI* promoter is widely exist from bryophyte to angiosperms. These data suggest that guard cell specific accumulated H₂O₂ in different plant species seems to apply a common mechanism to promote stomatal opening and starch metabolism, at least in the plants we have examined, indicating what we found in

Arabidopsis exists in other plant species widely.

- Lines 370-372. I also have strong reservations about their definition of spatial distribution, Normally, spatial distribution refers to subcellular distribution (e.g. cytosol vs chloroplasts). I disagree that the authors showed spatial pattern of H₂O₂. It's simply the accumulation of H₂O₂ to a certain concentration. Title does not reflect the data.

Response: Thank you for pointing this out. We changed the title as follow: Hydrogen Peroxide is Required for Light-Induced Stomatal Opening across Different Plant Species.

- One additional serious reservation is about the choice to study ROS by using the dye. Based on my personal experience, the H₂DCFDA is not reliable, and certainly risky to use it to quantify levels of H₂O₂. The intensity of the staining signal does not necessarily reflect the actual amount of H₂O₂. Nowadays, there are very powerful genetically encoded biosensors for ROS, which are much better suited to study changes in ROS amounts.

Response: Thank you for pointing this out. To provide evidence for the specific accumulation of H₂O₂ in leaf epidermis, we performed H₂DCFDA staining, BES-H₂O₂ staining and Hyper sensor analysis on mature rosette leaves of 4-week-old plants. These results indicated that H₂O₂ is specifically accumulation in guard cells under normal conditions. Additionally, we also analyze H₂O₂ content of guard cells in different plant species using BES-H₂O₂ staining method (Fig. 1a-f and Supplementary Fig. 17a-d).

Fig. 1 Specific accumulation of H₂O₂ in guard cells under normal conditions is required for light induced stomatal opening in Arabidopsis.

a-f Measurement of H₂O₂ in the epidermal cells on rosette leaves of four-weeks-old Col-0 plants

using H₂DCFDA staining (a, b), BES-H₂O₂-Ac staining (c, d) and HyPer fluorescent (e, f). Col-0 or *pHyPer* transgenic line were grown on 1/2 MS medium under a 12 h light/12 h dark photoperiod with a light intensity of 100 $\mu\text{M m}^{-2} \text{s}^{-1}$ for 28 days. The intensities of H₂DCFDA, BES-H₂O₂-Ac and HyPer fluorescent signals were analyzed using at least 100 guard cells from 10 different plants with ImageJ software. “PC” in confocal images represents pavement cell, “GC” represents guard cell. “Red” represents reduced state, “Ox” represents oxidized state. Scale bars represent 20 μm . Asterisk indicated statistically significant differences between the samples (Student’s t test, **** $p < 0.0001$).

Supplementary Fig.17 Specific accumulated H₂O₂ in guard cell exists in different plant species. BES-H₂O₂-Ac staining of H₂O₂ in the epidermal cells of *Arabidopsis thaliana* (dicotyledonous), *Triticum aestivum* (monocotyledonous), *Marsilea quadrifolia* (ferns) and *Selaginella doederleinii* (lycophytes). Scale bars in confocal images for H₂O₂ staining represent 20 μm .

- Lines 36-39. In amy3bam1 mutant, rate of proton pumping is similar that of WT, so the suggestion that Glc generated by starch degradation is necessary for energizing transport across the PM is not actually supported by the available data in the literature. Response: Sorry for this mistake. We have changed this sentence as follow: the *bam1amy3* mutant does not exhibit altered blue light-dependent ion transport, but delays the fast stomatal opening kinetics compared with wild-type plants, which are achieved by failure of replenishing carbohydrate pool from starch degradation during stomata opening.
- In experiments in Fig. 1, I am concerned about the use of KI. What are potential side

effects of KI?

Response: Thank you for pointing this out. To rule out any potential side effects of KI, we analyzed guard cell starch metabolism and stomatal movement by removing H_2O_2 through treatment with diethyldithiocarbamic acid (DDC), an inhibitor of dismutase and H_2O_2 production (Supplementary Fig. 1e-h). The results showed that DDC treatment reduced the H_2O_2 content in guard cells, inhibited guard cell starch degradation, and exhibited less stomatal apertures after 6 hours and 12 hours treatment. We also applied DDC treatment on different plant species, which also inhibits light-induced stomatal opening of *Triticum aestivum* and *Marsilea quadrifolia* plants (Figure 6d, e).

Supplementary Fig. 1 Reduction of H_2O_2 by KI or DDC treatment suppresses the light-induced stomatal opening.

e, f Measurement of H_2O_2 in guard cells of wild type cotyledon with or without Diethyldithiocarbamic acid (DDC) treatment using H_2DCFDA . Seedlings of Col-0 were grown on 1/2 MS medium under a 12 h light/12 h photoperiod with a $100 \mu M m^{-2} s^{-1}$ light intensity for 10 days, and then transferred to the medium containing mock solution or 1 mM DDC for different times before observation. **g, h** Quantification of stomatal apertures (**g**) and guard cell starch granules (**h**) in Col-0 plants with or without DDC treatment. Seedlings of Col-0 were grown on 1/2 MS medium under a 12 h light/12 h dark photoperiod with a $100 \mu M m^{-2} s^{-1}$ light intensity for 10 days, transferred to the medium containing mock solution or 1 mM DDC to grow for different times before the end of night, and then harvested at the indicated time points. H_2DCFDA signaling intensity from at least 100 guard cells of 8 different plants were analyzed using ImageJ software. Scale bars represents 50 μm . Different letters above the bars indicate statistically significant differences between samples (Two-way ANOVA analysis followed by Tukey's multiple comparisons test, $p < 0.05$). The starch granules area and the ratio of stomatal aperture width to length from more than 100 guard cells of at least 12 different plants were measured using ImageJ software. EoN means the end of light, and Light 1h means the white light illumination for 1 h after the end of night. Different letters above the bars indicate statistically significant differences between samples (Two-way ANOVA analysis followed by Tukey's multiple comparisons test, $p < 0.05$)

Figure 6. H₂O₂ is required for light-induced stomatal opening across different plant species. b-e Quantification of stomatal apertures in the leaves of *Triticum aestivum* (b, d) and *Marsilea quadrifolia* (c, e) with or without KI or DDC treatment. The leaves of *Triticum aestivum* and *Marsilea quadrifolia* were pre-incubated in the presence or absence of 1 mM KI or 1 mM DDC in the dark for 2 h before illumination with white light (100 μmol m⁻² s⁻¹) and for 3 h and 1 h respectively. The width of stomata of at least 50 guard cells from 8 different plants were measured using ImageJ software. Different letters above the bars indicate statistically significant differences between samples (Two-way ANOVA analysis followed by Tukey's multiple comparisons test, $p < 0.05$).

- Lines 131-134. Treatment with H₂O₂ and stomatal opening was done previously (their own work, ref 13). The claim that it's new is not correct.

Response: Thank you for pointing this out. We have changed the description as follow:
To investigate precise role of H₂O₂ in stomata movement, we conducted an analysis of stomatal apertures under varying concentrations of H₂O₂.

- kin10 is a sugar sensors, also very important for mesophyll cells. The authors do not show whether kin10 a starch and/or sugar phenotype under their conditions.

Response: Thank you for pointing this out. It has reported that *kin10kin11* double mutant, using virus-induced gene silencing, inhibits starch mobilization in leaves during night (Baena-Gonzalez et al., 2007). Additionally, the *kin10kin11(+/-)* partial SnRK1α loss-of-function mutant exhibits a lower sucrose content (Peixoto et al., 2021). Starch synthesis in guard cell is required for sugar transporting from mesophyll cells

(Flütsch et al., 2020). Therefore, the starch content in guard cells of *kin10* mutant is lower than that of wild-type plants.

Baena-González, E., Rolland, F., Thevelein, J. M. & Sheen, J. A central integrator of transcription networks in plant stress and energy signalling. *Nature* **448**, 938–942 (2007).

Peixoto, B. et al. Impact of the SnRK1 protein kinase on sucrose homeostasis and the transcriptome during the diel cycle. *Plant Physiol* **187**, 1357–1373 (2021).

Flütsch, S. et al. Glucose uptake to guard cells via STP transporters provides carbon sources for stomatal opening and plant growth. *EMBO Rep* **21**, e49719 (2020).

• KIN10 has an impact on GC starch degradation but we do not know if this is via H₂O₂ levels. What are the levels of H₂O₂ in *kin10*? And in *kin10cat2*?

Response: Thank you for pointing this out. To figure out whether *kin10* mutant regulate H₂O₂ levels, we performed BES-H₂O₂ staining on *kin10* and *kin10 cat2* mutants. The results showed that *kin10* mutant does not regulate H₂O₂ content in guard cells of either wild type plants or *cat2* mutant (Supplementary Fig. 5a, b).

Supplementary Fig. 5 KIN10 does not interfere H₂O₂ accumulates in the guard cells under normal condition. a, b BES-H₂O₂-Ac staining for H₂O₂ in guard cells on the rosette leaves of four-weeks-old Col-0, *cat2*, *kin10*, and *cat2 kin10* plants. These plants were grown on 1/2 MS medium under a 12 h light/12 h dark photoperiod with the 100 $\mu\text{M m}^{-2} \text{s}^{-1}$ light intensity for 28 days. Fluorescent signals were taken from more than 100 guard cells in rosette leaves of 10 different plants and analyzed using ImageJ software. Scale bars in confocal images represent 20 μm . Different letters above the bars indicate statistically significant differences between samples (One-way ANOVA analysis followed by Tukey's multiple comparisons test, $p < 0.05$).

- All these transgenics lines created with 35S promoter. Do we expect any side effects? Starch and sugars levels in the leaves?

Response: Thank you for pointing this out. To exclude side effects on KIN10 protein, we created transgenic plants in the *kin10* mutant background with KIN10 fused to a nuclear localization signal sequence (NLS-KIN10) or a nuclear export signal sequence (NES-KIN10) at the N-terminus, under the control of the native *KIN10* promoter (Supplementary Fig. 7a). Gene expression analysis showed that both *pKIN10::KIN10-YFP/kin10* and *pKIN10::NLS-KIN10-YFP/kin10* successfully restored *BAM1* gene expression in *kin10* mutant. However, *pKIN10::NES-KIN10-YFP/kin10* failed to rescue the reduced *BAM1* expression in *kin10* mutant (Supplementary Fig. 7b). As expected, the *pKIN10::NLS-KIN10-YFP* plants exhibited lower guard cell starch content and increased stomatal apertures compared to wild-type plants and *pKIN10::KIN10-YFP/kin10* transgenic plants in response to light exposure. In contrast, the *pKIN10::NES-KIN10-YFP* plants showed cytosolic localization of KIN10 in the guard cells, resulting in similar starch content and stomatal apertures as observed in the *kin10*

mutant (Supplementary Fig. 7c, d).

Supplementary Fig. 7 The nuclear localized KIN10 is essential for the light-induced stomatal opening.

a Subcellular localization of KIN10 protein in guard cell of *pKIN10::KIN10-YFP/kin10*, *pKIN10::NES-KIN10-YFP/kin10* and *pKIN10::NLS-KIN10-YFP /kin10* transgenic plants. Scale bars represents 20 μm . **b** qRT-PCR analysis of the expression of *BAMI* in Col-0, *kin10*, *pKIN10::KIN10-YFP/kin10*, *pKIN10::NES-KIN10-YFP/kin10* and *pKIN10::NLS-KIN10-YFP/kin10* plants. Plants of Col-0, *kin10*, *pKIN10::KIN10-YFP/kin10*, *pKIN10::NES-KIN10-YFP /kin10* and *pKIN10::NLS-KIN10-YFP /kin10* were grown 1/2 MS medium under a 12 h light/12 h dark photoperiod with a $100 \mu\text{M m}^{-2} \text{s}^{-1}$ light intensity for 28 days. Stomata from rosette leaves were enriched thought Stomata Tape-Peel method. Error bars indicate standard deviation (S.D.). Different letters above the bars indicate statistically significant differences between samples (One-way ANOVA analysis followed by uncorrected Fisher's LSD multiple comparisons test, $p < 0.05$). **c, d** Quantification of guard cell starch granules (**c**) and stomatal apertures (**d**) in Col-0, *kin10*, *pKIN10::KIN10-YFP/kin10*, *pKIN10::NES-KIN10-YFP/kin10* and *pKIN10::NLS-KIN10-YFP/kin10* plants. Seedlings were grown on 1/2 MS medium under a 12 h light/12 h dark photoperiod with the $100 \mu\text{M m}^{-2} \text{s}^{-1}$ light intensity for 10 days. The starch granules area and the ratio of stomatal aperture width to length from more than 100 guard cells of at least 12 different plants were measured using ImageJ software. EoN means the end of light, and Light 1 h means the white light illumination for 1 h after the end of night. Different letters above the bars indicate statistically significant differences between samples (Two-way ANOVA analysis followed by Tukey's multiple comparisons test, $p < 0.05$).

- Lines 230-231. Extended fig. 7a: treatment of WT GCs with H₂O₂. It's not clear that treatment enhanced GC starch degradation, as the levels of starch in WT treated with H₂O₂ are lower than Mock. It seems these results are not consistent with results presented in Extended Fig. 2D.

Response: Thank you for pointing this out. We applied different concentrations of H₂O₂ treatment, and found that 30 μM H₂O₂ treatment led to larger stomatal apertures and lower starch content in guard cells. This concentration of H₂O₂ was used for H₂O₂ treatment of *kin10* and *bzip30* mutants.

- Gene expression analyses were performed in entire seedlings, are not guard cell-specific. It's available an easy protocol to isolate GCs for qPCR analyses.

Response: Thank you for pointing this out. We have applicated Stomata Tape-Peel method (Lawrence et al., 2018) to enrich Arabidopsis epidermis and performed qRT-PCR experiment to detect *BAMI* and *AMY3* gene expression. The results showed that

the expression of *BAM1* and *AMY3* was significantly increased in the enriched epidermis samples compared to the mesophyll samples (Rebuttal Fig.1). Furthermore, within the stomata-enriched sample, *BAM1* and *AMY3* expression was decreased in *bzip30* and *kin10* mutants, which is consistent with our results using entire seedlings (Fig. 4a, b).

Lawrence, S., Pang, Q., Kong, W. & Chen, S. Stomata tape-peel: an improved method for guard cell sample preparation. *J Vis Exp* **137**, 57422 (2018).

Rebuttal Fig.1 *BAM1* and *AMY3* are preferentially expressed in stomata enriched samples. qRT-PCR analysis the expression of *BAM1* (a) and *AMY3* (b) in Col-0 plants using guard cell enriched epidermal samples and mesophyll cell samples. Seedlings of Col-0 were grown 1/2 MS medium under a 12 h light/12 h photoperiod with a $100 \mu\text{M m}^{-2} \text{s}^{-1}$ light intensity for 28 days. Stomata from rosette leaves were enriched thought Stomata Tape-Peel method. Error bars indicate standard deviation (S.D.). Asterisks between bars indicate statistically significant difference between samples (Student t' test, $**p < 0.01$).

Figure 4. bZIP30 induces *BAM1* and *AMY3* expression. a, b qRT-PCR analysis of the expression of *BAM1* and *AMY3* in Col-0, *kin10* and *bzip30* plants. Plants of Col-0, *kin10* and *bzip30* were grown on 1/2 MS medium under a 12 h light/12 h dark photoperiod with a $100 \mu\text{M m}^{-2} \text{s}^{-1}$ light intensity for 28 days. Stomata from rosette leaves were enriched thought Stomata Tape-Peel method. Error bars indicate standard deviation (S.D.). Asterisks between bars indicate statistically significant

difference between samples (Two-way ANOVA analysis followed by uncorrected Fisher's LSD multiple comparisons test, * $p < 0.05$, ** $p < 0.01$, *** $p < 0.001$ and **** $p < 0.0001$).

- In general, how do the author reconcile that fact that both AMY3 and BAM1 are redox-regulated enzyme, when they suggest that oxidation modify BZR1 to activate transcription? I cannot imagine how these two degrading enzymes can work under what the authors suggest to be oxidizing conditions.

Response: Thank you for pointing this out. Based on our previous study, we found that H₂O₂ is required for BZR1 to promote cell elongation and gene expression under normal growth conditions (Tian et al., 2018). These findings suggest that the oxidative modification of BZR1 necessitates a low concentration of H₂O₂, which is inadequate to deactivate BAM1 and AMY3 oxidation.

Tian, Y. et al. (2018). Hydrogen peroxide positively regulates brassinosteroid signaling through oxidation of the BRASSINAZOLE-RESISTANT1 transcription factor. *Nat Commun* **9**, 1063 (2018).

- Discussion is repetitive and lack depth.

Response: Thank you for pointing this out. Based on other reviewer's suggestion, we have shortened the evolution analysis on H₂O₂ in guard cells and have added more discussion about the functions of H₂O₂, including does dependent behavior, tissues or cell types specific distribution, and conservation role of H₂O₂ in stomatal opening across different plant species.

- Line 103. It's already known that the effect is concentration dependent.

Response: Thank you for pointing this out. We have changed this as follow: However, the precise role of H₂O₂ accumulation in guard cells on stomatal movement during normal growth conditions has yet been fully understood.

- Line 122. It's the PROPER accumulation of H₂O₂ that is important.

Response: Thank you for pointing this out. We have revised the statement as follow: These results suggested that a mild accumulation of H₂O₂ in guard cells is required for

light-induced stomatal opening.

- Line 128. It's not rate, they just did one time point.

Response: Thank you for pointing this out. We have revised the statement as follow: Conversely, the *cat2* mutant showed the decreased starch content in guard cells under both the end of night and 1h light exposure conditions.

- Lines 145-148. It not the spatial accumulation, it's the amount of H₂O₂.

Response: Thank you for pointing this out. We have revised the statement as follow: These results suggested that a mild accumulation of H₂O₂ in guard cells under normal growth conditions is necessary for light-induced guard cell starch degradation and stomatal opening.

- Analyses of GC starch and stomatal aperture of *kin10* and *cat2* mutant in Fig. 2 is redundant.

Response: Thank you for pointing this out. The guard cell starch and stomatal aperture of the *kin10* and *cat2* mutants in Figure 2 serve as the control for the *kin10 cat2* double mutant. This control provides evidence that KIN10 is required for H₂O₂-induced guard cell starch degradation and stomatal opening upon light exposure.

- Experiments in Extended fig. 3: DAPI staining?

Response: Thank you for pointing this out. We have generated the *pKIN10::KIN10-YFP/p35S::Histone 3-RFP* line, where Histone 3-RFP serves to indicate the position of nucleus.

- Figure legend Fig. 3a, a, KIN10 interacts with bZIP30 in yeast. This is uninformative. Please write a proper figure legend.

Response: Thank you for pointing this out. We have added the proper description of Y2H in figure legend as follow: Yeast two-hybrid assay shows the interaction between KIN10 and bZIP30. BD-KIN10 was used as the bait. Co-transforming BD-KIN10 with

empty-AD vector serves as the negative control. Yeast transformants were grown on the SD/-His/-Leu/-Trp selection medium for 3 days before photographing.

- Material and methods are incomplete.

Response: Thank you for pointing this out. We have added the missing experimental methods, such as transient gene expression assays, the detail of H₂O₂ quantification and epidermal peel enrichment for qRT-PCR analysis.

- English is very poor.

Response: Thank you for pointing this out. We have invited a native English speaker to assist with revising the language.

Reviewer #2 (Remarks to the Author):

In a previous manuscript (Shi et al 2022; Nature Comms. 13: 5040), the authors of this current manuscript established an exciting connection between H₂O₂ distribution and stomata development. This current manuscript extends their previous discovery by studying the role of H₂O₂ distribution on stomata function – particularly stomata opening in response to light. They demonstrate that endogenous H₂O₂ levels in guard cells (which they manipulate using genetics as well as pharmacological agents) affects the ability of stomata to open in response to light. They demonstrate that H₂O₂-mediated stomata regulation requires KIN10, the catalytic subunit of SnRK kinase. They present a model where H₂O₂ induces KIN10 movement into the nucleus in guard cells, where it phosphorylates a bZIP30 transcription factor. Using mutants, they convincingly demonstrate that bZIP30 is important for stomata opening. They also show that the phosphorylated bZIP30 interacts with BZR1, and they act together to activate AMY3/BAM1 expression. These two enzymes have been previously shown to activate guard cell starch degradation.

Overall, the experiments are of high quality. The manuscript is complete in that the authors have put together a full, plausible pathway – from the signalling molecule H₂O₂, all the way to the expression of starch degradation enzymes. The authors also provide convincing evidence that nuclear localisation of KIN10 in guard cells is important for guard cell regulation, and the implication of bZIP30 in stomata regulation is also novel. For these reasons, I think this is a strong manuscript, but I have a few questions/issues (mainly regarding presentation) for the authors to consider.

Response: Thanks for your positive comments.

Major points:

1. Lines 71-79: I felt this part of the introduction could benefit from revision. I do not think it is completely necessary to mention all the effects of H₂O₂ in the development

of apical meristems. This part could probably be shortened. However, the reader would benefit from a clear definition of what is meant by “spatial pattern” or “spatial accumulation” of H₂O₂ throughout the manuscript. Clearly state what spatial scale is relevant - sub-cellular, cell-type, tissue level or organ level.

Response: Thank you for pointing this out. We have revised the introduction of the manuscript to provide a shorter description of the effects of H₂O₂ on apical meristem development, while placing more emphasis on the spatial distribution of H₂O₂ at the tissues level and cellular level in plants. Specifically, we focus on the specific distribution of H₂O₂ in meristemoid cells and guard cells within the leaf epidermis. However, since the aim of this study is to explain the function of guard cell accumulated H₂O₂, we have decided to remove “spatial pattern” description from the revised manuscript.

2. Line 111: A word of caution regarding KI treatments: Potassium iodide influences H₂O₂ levels, but it is also a potent starch stain. It is one of the key ingredients of Lugol’s solution. It is possible that KI can block starch degradation due to it complexing with the starch itself. For this reason, I find the genetic manipulations of H₂O₂ (e.g., the analysis of *cat2* mutants) more convincing than the pharmacological treatments. I am not saying such experiments should be removed, but I think the possible effects of KI on the starch should be directly discussed in the manuscript, unless the authors can prove that KI has no effect on starch degradation *in vivo*.

Response: Thank you for your insightful comment. We appreciate your caution regarding the use of KI treatments and its potential influence on both H₂O₂ levels and starch staining. You raise a valid point about the possibility of KI affecting starch degradation by forming complexes with the starch itself, especially considering its role in Lugol's solution. To further investigate the effects of H₂O₂ on guard cell starch degradation and stomatal opening, we applied diethyldithiocarbamic acid (DDC), which is inhibitor for dismutase and H₂O₂ production, to analyze starch degradation and stomatal opening (Supplementary Fig. 1e-h). The results showed that DDC treatment reduces the H₂O₂ content in guard cells, inhibits guard cell starch degradation

and results in smaller stomatal apertures after 6 hours and 12 hours treatment.

Supplementary Fig. 1 Reduction of H₂O₂ by KI or DDC treatment suppresses the light-induced stomatal opening.

e, f Measurement of H₂O₂ in guard cells of wild type cotyledon with or without Diethyldithiocarbamic acid (DDC) treatment using H₂DCFDA. Seedlings of Col-0 were grown on 1/2 MS medium under a 12 h light/12 h photoperiod with a 100 µM m⁻² s⁻¹ light intensity for 10 days, and then transferred to the medium containing mock solution or 1 mM DDC for different times before observation. **g, h** Quantification of stomatal apertures (**g**) and guard cell starch granules (**h**) in Col-0 plants with or without DDC treatment. Seedlings of Col-0 were grown on 1/2 MS medium under a 12 h light/12 h dark photoperiod with a 100 µM m⁻² s⁻¹ light intensity for 10 days, transferred to the medium containing mock solution or 1 mM DDC to grow for different times before the end of night, and then harvested at the indicated time points. H₂DCFDA signaling intensity from at least 100 guard cells of 8 different plants were analyzed using ImageJ software. Scale bars represents 50 µm. Different letters above the bars indicate statistically significant differences between samples (Two-way ANOVA analysis followed by Tukey’s multiple comparisons test, *p* < 0.05). The starch granules area and the ratio of stomatal aperture width to length from more than 100 guard cells of at least 12 different plants were measured using ImageJ software. EoN means the end of light, and Light 1h means the white light illumination for 1 h after the end of night. Different letters above the bars indicate statistically significant differences between samples (Two-way ANOVA analysis followed by Tukey’s multiple comparisons test, *p* < 0.05)

3. Lines 132-147: Here, I think the authors need to better explain what is defined as a “normal” and “stress” concentration. Also, Lines 132-133 probably need a citation.

Response: Thank you for pointing this out. To avoid confusion, we have changed it as follow: To investigate precise role of H₂O₂ in stomata movement, we conducted an analysis of stomatal apertures under varying concentrations of H₂O₂ in wild-type plants.

4. Lines 181-182: It is striking that the nuclear localisation of KIN10 changes in the cat2 and rbohD rbohF mutants. However, this is the KIN10 localisation adopted after

long-term exposure to altered H₂O₂ levels. However, the paper focuses on more dynamic changes that lead to rapid stomata opening in response to light. In this regard, I wonder whether the authors have tried simply treating the plants with different H₂O₂ concentrations, as in the experiment of Extended Data Fig 2, and assessed whether the KIN10 localisation in the nucleus is most pronounced in the 10-30 μ M range?

Response: Thank you for pointing this out. We have analyzed the localization of KIN10 in response to different concentrations of H₂O₂ using *pKIN10::KIN10-YFP* transgenic plants (Supplementary Fig. 6 e,f). The result showed that treatment with exogenous H₂O₂ at varying concentrations all leads to an increased nuclear-to-cytoplasmic ratio of KIN10 protein. H₂O₂ ranging from 30 μ M H₂O₂ to 1 mM H₂O₂ induced a similar nuclear/cytoplasmic ratio of KIN10 protein, indicating that the nuclear localization of KIN10 protein is sensitive to H₂O₂. However, it is worth noting that high concentrations of H₂O₂ in the millimolar range can inhibit the kinase activity of KIN10 (Wurzinger et al., 2017). Therefore, an appropriate amount of H₂O₂ is necessary to induce the nuclear localization of KIN10 protein and maintain a high level of KIN10 activity for phosphorylating bZIP30 in guard cells.

Wurzinger, B. et al. Redox state-dependent modulation of plant SnRK1 kinase activity differs from AMPK regulation in animals. *FEBS Lett* **591**, 3625–3636 (2017).

Supplementary Fig. 6 H₂O₂ promotes nuclear localization of KIN10 in guard cells.
e, f H₂O₂ induces the nuclear localization of KIN10 in guard cells of plants. Seedlings of *pKIN10::KIN10-YFP* were grown on 1/2 MS medium under a 12 h light/12 h dark photoperiod with the 100 μ M m⁻² s⁻¹ light intensity for 10 days and then treated with or without different concentrations of H₂O₂ for 6 hours. The ratio of KIN10-YFP nuclear and cytoplasmic signal intensity from at least 100 guard cells of 10 different plants were analyzed by ImageJ software. Scale bars represents 20 μ m. Different letters above the bars indicate statistically significant

differences between samples (One way ANOVA analysis followed by Uncorrected Fisher's LSD multiple comparisons test, $p < 0.05$).

5. Lines 202-206: The introduction of bZIP30 is quite abrupt. The authors say they chose to focus on it because it was in the Y2H and it was related to bZIP29 (which is known to be expressed in stomata). If this is the case, I am wondering why they did not test for an interaction between KIN10 and bZIP29? I am guessing that bZIP29 was not present in the Y2H screen with KIN10, but this needs to be clearly stated.

Response: Thank you for pointing this out. Indeed, we performed a yeast two-hybrid (Y2H) screening to identify proteins that interact with KIN10, and we found only bZIP30, not bZIP29, as an interacting partner. Considering the specific expression pattern of *bZIP29* in stomata, we decided to concentrate our investigation on bZIP30, which is the homologous gene of *bZIP29*. In the revised manuscript, we have provided a comprehensive explanation of the selection process.

6. Line 220: What is meant by "universal"? The images in Extended Data Fig 6a-d are too small to see the localisation clearly, and the cell area plots need to be explained in the text. What are they meant to show?

Response: Thank you for pointing this out. We apologize for any confusion caused by the term "universal". In this context, it refers to the observation of fluorescent signals from *pbZIP30::bZIP30-YFP* in all leaf epidermal cells. However, these signals showed a high nuclear-to-cytoplasmic ratio specifically in guard cells and stomatal lineage cells. To better illustrate this result, we have generated new images to display the distribution pattern of bZIP30 in the epidermal cells. Additionally, we changed protein nuclear-to-cytoplasmic ratio in different cell area to nuclear-to-cytoplasmic ratio of different cell types (Supplementary Fig. 9a-c).

Supplementary Fig.9 bZIP30 and KIN10 possesses similar protein localization pattern in Arabidopsis epidermis and promotes stomatal development.

a-c, The subcellular locations of *pbZIP30::bZIP30-YFP* and *pKIN10::KIN10-YFP* are similar in Arabidopsis cotyledon epidermal cells. Seedlings of *pbZIP30::bZIP30-YFP* and *pKIN10::KIN10-YFP* were grown in 1/2 MS medium under a 12 h light/12 h dark photoperiod with the $100 \mu\text{M m}^{-2} \text{s}^{-1}$ light intensity for 4 days. bZIP30-YFP and KIN10-YFP are in green, PI-marked cell outlines are in purple. Scale bars in confocal images represent 20 μm. Scatter plots showed bZIP30-YFP (b) and KIN10-YFP (c) exhibit similar nuclear-to-cytoplasmic ratio in different types of epidermal cells. “SLC” indicates stomatal lineage cells, “PC” indicates pavement cells, “GC” indicates guard cells. Different letters above the bars indicate statistically significant differences between samples (One-way ANOVA analysis followed by Uncorrected Fisher’s LSD multiple comparisons test, $p < 0.05$).

7. Lines 246-248: Here it is stated that the phosphomimic mutant only “rescues” the defective stomatal opening and guard cell starch degradation phenotypes of *bzip30*. However, why does it only “rescue”? Wouldn’t the authors expect that this phosphomimic bZIP30 is constitutively active, and should rescue beyond the WT version? Indeed, in Fig 3i and 3j, the phosphomimic lines have an EoN stomata aperture and starch contents that are statistically different from the Col-0. This should be discussed.

Response: Thank you for pointing this out. We also expect that phosphomimic bZIP30 would be more active. In terms of starch degradation, *bZIP30^{S18D/T22D}* was slightly more effective from EoN to 1 h light exposure. However, *bZIP30^{S18D/T22D}* did not exhibit larger stomatal apertures compared to bZIP30 rescue lines. The possibility is that *bZIP30^{S18D/T22D}* and bZIP30 possess similar activity, indicating that bZIP30 can be phosphorylated by KIN10 effectively.

8. Line 265: In this assay, BAM1 and AMY3 activation is indeed further enhanced when KIN10 and bZIP30 are co-expressed. But why is KIN10 alone able to very substantially activate BAM1 and AMY3 expression? Could it be working with an endogenous bZIP30 in the cells? If the phosphomimic form of bZIP30 is used in this assay, would it match the expression levels achieved with KIN10+bZIP30? Also, I could not find the methods related to this experiment (I am very sorry if I missed it) – in what system were these proteins transiently expressed?

Response: Thank you for pointing this out. We have analyzed the effects of bZIP30^{S18D/T22D} on *BAM1* and *AMY3* gene expression in transient expression system. The results showed that bZIP30^{S18D/T22D} also induces *BAM1* and *AMY3* gene expression, similar to the effect of wild-type bZIP30 (Fig. 4d, Supplementary Fig 13b). This suggests that bZIP30 protein can be effectively phosphorylated by KIN10 in plant cells.

Figure 4. bZIP30 induces *BAM1* and *AMY3* expression. **d** Transient assays showed that the expression of *BAM1* was induced by KIN10-YFP, bZIP30-YFP and bZIP30^{S18/T22D}-YFP, but not by bZIP30^{S18/T22A}-YFP. The promoter of *BAM1* fused to the luciferase reporter gene were co-transfected with KIN10-YFP, bZIP30-YFP, bZIP30^{S18/T22A}-YFP or bZIP30^{S18/T22D}-YFP into mesophyll protoplasts of wild-type plants. The luciferase activities were normalized with *Renilla* luciferase as an internal control. Error bars represent standard deviation (S.D.). Different letters above the bars indicate statistically significant differences between samples (One-way ANOVA analysis followed by uncorrected Fisher's LSD multiple comparisons test, $p < 0.05$).

Supplementary Fig.13 bZIP30 directly induces *AMY3* expression.

b Transient assays showed that the expression of *AMY3* was induced by KIN10-YFP, bZIP30-YFP and bZIP30^{S18/T22D}-YFP, but not by bZIP30^{S18/T22A}-YFP. The promoter of *AMY3* fused to the luciferase reporter gene was co-transfected with KIN10-YFP, bZIP30-YFP, bZIP30^{S18/T22A}-YFP or bZIP30^{S18/T22D}-YFP into mesophyll protoplasts of wild-type plants. The luciferase activities were normalized with *Renilla* luciferase as an internal control. Error bars represent standard deviation (S.D.). Different letters above the bars indicate statistically significant differences between samples (One-way ANOVA analysis followed by uncorrected Fisher's LSD multiple comparisons test, $p < 0.05$).

Minor points:

In general, I find that this manuscript needs a thorough proofread for spelling and minor grammatic errors. I can only mention a few here:

Line 7: omit "the"

Line 11: "prefers" sounds like the protein makes an active choice to be there.

Line 23: environments

Line 23: "are surrounded"

Line 32: "the guard cell"

Line 33: "on the other side"

Line 56: "Under insufficient energy supply"

Line 73: This sentence should not start with "And".

Line 80: “epidermis”

Line 99: “previous”

Line 284: “interacts”

Line 318: “on the other side”

Line 325: “which could help induce BAM1...”

Line 399: “degradation”

Line 425: omit “of”

Line 447: omit “does”

Line 535: “pbZIP30:bZIP30”

Response: Thank you for your careful review and assistance in improving our manuscript. We apologize for the spelling errors and have corrected all the mistakes you mentioned. Additionally, we have enlisted the help of a native speaker to assist with revising the languages.

Reviewer #3 (Remarks to the Author):

The MS by Shi et al explores the potential distribution and role of H₂O₂ in starch degradation in guard cells and the impact on stomatal aperture. A considerable amount of work has gone into the manuscript using a range of approaches and techniques and a great deal of data are presented, however there are still some major concerns with the MS. Additionally, due to the large number of approaches and questions addressed, the MS is difficult to follow in places and lacks focus. This would reduce appeal to a broad audience expected by this journal.

The rationale for the importance of this study is not clear, the importance of the findings and conclusions are not put into context. There is a lack of convincing evidence for many of the statements made and this is acknowledged by the authors who use phrases such as "...this might suggest" and "...this might show. More details regarding this are highlighted below.

Response: Thank you for pointing this out. As you and other reviewer suggested, we have restructured the introduction and discussion sections of this manuscript to align with our result. Additionally, we have enlisted the assistance of a native speaker to help revise the languages to ensure clarity and ease of understanding.

The MS has a focus (including the title) on spatial distribution of H₂O₂ – the imaging processes reporting this are not entirely clear or convincing and also not quantified apart from a few small images.

Response: Thank you for pointing this out. We have created new figures to show rosette leaves image and how to measure the H₂O₂ intensity in different types of epidermal cells. Additionally, we have included a detailed description of the quantification process in Material and Methods section.

Rebuttal Fig. 2 The schematic diagram of method for the quantification of fluorescent intensity in H₂DCFDA and BES-H₂O₂-Ac staining. The average gray values based on fluorescent signals of H₂DCFDA staining, BES-H₂O₂-Ac staining or pHHyper from the whole stomata or the pavement cells were selected and quantified using ImageJ software.

The abstract introduces a number of genes, proteins, TF that are not explained and expects the reader to have a clear understanding of the interactions between all of these.

Response: Thank you for pointing this out. We have revised the abstract to include a proper introduction to different enzymes and proteins, making it easy to understand.

KI was used to reduced H₂O₂ – how can the authors be completely convinced that the amount and entry into the tissue was identical between experiments or uniform across tissues and experiments?

Response: Thank you for pointing this out. The issue you mentioned is indeed a common challenge with chemical treatment. To address this, we applied H₂DCFDA to detect H₂O₂ content and assess the effectiveness of the treatment. Additionally, we employed genetic manipulations of H₂O₂ using different mutants involved in H₂O₂ metabolism. The results supported our conclusion that the level of H₂O₂ in stomata is crucial for stomata opening and starch degradation in guard cells.

Line 25 – “.believed to “ – the role of stomata is well established in plant physiology in leaves and green tissue.

Response: Thank you for pointing this out. We have changed it as follow: it has been well established that stomata in vascular plants open to acquire carbon dioxide and close to prevent water loss.

The authors only consider stomatal responses to blue light – however this is not the only pathway for guard cell responses to light, or starch degradation and furthermore the response to blue light is not conserved across species.

Response: Thank you for pointing this out. We have included red light-induced starch metabolism in guard cells in the introduction section.

There are several paragraphs in the introduction that are focussed on a small part of the MS – e.g the section on the early land plants. This is not the focus of the paper, however it is a large focus of the MS.

Response: Thank you for pointing this out. We have changed this part of introduction.

Line 68 –“ forces ancient plant to adapt the more abundant...”?L

Response: Thank you for pointing this out. The purpose of this sentence is to introduce the idea that ancient plant, upon landing, encountered increased oxygen levels, which can easily transfer to ROS in cells. Therefore, various types of plant cells, including guard cells, have developed molecular mechanism to adapt to ROS.

The MS needs editing for English and grammar – and there are numerous odd phrases used e.g. line 33 “one the other side” – on the other hand?; Line 55 “Under energy supplies insufficient” – unclear.

Response: Thank you for pointing this out. We apologize for the spelling and grammatic errors. We have corrected the mistakes you mentioned here. And we have enlisted the help of a native speaker to assist in revising the languages.

Line 82 – which ROS scavenging genes.

Response: Thank you for pointing this out. We have changed this part of introduction as follow: SPCH is responsible for establishing the H₂O₂ spatial pattern in leaf epidermis during stomatal development by regulating the expression of *Ascorbate Peroxidase 1* and *Catalase 2 (CAT2)*.

The images suggest that the ROS is accumulating in the guard cell chloroplasts. What are the wavelengths of detection for the H₂DCFDA? The staining is not in all cells in some of the images e.g. Extended data Fig.2 – explain.?

Response: Thank you for pointing this out. The H₂O₂ staining images represents lots of dots in guard cells, some of which are colocalized with chloroplasts autofluorescence. We used excitation 488 nm and emission 500–530 nm for H₂DCFDA staining to avoid capturing chloroplasts autofluorescence. We think the reason why the staining is not observed in all cells can be attributed to two points. First, H₂O₂ is generated in subcellular compartments, including chloroplasts, which may not to spread uniformly throughout the entire cell. Second, the levels of H₂O₂ may not be abundant enough, as the ROS scavenging system continuously work to remove H₂O₂. However, under stress conditions, liking treatment with 1mM exogenous H₂O₂, H₂O₂ can be detected in all parts of guard cells.

rhobD and F need to be introduced to the reader.

Response: Thank you for pointing this out. We have included a comprehensive description of these genes.

It is not clear if images and apertures were taken on the same material – the material and methods suggest not. This is not appropriate as it is clear that there is significant heterogeneity within the images. Furthermore, there is limited quantification of images to directly relate activities to apertures.

Response: Thanks for pointing this out. We conducted measurements of the H₂O₂ content, stomatal apertures, starch granules area in guard cells by analyzing at least 100 guard cells from 4-10 rosette or cotyledon leaves across 4-12 different plants. Furthermore, we have included images depicting starch granules and stomatal opening of guard cells in the revised manuscript. These images are selected from the images we captured to illustrate our measurements.

It would be useful to see images of the starch granules – rather than just the quantification of these.

Response: Thank you for pointing this out. We have included images depicting starch granules of guard cells in the Fig1 of revised manuscript (Fig 1i-l). These images are selected from the images we captured to illustrate our measurements.

Fig. 1 Specific accumulation of H_2O_2 in guard cells under normal conditions is required for light induced stomatal opening in Arabidopsis.

i-l Quantification of stomatal apertures (**i, j**) and guard cell starch granules (**k, l**) in rosette leaves of Col-0, *cat2*, *p35S::CAT2-myc* and *rbohD rbohF* plants. Plants were grown on 1/2 MS medium under a 12 h light/12 h dark photoperiod with a $100 \mu\text{M m}^{-2} \text{s}^{-1}$ light intensity for 28 days. The starch granules area and the ratio of stomatal aperture width to length from more than 100 guard cells of at least 4 different plants were measured using ImageJ software. EoN means the end of light, and Light 1 h means the white light illumination for 1 hour after the end of night. Scale bars in the pictures of stomata represent $20 \mu\text{m}$. Different letters above the bars indicate statistically significant differences between samples (Two-way ANOVA analysis followed by Tukey's multiple comparisons test, $p < 0.05$).

Line 128 – “accelerated breakdown” or less observed when measured. Was there a control at the start to ensure that breakdown was accelerated?

Response: Thank you for pointing this out. We have revised the description of *cat2* mutant starch phenotype in guard cells as follow. “Conversely, the *cat2* mutant exhibited less content of guard cell starch under both dark and 1 hour light exposure conditions”.

The authors do not provide any information about where the H₂O₂ is generated in non-stress tissues and how this differs from stress conditions when stomatal closure is induced by H₂O₂.

Response: Thank you for pointing this out. There are several potential sources of H₂O₂ generation in guard cells under normal conditions. The expression patterns of ROS scavenging genes *CAT2* and *ASCORBATE PEROXIDASE 1 (APX1)* in Arabidopsis epidermal cells show lower levels in guard cells and stomatal lineage cells, but higher levels in pavement cells, indicating their potential role in H₂O₂ scavenging in guard cell (Shi et al., 2022). Another possibility for H₂O₂ accumulated in guard cells may be related to cell-specific metabolic characteristics. Guard cells require a high demand for energy in the form of ATP to facilitate the stomatal movement, which is due to the large number of ion channels and transporters involved in these processes (Raghavendra, 1981). ROS, including H₂O₂, are generated by the NADH dehydrogenase of the respiratory chain Complex I. Guard cells have a higher number of mitochondria and greater oxidative phosphorylation ability compared to mesophyll cells (Vani et al., 1994; Mawson, 1993), This may explain why H₂O₂ specifically accumulates in guard cells.

We propose that H₂O₂ regulates stomatal movement in a dose-dependent manner, promoting stomatal opening at low concentrations and inducing stomatal closure at high concentrations. Under normal growth conditions, the H₂O₂ content in guard cells is at low levels, promoting stomatal opening. Under stress conditions, H₂O₂ accumulates at high levels, resulting in stomatal closure. To further verify this, we analyzed the stomatal apertures with different concentrations of exogenous H₂O₂ treatment. The results showed that low H₂O₂ concentration treatment (ranging from 10

μM to $50 \mu\text{M}$) promotes stomatal opening, while high H_2O_2 concentration treatment (from $100 \mu\text{M}$ to 1mM , known to induce stress effects on many physiology processes, such as seed germination and seedling growth) inhibits stomatal opening.

Shi, W. et al. Spatially patterned hydrogen peroxide orchestrates stomatal development in Arabidopsis. *Nature Commun* **13**, 5040 (2022).

Raghavendra A. S. (1981). Energy supply for stomatal opening in epidermal strips of commelina benghalensis. *Plant Physiol* **67**, 385–387 (1981).

Vani, T., & Raghavendra, A. S. High Mitochondrial Activity but Incomplete Engagement of the Cyanide-Resistant Alternative Pathway in Guard Cell Protoplasts of Pea. *Plant Physiol* **105**, 1263–1268 (1994).

Mawson, B. T. Modulation of photosynthesis and respiration in guard and mesophyll cell protoplasts by oxygen concentration. *Plant Cell Environ.* **16**, 207–214 (1993).

Lione 141 – there is no figure for stress conditions?

Response: Thank you for pointing this out. This study primarily examines the role of H_2O_2 in promoting stomatal opening under normal growth conditions, without specifically studying its function under stress conditions. Therefore, we have removed the description about the stress to maintain the focus on the promotion of H_2O_2 in stomatal opening.

Extended data set 2 – the spatial variation in H_2O_2 production is not convincing in these images and there is limited quantification of this.

Response: Thank you for pointing this out. The purpose of H_2O_2 staining in this figure was to detect H_2O_2 content in guard cells after different concentration of exogenous H_2O_2 treatment.

Line 152 – “a hypothesis regarding ...” – what was the hypothesis?

Response: Sorry for this misleading. We have revised the sentence as follow: Therefore, we hypothesized that the specific accumulation of H_2O_2 in guard cells under normal growth conditions promotes light-induced guard cell starch degradation and stomatal opening by controlling KIN10 activity.

Line 160+ - loss of Kin10 – prevented starch breakdown but was it because the starch was low to start with?

Response: Thank you for pointing this out. Indeed, the *kin10* mutant exhibits lower levels of starch in guard cells at the end of night. However, the guard cell starch does not degrade after 1 h light exposure. The qRT-PCR experiment revealed that the *kin10* mutant showed the decreased *BAMI* and *AMY3* gene expression. Therefore, the *kin10* mutant inhibits starch breakdown in guard cells.

Can the authors confirm that the transgenic and mutants used have similar expression patterns and activities in all tissues measured? This could account for some of the spatial variation and general variation in measurements?

Response: Thank you for pointing this out. In this experiment, transgenic plants with comparable expression levels of *KIN10* were selected for subsequent experiments. To further investigate the effects of different subcellular localization on KIN10 function, we created transgenic plants in the *kin10* mutant background with KIN10 fused to a nuclear localization signal sequence (NLS-KIN10) or a nuclear export signal sequence (NES-KIN10) at the N-terminus, under the control of the native *KIN10* promoter (Supplementary Fig. 7a). Gene expression analysis showed that both *pKIN10::KIN10-YFP/kin10* and *pKIN10::NLS-KIN10-YFP/kin10* successfully restored *BAMI* gene expression in *kin10* mutant. However, *pKIN10::NES-KIN10-YFP/kin10* failed to rescue the reduced *BAMI* expression in *kin10* mutant (Supplementary Fig. 7b). As expected, the *pKIN10::NLS-KIN10-YFP* plants exhibited lower guard cell starch content and increased stomatal apertures compared to wild-type plants and *pKIN10::KIN10-YFP/kin10* transgenic plants in response to light exposure. In contrast, the *pKIN10::NES-KIN10-YFP* plants showed cytosolic localization of KIN10 in the guard cells, resulting in similar starch content and stomatal apertures as observed in the *kin10* mutant (Supplementary Fig. 7c, d).

Supplementary Fig. 7 The nuclear localized KIN10 is essential for the light-induced stomatal opening. **a** Subcellular localization of KIN10 protein in guard cell of *pKIN10::KIN10-YFP/kin10*, *pKIN10::NES-KIN10-YFP/kin10* and *pKIN10::NLS-KIN10-YFP/kin10* transgenic plants. Scale bars represents 20 μm . **b**, qRT-PCR analysis of the expression of *BAM1* in Col-0, *kin10*, *pKIN10::KIN10-YFP/kin10*, *pKIN10::NES-KIN10-YFP/kin10* and *pKIN10::NLS-KIN10-YFP/kin10* plants. Plants of Col-0, *kin10*, *pKIN10::KIN10-YFP/kin10*, *pKIN10::NES-KIN10-YFP/kin10* and *pKIN10::NLS-KIN10-YFP/kin10* were grown 1/2 MS medium under a 12 h light/12 h dark photoperiod with a $100 \mu\text{M m}^{-2} \text{s}^{-1}$ light intensity for 28 days. Stomata from rosette leaves were enriched through Stomata Tape-Peel method. Error bars indicate standard deviation (S.D.). Different letters above the bars indicate statistically significant differences between samples (One-way ANOVA analysis followed by uncorrected Fisher's LSD multiple comparisons test, $p < 0.05$). **c**, **d** Quantification of guard cell starch granules (**c**) and stomatal apertures (**d**) in Col-0, *kin10*, *pKIN10::KIN10-YFP/kin10*, *pKIN10::NES-KIN10-YFP/kin10* and *pKIN10::NLS-KIN10-YFP/kin10* plants. Seedlings were grown on 1/2 MS medium under a 12 h light/12 h dark photoperiod with the $100 \mu\text{M m}^{-2} \text{s}^{-1}$ light intensity for 10 days. The starch granules area and the ratio of stomatal aperture width to length from more than 100 guard cells of at least 12 different plants were measured using ImageJ software. EoN means the end of light, and Light 1 h means the white light illumination for 1 h after the end of night. Different letters above the bars indicate statistically significant differences between samples (Two-way ANOVA analysis followed by Tukey's multiple comparisons test, $p < 0.05$).

Although aperture was measured, for full functional assessment – physiology would be required. As there is change in both density and function – one could simply be compensating for the other.

Response: Thank you for pointing this out. We analyzed the stomatal density of *kin10* and *bzip30*. Both mutants possess less stomatal density and decreased stomatal apertures. Therefore, stomatal density does not compensate the deficiency of stomata opening.

YFP signal could be influenced by many physical factors within the leaf as well as the measurement approaches and is therefore not entirely reliable.

Response: Thank you for pointing this out. We acknowledge that the YFP signal can be affected by various physical factors within the leaf. To address this concern, we performed additional experiments to validate our findings and ensure the reliability of our results. These included quantitative RT-PCR analysis of gene expression levels and Western blot analysis of protein levels. We also performed transient transformation bZIP30-GFP, bZIP30^{S18/T22A}-GFP and bZIP30^{S18/T22D}-GFP with KIN10-RFP in tobacco leaves. The bZIP30 and related bZIP30 mutations exhibit similar localization with what we observed of bZIP30-YFP in Arabidopsis epidermis. By employing these complementary techniques, we think the possibility that YFP signal influenced by physical factor in leaf can be excluded.

The later part of the MS focuses on the interactions between various proteins and TF and uses a range of approaches to assess them – this feels and reads like a different paper, and is often not linked to function.

Response: Sorry for any confusion. The purpose of the experiments in this section is to demonstrate that H₂O₂ promotes stomatal opening by inducing the formation of the BZR1-bZIP30 transcription complex. H₂O₂ treatment not only triggers the nuclear localization of KIN10, which leads to the phosphorylation of bZIP30, but also induces the oxidative modification of BZR1. Phosphorylated bZIP30 by KIN10 and oxidation-

modified BZR1 by H₂O₂ are more inclined to form a heterodimer, resulting in the induction of *BAMI* expression, consequently promoting starch degradation and stomatal opening.

The discussion section does not fully reflect the findings and focuses on the evolutionally aspects of the MS, which is the simplest and least explored apart from the H₂O₂ potentially been detected and the same genes being present. The discussion should focus on the main findings and put these into context. This is another major weakness of the MS.

Response: Thank you for pointing this out. We have revised the discussion section to reduce the evolution aspect and provide more insight into the function of H₂O₂ in guard cells. Additionally, we have included a more detailed discussion on how H₂O₂ is generated in guard cells.

Line 426 – it has been clearly shown that CO₂ fixation does take place in guard cells and that all the enzymes are present. The authors need to correct this to not mislead readers.

Response: Thank you for pointing this out. In our revision manuscript, this part of discussion was excluded.

Line 427 – although NTTT were shown to be important in this paper – application of DCMU also showed that there was a role for GC electron transport.

Response: Thank you for pointing this out. In our revision manuscript, this part of discussion was excluded.

REVIEWER COMMENTS

Reviewer #1 (Remarks to the Author):

I appreciate the authors effort in revising the manuscript by performing additional experiments and adjusting the text according to my suggestions.

The current version of the manuscript is much improved.

I appreciate that the authors performed a series of experiments on leaves of mature plants to address one of my major concerns. However, I still have reservation concerning the choice of the plant material: growing plants on ½ MS is not ideal to study guard cell performance, as transpiration on agar plate (presumably sealed to avoid contamination) is lower than in atmospheric air when plant are grown in soil. This may significantly affect the physiology of the plant under normal day-night growth conditions.

In addition, there are differences between seedlings and mature plants in terms of stomatal behaviour, which need to be acknowledged in the text. For example:

- Amounts of starch are remarkably different with seedlings having elevated guard cell starch contents compared to mature leaves (approx. 100 μm^2 in seedlings vs. 25 μm^2 in mature leaves).
- Line 159. The GC starch phenotype of *cat2* mutant is different at seedling stage compared to mature leaves. In Fig. 1I, *cat2* mutant has similar starch amounts as WT at the EoN, but it degrades more than WT. In Suppl Fig. 2d, *cat* mutant has less starch than WT at the EoN but degrades similarly to WT. The difference is the age of the plant (mature vs seedlings). This is not discussed, but it confirms my concern that the age of the plant does indeed influences some aspects of stomatal behavior.
- Line 198: “*cat2* mutants exhibited less starch content in guard cells” it’s not what I see in fig. 2f, in which starch content at the end of the night in *cat2* mutant is similar to wild type as confirmed by the statistical analysis. However, as shown in Fig. 1I, *cat2* mutants degrades more starch than wt upon transition to light.

Additional minor points:

- Line 23 Abstract. What does it mean “exchange gases and fitness to terrestrial habitat”? There must be a language mistake.
- Line 146, please correct: *cat2* mutants has more opened stomata compared to WT, but they open less in response to light compared to WT.
- Line 338: spelling mistake, nuclus \rightarrow nucleus.

- Line 488-489: the sentence “stomata are small opening pores on the surfaces of plant tissues..” is not necessary in the discussion. Remove.

Reviewer #2 (Remarks to the Author):

The authors have sufficiently addressed my concerns. I thank the authors for considering them.

Reviewer #3 (Remarks to the Author):

The authors have made a good job of addressing the majority of the comments, resulting in significant improvements in the flow and accessibility of the manuscript and its findings. This effort is particularly crucial given the substantial volume of data presented, utilizing a diverse range of approaches and techniques.

Ensuring clarity and coherence in presenting findings is paramount, and the authors have succeeded in enhancing the manuscript's readability and comprehensibility. By incorporating feedback and refining the organization of the manuscript, they have effectively facilitated the understanding of the research outcomes. There are still a number of minor points that need clarifying that will help guide the reader through the rationale.

Line 23 ...”fitness to terrestrial habitats” – what exactly does this mean?

Line 26 – what are “normal” growing conditions?

Line 46 - ...”close in response to light, CO₂ and...” – need to say close in decrease light and high CO₂ concentration – [CO₂] rather than CO₂.

Line 48 – stomata are plural – “The stomata are opened..” = correct English. Stomata open when guard cell.....

Line 52 – red light “accelerates” photosynthesis – do the authors mean increase?

Line 556 – “On the other hand” – other hand to what – remove?

RBOH mutants still need fully explained or introduced to the readers.

What is mild H₂O₂?

Where is the ROS generated or how is the ROS generated?

What does too much H₂O₂ do?

Line 162 – did BAM expression change at the whole plant levels or just guard cells level. Also was photosynthesis altered in these plants?

When gene expression is mentioned it should be made clear which tissues these results are from.

Line 188 and elsewhere in the MS – it is not always clear from the text if the apertures are larger than controls? It is good to see the inclusion of the schematic diagrams of example aperture.

Line 198 – less starch in the guard cells because it has been broken down? Clarify.

Line 234 – 236 - provide further explanation as to why the starch levels are the same. Guide the reader through the rationale.

Line 325 – Transient expression does not usually work or influence guard cells because they are symplastically isolated. The authors state transient expression was observed in epidermal cells – can they clarify if it was in the guard cells?

The discussion should include some indication if the role of H₂O₂ in stomatal opening is confined to only the starch breakdown aspects of stomatal opening. It is well established that there are other possible pathways and osmotica involved in osmoregulation where the impact of H₂O₂ may not be apparent. For example starch degradation is linked to morning opening and blue light driven opening.

Line 489 – “Stomata are small pores” – this seem completely in the wrong place and more introductory text.

M&M – how were cells sizes determined?

Reviewer #1 (Remarks to the Author):

I appreciate the authors effort in revising the manuscript by performing additional experiments and adjusting the text according to my suggestions.

The current version of the manuscript is much improved.

Response: Thanks for your positive comments.

I appreciate that the authors performed a series of experiments on leaves of mature plants to address one of my major concerns. However, I still have reservation concerning the choice of the plant material: growing plants on ½ MS is not ideal to study guard cell performance, as transpiration on agar plate (presumably sealed to avoid contamination) is lower than in atmospheric air when plant are grown in soil. This may significantly affect the physiology of the plant under normal day-night growth conditions.

Response: Thanks for pointing this out. In order to further investigate the involvement of H₂O₂ in light-induced stomatal opening and guard cell starch degradation in rosette leaves of soil-grown plants, we conducted an analysis of stomatal apertures and guard cell starch levels in Col-0 plants grown on soil for 4 weeks. The results revealed that plants grown on soil exhibit similar stomatal apertures with the plants grown on 1/2 MS medium. Although starch content in guard cells of plants grown on soil is lower than that of plants grown on 1/2 MS medium, starch in guard cells is also degraded after light exposure when plant grown on soil (Rebuttal figure 1). Moreover, we have performed analysis of H₂O₂ content and starch aperture for wheat seedlings grown on soil in figure 6. The results showed that H₂O₂ also specific accumulated in guard cells (Fig 6a). The application of KI and DDC also inhibits the stomatal opening after light exposure in wheat grown on soil condition (Fig 6b, d). These findings closely resemble those observed in rosette leaves grown on the 1/2 MS medium, indicating that the significant role of H₂O₂ in light-induced stomatal opening and starch degradation is consistent across leaves under varying growth conditions.

Rebuttal Fig 1. Different growth conditions did not influence stomatal behavior in Arabidopsis. Quantification of stomatal apertures (a) and guard cell starch granules (b) in rosette leaves of Col-0 plants grown in different conditions. Col-0 plants were grown on 1/2 MS medium or soil under a 12 h light/12 h dark photoperiod with a $100 \mu\text{M m}^{-2} \text{s}^{-1}$ light intensity for 28 days. The starch granules area and the ratio of stomatal aperture width to length from more than 100 guard cells of at least 4 different plants were measured using ImageJ software. EoN means the end of light, and Light 1 h means the white light illumination for 1 h after the end of night. Different letters above the bars indicate statistically significant differences between samples (Two-way ANOVA analysis followed by Tukey's multiple comparisons test, $p < 0.05$).

In addition, there are differences between seedlings and mature plants in terms of stomatal behaviour, which need to be acknowledged in the text. For example:

- Amounts of starch are remarkably different with seedlings having elevated guard cell starch contents compared to mature leaves (approx. $100 \mu\text{m}^2$ in seedlings vs. $25 \mu\text{m}^2$ in mature leaves).

Response: Thanks for pointing this out. We indeed observed variations in starch content between the cotyledon and rosette leaves, and we have included a discussion of this phenomenon in the revised manuscript's discussion section.

“Firstly, the starch amount in guard cells is quite different between cotyledon of seedlings and mature leaves. Starch synthesis in guard cells depends on sugar transport from mesophyll cells. In seedlings, there are only two apical meristems (shoot and root) consuming carbon sources. In contrast, mature plants develop more sink tissues, including a larger number of stomata, to consume nutrients. Consequently, a single guard cell in mature plants has less carbon source available for starch synthesis.

Importantly, our data indicates that plants with different genetic constructs exhibit similar starch dynamics after light exposure in both seedling and mature leaves.”

- Line 159. The GC starch phenotype of *cat2* mutant is different at seedling stage compared to mature leaves. In Fig. 1I, *cat2* mutant has similar starch amounts as WT at the EoN, but it degrades more than WT. In Suppl Fig. 2d, *cat* mutant has less starch than WT at the EoN but degrades similarly to WT. The difference is the age of the plant (mature vs seedlings). This is not discussed, but it confirms my concern that the age of the plant does indeed influence some aspects of stomatal behavior.

Response: Thanks for pointing this out. We have indeed observed discrepancies in the starch content between the cotyledon and rosette leaves of *cat2* mutants at the EoN and upon light exposure. We have included a discussion of this phenomenon in the revised manuscript's discussion section.

“Differences were observed in the starch degradation rate and starch content at the EoN in the guard cells of the *cat2* mutant between cotyledon and rosette leaves. Accumulation of H₂O₂ was detected in the guard cells of the *cat2* mutant compared to wild type plants, potentially affecting starch synthesis ability. The demand for starch synthesis is greater in cotyledons than in rosette leaves. As a result, the *cat2* mutant, with reduced starch synthesis ability, only shows significant differences compared to the wild type in cotyledons, where more starch accumulation is needed due to increased carbon source availability.”

- Line 198: “*cat2* mutants exhibited less starch content in guard cells” it’s not what I see in fig. 2f, in which starch content at the end of the night in *cat2* mutant is similar to wild type as confirmed by the statistical analysis. However, as shown in Fig. 1I, *cat2* mutants degrade more starch than wt upon transition to light.

Response: Thanks for pointing this out. We have changed this as follows: whereas the *cat2* mutant exhibited less starch content in guard cells and larger stomatal apertures after light exposure.

Additional minor points:

- Line 23 Abstract. What does it mean “exchange gases and fitness to terrestrial habitat”?

There must be a language mistake.

Response: Thanks for pointing this out. We have changed this as follow: Stomatal movement is vital for plants to exchange gases and adaption to terrestrial habitats.

- Line 146, please correct: *cat2* mutants has more opened stomata compared to WT, but they open less in response to light compared to WT.

Response: Thanks for pointing this out. We have changed this as follow: The results showed that *cat2* mutants displayed larger stomatal apertures and exhibit the weaker response to light exposure compared to wild-type plants.

- Line 338: spelling mistake, nuclus □ nucleus.

Response: Sorry for this spell mistake, we have changed this.

- Line 488-489: the sentence “stomata are small opening pores on the surfaces of plant tissues..” is not necessary in the discussion. Remove.

Response: Thanks for pointing this out. We have removed this sentence.

Reviewer #2 (Remarks to the Author):

The authors have sufficiently addressed my concerns. I thank the authors for considering them.

Response: Thanks for your positive comments.

Reviewer #3 (Remarks to the Author):

The authors have made a good job of addressing the majority of the comments, resulting in significant improvements in the flow and accessibility of the manuscript and its findings. This effort is particularly crucial given the substantial volume of data presented, utilizing a diverse range of approaches and techniques.

Response: Thanks for your positive comments.

Ensuring clarity and coherence in presenting findings is paramount, and the authors have succeeded in enhancing the manuscript's readability and comprehensibility. By incorporating feedback and refining the organization of the manuscript, they have effectively facilitated the understanding of the research outcomes. There are still a number of minor points that need clarifying that will help guide the reader through the rationale.

Line 23 ...”fitness to terrestrial habitats” – what exactly does this mean?

Response: Thanks for pointing this out. We have changed this as follow: Stomatal movement is vital for plants to exchange gases and adaption to terrestrial habitats.

Line 26 – what are “normal” growing conditions?

Response: Thanks for pointing this out. Normal condition we mentioned here means the condition which is optimal for plant growth compared to stress condition. We have changed this as follow: H₂O₂ accumulates specifically in guard cells even when plants under unstressed conditions.

Line 46 - ...”close in response to light, CO₂ and....” – need to say close in decrease light and high CO₂ concentration – [CO₂] rather than CO₂.

Response: Thanks for pointing this out. We have changed this as follow: It has been well established that stomata in vascular plants open to acquire carbon dioxide in response to light and close to prevent water loss in response to high concentration of CO₂ and the phytohormone abscisic acid (ABA)

Line 48 – stomata are plural – “The stomata are opened..” = correct English. Stomata

open when guard cell.....

Response: Thanks for pointing this out. We have changed this as follow: The stomata are opened when the volumes of guard cells are changed by exposure to light

Line 52 – red light “accelerates” photosynthesis – do the authors mean increase?

Response: Thanks for pointing this out. We have changed this as follow: red light enhances photosynthesis and causes a decrease of intercellular CO₂ concentration in leaves

Line 56 – “On the other hand” – other hand to what – remove?

Response: Thanks for pointing this out. We have removed this.

RBOH mutants still need fully explained or introduced to the readers.

Response: Thanks for pointing this out. We have changed this as follow: mutants of RBOHD and RBOHF, which encode respiratory burst NADPH oxidase homologs (RBOHs) and are responsible for producing apoplastic H₂O₂ during plant defense response.

What is mild H₂O₂?

Where is the ROS generated or how is the ROS generated?

What does too much H₂O₂ do?

Response: Thanks for pointing this out. Here the mild H₂O₂ we mentioned is the specific accumulated H₂O₂ in guard cells when plants growing under unstressed condition. This endogenous H₂O₂ in guard cells may from the increased number of mitochondria and greater oxidative phosphorylation ability compared to mesophyll cells (Vani et al., 1994; Mawson et al.,1993), and also might be involved in lower expression levels of ROS scavenging genes (Shi et al., 2022). Considering the difficulty for quantifying the amount of H₂O₂ in guard cells precisely, we have changed this as follow: These results suggested that accumulation of H₂O₂ in guard cells under unstressed conditions is required for light-induced stomatal opening.

Vani, T. & Raghavendra, A.S. High mitochondrial activity but incomplete engagement of the cyanide-resistant alternative pathway in guard cell protoplasts of pea. *Plant Physiol.* 105, 1263-1268 (1994).

Mawson, B. T. Modulation of photosynthesis and respiration in guard and mesophyll cell protoplasts by oxygen concentration. *Plant Cell Environ.* 16, 207–214 (1993).

Shi, W., et al. Spatially patterned hydrogen peroxide orchestrates stomatal development in *Arabidopsis*. *Nature Commun.* 13(1), 5040 (2022).

Line 162 – did BAM expression change at the whole plant levels or just guard cells level. Also was photosynthesis altered in these plants? When gene expression is mentioned it should be made clear which tissues these results are from.

Response: Thanks for pointing this out. *BAMI* is expressed in guard cells specifically. And we apply leaves epidermis enrichment for RT-PCR analysis. Therefore, the change of *BAMI* expression among different mutants is mainly occurred in guard cells level. We changed this as follow: *BAMI* is responsible for guard cell starch degradation and expressed in guard cells specifically. Quantitative RT-PCR analysis using guard cell enriched samples showed that the expression levels of *BAMI* in guard cells significantly increased in the *cat2* mutant, but decreased in *p35S::CAT2-myc* and *rbohD rbohF* mutants.

Line 188 and elsewhere in the MS – it is not always clear from the text if the apertures are larger than controls? It is good to see the inclusion of the schematic diagrams of example aperture.

Response: Thanks for pointing this out. We applied the violin plots to represent the collected data of stomatal apertures and starch granule areas. Solid lines of violin plots represent the Median, which can be used for distinguishing the difference between samples. In order to make it easy to follow, we add this introduction of detail of violin plots in material and methods.

Line 198 – less starch in the guard cells because it has been broken down? Clarify.

Response: Thanks for pointing this out. We are sorry for this incorrect description of the data in fig 2f. We have changed this as follow: whereas the *cat2* mutant exhibited less starch content in guard cells and larger stomatal apertures after light exposure.

Line 234 – 236 - provide further explanation as to why the starch levels are the same. Guide the reader through the rationale.

Response: Thanks for pointing this out. We have added further explanation as follow: These data suggested that the overexpression of *KINβ2* prevents H₂O₂ induced KIN10 nuclear localization, nuclear KIN10 promoted stomatal opening and guard cell starch degradation.

Line 325 – Transient expression does not usually work or influence guard cells because they are symplastically isolated. The authors state transient expression was observed in epidermal cells – can they clarify if it was in the guard cells?

Response: Thanks for pointing this out. We applied transient expression of bZIP30 protein in the pavement cells of tobacco leaves to analyze the protein subcellular localization. And we also analyze the subcellular localization of bZIP30 and mutated bZIP30 in guard cells using Arabidopsis transgenic lines.

The discussion should include some indication if the role of H₂O₂ in stomatal opening is confined to only the starch breakdown aspects of stomatal opening. It is well established that there are other possible pathways and osmotica involved in osmoregulation where the impact of H₂O₂ may not be apparent. For example starch degradation is linked to morning opening and blue light driven opening.

Response: Thanks for pointing this out. We have revised that related discussion section as follow: Starch degradation in guard cells is required for H₂O₂ promoted stomatal opening upon light exposure. In this study, we have showed that removal H₂O₂ content through applying DDC or KI inhibits starch degradation in guard cell. Exogenous treatment of H₂O₂ with low concentration promotes starch degradation and *BAMI* expression, which is required for the function of KIN10 and bZIP30. Our previous study

also showed that the mutations of *BAMI* and *AMY3* repress H₂O₂ induced stomatal opening. All these data suggested that guard cell starch degradation is one of the reasons for H₂O₂ promoted stomatal opening. And it is also worth to test whether H₂O₂ with low concentration regulates the activity of ion channels, including AHA or S-Type anion channels.

Line 489 – “Stomata are small pores” – this seem completely in the wrong place and more introductory text.

Response: Thanks for pointing this out. We have removed this sentence.

M&M – how were cells sizes determined?

Response: Thanks for pointing this out. We have added additional section about how to quantify stomatal index and number and measurement of cells sizes.

REVIEWERS' COMMENTS

Reviewer #1 (Remarks to the Author):

I have no further comments - thanks for addressing all my concerns.

Reviewer #3 (Remarks to the Author):

The authors have attempted to make further corrections to the paper – however the paper is still not well written. There are also still a number of misleading sentences – I have highlighted some of these below.

Logical flow and presentation of arguments also still need improving. Several sentences are just statements of facts rather than providing a logical flow and background to guide the reader through the rationale.

Some statements also seem irrelevant for the paper.

Line 23 sentence does not make sense.

Line 51 – implies that blue light signaling is the only signal that opens stomata – misleading
Red light responses and mechanisms not entirely known.

There is still no indication where the ROS is produced and by what mechanism.

Some of the results have still not been clarified – e.g.

“The results showed that cat2 mutants displayed larger stomatal apertures and 148 exhibit the weaker response to light exposure compared to wild-type plants,” - what is a weaker response to light ? Also if the stomata are already open then the stimuli to open further may be compromised.

Some of the results are over interpreted. For example “These findings indicated that H₂O₂ promotes the light induced starch degradation in guard cells through inducing the expression of BAM1” QRT-PCR cannot demonstrate this unequivocally – it only provides a measure of transcript abundance and not activity.

In general the conclusions are mostly based on correlative analysis which does not demonstrate causation.

The discussion needs improving in terms of logical flow – for example there is still a section in the middle that talks of stomatal evolution which seems to be completely in the wrong place. Also there are many sections that are repetitious of the results and or the discussion.

REVIEWERS' COMMENTS

Reviewer #1 (Remarks to the Author):

I have no further comments - thanks for addressing all my concerns.

Response: Thanks for your positive comments.

Reviewer #3 (Remarks to the Author):

The authors have attempted to make further corrections to the paper – however the paper is still not well written. There are also still a number of misleading sentences – I have highlighted some of these below.

Logical flow and presentation of arguments also still need improving. Several sentences are just statements of facts rather than providing a logical flow and background to guide the reader through the rationale.

Some statements also seem irrelevant for the paper.

Response: Thanks for pointing this out. We sincerely appreciate the valuable comments.

Line 23 sentence does not make sense.

Response: Thanks for pointing this out. We have changed this as follow: Stomatal movement is vital for plants to exchange gases and adaption to terrestrial habitats, which is regulated by environmental and phytohormonal signals.

Line 51 – implies that blue light signaling is the only signal that opens stomata – misleading. Red light responses and mechanisms not entirely known.

Response: Thanks for pointing this out. We have changed this as follow: The stomata are opened when the volumes of guard cells are changed by exposure to light. One factor contributing to this phenomenon is the hyperpolarization of the plasma membrane driven by H⁺-ATPase (AHA) and blue light signaling.

There is still no indication where the ROS is produced and by what mechanism.

Response: Thanks for pointing this out. ROS can be produced in different subcellular organelles of plant cells. Here we showed that specific cellular level of H₂O₂ accumulated in guard cell and is required for stomatal opening and guard cell starch degradation. The expression patterns of *CAT2* and *APX1* in Arabidopsis epidermal cells exhibits the lower level in guard cells (Shi et al., 2022), suggesting a possible reason for the specific accumulation of H₂O₂ in guard cells. The identification of the specific transcription factors responsible for inhibiting the low expression of *CAT2* and *APX1* in guard cells remains to be explored. Another possibility for the accumulation of H₂O₂ in guard cell may be related to their specific metabolic characteristics. Guard cells require a high demand for energy in the form of ATP to facilitate the stomatal movement, which is due to the large number of ion channels and transporters involved in these processes (Lim et al., 2022; Raghavendra., 1981). ROS, including H₂O₂, are generated by the NADH dehydrogenase of the respiratory chain Complex I. Guard cells have a higher number of mitochondria and greater oxidative phosphorylation ability compared to mesophyll cells (Vani et al., 1994; Mawson, 1993). This may explain why H₂O₂ specifically accumulates in guard cells. However, further research is needed to understand how guard cells accumulate H₂O₂ under normal growth conditions.

Shi, W., Wang, L., Yao, L., Hao, W., Han, C., Fan, M., Wang, W., & Bai, M. Y. (2022). Spatially patterned hydrogen peroxide orchestrates stomatal development in Arabidopsis. *Nature communications*, 13(1), 5040.

Lim, S. L., Flütsch, S., Liu, J., Distefano, L., Santelia, D., & Lim, B. L. (2022). Arabidopsis guard cell chloroplasts import cytosolic ATP for starch turnover and stomatal opening. *Nature communications*, 13(1), 652.

Raghavendra A. S. (1981). Energy supply for stomatal opening in epidermal strips of *Commelina benghalensis*. *Plant physiology*, 67(2), 385–387.

Vani, T., & Raghavendra, A. S. (1994). High mitochondrial activity but incomplete engagement of the cyanide-resistant alternative pathway in guard cell protoplasts of pea. *Plant physiology*, 105(4), 1263–1268.

Mawson, B. T. (1993). Modulation of photosynthesis and respiration in guard and mesophyll cell protoplasts by oxygen concentration. *Plant Cell Environ.* 16, 207–214.

Some of the results have still not been clarified –

e.g. “The results showed that *cat2* mutants displayed larger stomatal apertures and 148 exhibit the weaker response to light exposure compared to wild-type plants,” - what is a weaker response to light ? Also if the stomata are already open then the stimuli to open further may be compromised.

Response: Thanks for pointing this out. We have changed this as follow: The results showed that *cat2* mutant displayed larger stomatal apertures under both the end of night and 1 hour light exposure conditions. Therefore, *cat2* mutant exhibit the weaker response to light exposure compared to wild-type plants.

Some of the results are over interpreted. For example “These findings indicated that H₂O₂ promotes the light induced starch degradation in guard cells through inducing the expression of BAM1” QRT-PCR cannot demonstrate this unequivocally – it only provides a measure of transcript abundance and not activity.

Response: Thanks for pointing this out. This conclusion is not only from qRT-PCR analysis, but also the starch metabolism phenotypes of H₂O₂ related mutant. Phenotype data showed that impaired H₂O₂ in guard cells overexpression of *CAT2*, mutations of *RBOHD* and *RBOHF* or treatment with KI or DDC leads to accumulated and undegradable starch in guard cells. BAM1 is one of key enzyme for guard cell starch degradation. Through qRT-PCR analysis, overexpression of *CAT2*, mutations of *RBOHD* and *RBOHF* cause less abundant of BAM1. Therefore, we concluded H₂O₂ promotes the light-induced starch degradation in guard cells through inducing the expression of *BAM1*. To further exclude this misleading, we changed this as follow: These findings based on starch metabolism phenotype and qRT-PCR analysis indicated that H₂O₂ promotes the light induced starch degradation in guard cells through inducing the expression of *BAM1*.

In general the conclusions are mostly based on correlative analysis which does not demonstrate causation. The discussion needs improving in terms of logical flow – for

example there is still a section in the middle that talks of stomatal evolution which seems to be completely in the wrong place. Also there are many sections that are repetitious of the results and or the discussion.

Response: Thanks for your suggestion. We have reorganized the sequence of different paragraphs in discussion section. Firstly, we summarized our work. And then we discussed the importance of H₂O₂ does-dependent manner in regulating stomata functions. And we discussed spatial accumulated H₂O₂ in different organ and analyze why H₂O₂ specific accumulated in guard cells. Secondly, we discussed H₂O₂ induced starch degradation is important for stomatal opening and starch dynamic is influenced by stage of leaf development. At last, we discussed that H₂O₂ induced stomatal opening is conserved among different species and basal lineage of vascular plants. And we talk about the function of KIN10-bZIP30-BZR1 module on stomatal opening is conserved among different plant species.